# On the Variance, Admissibility, and Stability of Empirical Risk Minimization

**Gil Kur**
EECS
MIT
gilkur@mit.edu

**Eli Putterman**
Mathematics Department
Tel Aviv University
putterman@mail.tau.ac.il

**Alexnader Rakhlin**
BCS & IDSS
MIT
rakhlin@mit.edu

## Abstract

It is well known that Empirical Risk Minimization (ERM) may attain minimax suboptimal rates in terms of the mean squared error (Birgé and Massart, 1993). In this paper, we prove that, under relatively mild assumptions, the suboptimality of ERM *must* be due to its bias. Namely, the variance error term of ERM (in terms of the bias and variance decomposition) enjoys the minimax rate. In the fixed design setting, we provide an elementary proof of this result using the probabilistic method. Then, we extend our proof to the random design setting for various models. In addition, we provide a simple proof of Chatterjee's admissibility theorem (Chatterjee, 2014, Theorem 1.4), which states that in the fixed design setting, ERM cannot be ruled out as an optimal method, and then we extend this result to the random design setting. We also show that our estimates imply *stability* of ERM, complementing the main result of Caponnetto and Rakhlin (2006) for non-Donsker classes. Finally, we highlight the somewhat irregular nature of the loss landscape of ERM in the non-Donsker regime, by showing that functions can be close to ERM, in terms of $L_2$ distance, while still being far from almost-minimizers of the empirical loss.

## 1 Introduction

Maximum Likelihood and the method of Least Squares are fundamental procedures in statistics. The study of the asymptotic consistency of Maximum Likelihood has been central to the field for almost a century (Wald, 1949). Along with consistency, failures of Maximum Likelihood have been thoroughly investigated for nearly as long (Neyman and Scott, 1948; Bahadur, 1958; Ferguson, 1982). In the setting of non-parametric estimation, the seminal work of (Birgé and Massart, 1993) provided *sufficient* conditions for minimax optimality (in a non-asymptotic sense) of Least Squares while also presenting an example of a model class where this basic procedure is sub-optimal. Three decades later, we still do not have necessary and sufficient conditions for minimax optimality of Least Squares when the model class is large. While the present paper does not resolve this question, it makes several steps towards understanding the behavior of Least Squares — equivalently, Empirical Risk Minimization (ERM) with square loss — in large models.

Beyond intellectual curiosity, the question of minimax optimality of Least Squares is driven by the desire to understand the current practice of fitting large or overparametrized models, such as neural networks, to data (cf. (Belkin et al., 2019; Bartlett et al., 2020)). At the present moment, there is little theoretical understanding of whether such unregularized data-fitting procedures are optimal, and the study of their statistical properties may lead to new methods with improved performance.

In addition to minimax optimality, many other important properties of Least Squares on large models are yet to be understood. For instance, little is known about its *stability* with respect to perturbations of the data. It is also unclear whether *approximate* minimizers of empirical loss enjoy similar statistical

properties as the exact solution. Conversely, one may ask whether in the landscape of possible solutions, a small perturbation of the minimizer output by Least Squares itself is a near-optimizer of empirical loss.

The contribution of this paper is to provide novel insights into the aforementioned questions for convex classes of functions in a quite generic setting. In detail, we show the following:

1. We prove that in the fixed design setting, the variance term of ERM is upper bounded by the minimax rate of estimation; thus, if the ERM is minimax suboptimal, this must be due to the bias term in the bias-variance decomposition. Then, we extend this result to random design, and provide an upper bound for the variance error term under a uniform boundedness assumption on the class. This bound also implies that under classical assumptions in empirical process theory, the variance error term is minimax optimal.

2. We show that under an isoperimetry assumption on the noise, the expected conditional variance error term of ERM is upper bounded by the "lower isometry" remainder, a parameter that was introduced in Bartlett et al. (2005); Mendelson (2014). Next, under an additional isoperimetry assumption on the covariates, we prove that the variance of ERM is upper bounded by the lower isometry remainder on any robust learning architecture (namely, a class consisting of functions which are all $O(1)$-Lipschitz) which almost interpolates the observations (cf. Bubeck and Sellke (2023)).

3. It is known that ERM is always admissible in the fixed design setting (Chatterjee, 2014; Chen et al., 2017); that is, for any convex function class, there is no estimator having a lower error than ERM (up to a multiplicative absolute constant) on *every* regression function. We provide a short proof of this result for fixed design via a fixed-point theorem. Using the same method, we also prove a somewhat weaker result in the random design case, generalizing the main result of Chatterjee (2014).

4. We show that ERM is stable, in the sense that all almost-minimizers (up to the minimax rate) of the squared loss are close in the space of functions. This result is a non-asymptotic analogue of the asymptotic analysis in Caponnetto and Rakhlin (2006), and extends its scope to non-Donsker classes.

5. While any almost-minimizer of the squared loss is close to the minimizer with respect to the underlying population distribution, the converse is incorrect. We prove that for any non-Donsker class of functions, there exists a target regression function such that, with high probability, there exists a function with high empirical error near the ERM solution; this means that the landscape of near-solutions is, in some sense, irregular.

**Conclusions** Our results show that the ERM enjoys an optimal variance error term in two distinct regimes: the classical regime (van de Geer, 2000), where the function class is fixed and the number of samples is increasing, and the "benign overfitting" setting (Belkin et al., 2019; Bartlett et al., 2020), in which the "capacity" of the class is large compared to the number of samples. In both these settings, our work implies that the minimax optimality of ERM is only determined by its bias error term (or its implicit bias). For models with "few" parameters, *computationally efficient* bias correction methods do exist and are commonly used in practice (cf. (Efron and Tibshirani, 1994)), however, these methods fail over large function classes, in which the bias causes the statistical sub-optimality. Our work reveals the importance of inventing computationally efficient debiasing methods for rich function classes, including non-parametric models and high-dimensional models. The main message of our work is that such methods, if discovered, may significantly improve the statistical performance of ERM over large models in practice.

## 1.1 Prior Work

**Stability of ERM** The stability of learning procedures, which was an active area of research in the early 2000's, has recently seen a resurgence of interest because of its connections to differential privacy and to robustness of learning methods with respect to adversarial perturbations. In the interest of space, we only compare present results to those of Caponnetto and Rakhlin (2006). In the latter paper, the authors showed that the $L_1$-diameter of the set of almost-minimizers of empirical error (with respect to any loss function) asymptotically shrinks to zero as long as the perturbation is $o(n^{-1/2})$ and the function class is Donsker. The analysis there relies on passing from the empirical

process to the associated Gaussian process in the limit, and studying uniqueness of its maximum using anti-concentration properties. While the result there holds without assuming that the class is convex, it is limited by (a) its asymptotic nature and (b) the assumption that the class is not too complex. In contrast, the present paper uses more refined non-asymptotic concentration results, at the expense of additional assumptions such as convexity and minimax optimal lower and upper lower isometry remainders. Crucially, the present result, unlike that of Caponnetto and Rakhlin (2006), holds for non-Donsker classes — those for which the empirical process does not converge to the Gaussian process.

**Shape-constrained regression**   The term "shape-constrained regression" refers to function classes consisting of functions with a certain "shape" property, such as convexity or monotonicity (Samworth and Sen, 2018). In these problems, a common theme is that the statistical behavior of the class exhibits a phase transition when the dimension $d$ of the domain reaches a certain value. For instance, in convex (Lipschitz) regression, the transition happens at $d = 5$: when $d < 5$, the ERM procedure is minimax optimal (Seijo and Sen, 2011; Han and Wellner, 2016; Kim and Samworth, 2016; Seijo and Sen, 2011; Guntuboyina, 2012), but in higher dimensions, it is known to be suboptimal (Kur et al., 2020a,b). In other shape-constrained models, however, the ERM is minimax optimal even in high dimensions, such as *isotonic regression* and *log-concave density estimation* (Han et al., 2019; Kur et al., 2019; Carpenter et al., 2018). Our results show that the sub-optimality of ERM in shape-constrained regression can only be due to the high bias of ERM. These results also align with the empirical observation that for the problem of estimation of convex sets, the ERM has a bias towards "smooth" convex sets (Soh and Chandrasekaran, 2019; Ghosh et al., 2021).

**High-dimensional statistics**   In classical statistics, the Maximum Likelihood (MLE) typically has a low bias compared to its variance, and the standard approach is to introduce bias into the procedure in order to reduce the variance, overall achieving a better trade-off (see (Sur and Candès, 2019, §1) and references therein). In contrast, in high-dimensional models, the MLE may suffer from high bias even in tasks such as logistic regression and sparse linear regression (cf. Candès and Sur (2020); Javanmard and Montanari (2018)). Our results align with this line of work, showing that the problem of high bias may also arise in the task of regression over rich function classes.

## 2   Main Results

In §2.1, we present the setting of our model and the all required preliminaries, and in the remaining sub-sections, we present our results. In details, in §2.2-2.3, we present our results on the variance of ERM in fixed and random designs respectively, and in §3 we provide sketches of some of our proofs. For lack of space, we presents our admissibility and our landscape results on ERM in the supplementary material (§4.1-4.2).

### 2.1   Preliminaries

Let $\mathcal{X}$ be some fixed domain, $\mathcal{F}$ be a class of functions from $\mathcal{X}$ to $\mathbb{R}$, and $f^* \in \mathcal{F}$ an unknown target regression function. We are given $n \geq 2$ data points $X_1 \ldots, X_n \in \mathcal{X}$ and $n$ noisy observations

$$Y_i = f^*(X_i) + \xi_i, \quad i = 1, \ldots, n \tag{1}$$

which we denote by $\mathcal{D} := \{(X_i, Y_i)\}_{i=1}^n$; $\boldsymbol{\xi} := (\xi_1, \ldots, \xi_n)$ is the random noise vector.

In the *fixed design* setting, the observations $X_1 = x_1, \ldots, X_n = x_n$ are arbitrary and fixed, and we denote the uniform measure on this set of points by $\mathbb{P}^{(n)}$. In the *random design* setting, the data points $\mathbf{X} := (X_1, \ldots, X_n)$ are drawn i.i.d. from a probability distribution over $\mathcal{X}$, denoted by $\mathbb{P}$, and the noise vector $\boldsymbol{\xi}$ is drawn independently of $\mathbf{X}$. Note that this model is general enough to cover the high-dimensional setting, as both the function class $\mathcal{F}$ and the distributions of $\boldsymbol{\xi}$ and $X$ are allowed to depend on the number of samples $n$.

An estimator for the regression task is defined as a measurable function $\bar{f}_n : \mathcal{D} \mapsto \{\mathcal{X} \to \mathbb{R}\}$, that for any realization of the input $\mathcal{D}$, outputs some real-valued measurable function on $\mathcal{X}$. The risk of $\bar{f}_n$ is defined as

$$\mathcal{R}(\bar{f}_n, \mathcal{F}, \mathbb{Q}) := \sup_{f^* \in \mathcal{F}} \mathbb{E}_{\mathcal{D}} \int (\bar{f}_n - f^*)^2 d\mathbb{Q}, \tag{2}$$

where $\mathbb{Q} = \mathbb{P}$ in the random design case, and $\mathbb{Q} = \mathbb{P}^{(n)}$ in the fixed design case. Note that in fixed design, the expectation $\mathbb{E}_{\mathcal{D}}$ is taken over the noise $\boldsymbol{\xi}$, while in random design the expectation $\mathbb{E}_{\mathcal{D}}$ is

taken both over the random data points $\mathbf{X}$ and noise $\boldsymbol{\xi}$. The minimax rate is defined via

$$\mathcal{M}(n, \mathcal{F}, \mathbb{Q}) := \inf_{\bar{f}_n} \mathcal{R}(\bar{f}_n, \mathcal{F}, \mathbb{Q}). \tag{3}$$

In the fixed design setting, we also denote the minimax rate by $\mathcal{M}(\mathcal{F}, \mathbb{P}^{(n)})$, as the dependence in $n$ is already present in $\mathbb{P}^{(n)}$.

The most natural estimation procedure is the Least Squares (LS) or Empirical Risk Minimization (ERM) with squared loss, defined as

$$\widehat{f}_n \in \operatorname*{argmin}_{f \in \mathcal{F}} \sum_{i=1}^{n} (f(X_i) - Y_i)^2. \tag{4}$$

When studying fixed design, we will abuse notation and treat $\widehat{f}_n$ as a vector in $\mathbb{R}^n$. We emphasize that many of our results hold for many other estimators, including various *regularized* ERM procedures (see the relevant remarks below).

In both the fixed and random design settings, we shall assume the following:

**Assumption 1.** *$\mathcal{F}$ is a closed convex subset of $L_2(\mathbb{Q})$, where $\mathbb{Q} \in \{\mathbb{P}^{(n)}, \mathbb{P}\}$.*

The convexity of $\mathcal{F}$ means that any $f, g \in \mathcal{F}$ and $\lambda \in [0, 1]$: $\lambda f + (1 - \lambda)g \in \mathcal{F}$; closedness means that for any sequence $\{f_n\}_{n=1}^{\infty} \subset \mathcal{F}$ converging to $f$ with respect to the norm of $L_2(\mathbb{Q})$, the limit $f$ lies in $\mathcal{F}$. The closedness ensures that $\widehat{f}_n$ is well-defined.

Assumption 1 is standard in studying the statistical performance of the ERM (cf. Lee et al. (1996); Bartlett et al. (2005); Mendelson (2014)). In particular, under this assumption, the values of $\widehat{f}_n$ at the observation points $X_1, \ldots, X_n$ is uniquely determined for any $\boldsymbol{\xi}$. Note that in general, the values of a function $f \in \mathcal{F}$ at the points $X_1, \ldots, X_n$ does not uniquely identify $f$ among all functions in the class.

In addition to $\widehat{f}_n$, we analyze properties of the set of $\delta$-approximate minimizers of empirical loss, defined for $\delta > 0$ via

$$\mathcal{O}_\delta := \left\{ f \in \mathcal{F} : \frac{1}{n} \sum_{i=1}^{n} (Y_i - f(X_i))^2 \leq \frac{1}{n} \sum_{i=1}^{n} (Y_i - \widehat{f}_n(X_i))^2 + \delta \right\}. \tag{5}$$

Note that $\mathcal{O}_\delta$ is a random set, in both fixed and random designs.

It is well-known that the squared error of any estimator, in particular that of LS, decomposes into variance and bias components:

$$\mathbb{E}_{\mathcal{D}} \int (\widehat{f}_n - f^*)^2 d\mathbb{Q} = \underbrace{\mathbb{E}_{\mathcal{D}} \int (\widehat{f}_n - \mathbb{E}_{\mathcal{D}} \widehat{f}_n)^2 d\mathbb{Q}}_{V(\widehat{f}_n)} + \underbrace{\int (\mathbb{E}_{\mathcal{D}} \widehat{f}_n - f^*)^2 d\mathbb{Q}}_{B^2(\widehat{f}_n)}, \tag{6}$$

where $\mathbb{Q} = \mathbb{P}^{(n)}$ in the fixed design setting and $\mathbb{Q} = \mathbb{P}$ in the random design setting. Also, for simplicity of the presentation of our results, we denote the maximal variance error term of $\bar{f}_n$ by $\mathcal{V}(\bar{f}_n, \mathcal{F}, \mathbb{Q})$, i.e.

$$\mathcal{V}(\bar{f}_n, \mathcal{F}, \mathbb{Q}) := \sup_{f^* \in \mathcal{F}} V(\bar{f}_n) = \mathbb{E}_{\mathcal{D}} \int (\bar{f}_n - \mathbb{E}_{\mathcal{D}} \bar{f}_n)^2 d\mathbb{Q}$$

In the random design setting, we also have the law of total variance:

$$V(\widehat{f}_n) = \underbrace{\mathbb{E}_{\mathbf{X}} \mathbb{E}_{\boldsymbol{\xi}} \left[ \int \left( \widehat{f}_n - \mathbb{E}_{\boldsymbol{\xi}} \left[ \widehat{f}_n | \mathbf{X} \right] \right)^2 d\mathbb{P} \right]}_{\mathbb{E}V(\widehat{f}_n | \mathbf{X})} + \underbrace{\mathbb{E}_{\mathbf{X}} \left[ \int \left( \mathbb{E}_{\boldsymbol{\xi}} \left[ \widehat{f}_n | \mathbf{X} \right] - \mathbb{E}_{\mathbf{X}, \boldsymbol{\xi}} \widehat{f}_n \right)^2 d\mathbb{P} \right]}_{V(\mathbb{E}(\widehat{f}_n | \mathbf{X}))}. \tag{7}$$

We refer to the two terms as the expected conditional variance and the variance of the conditional expectation, respectively. We conclude this introductory section with a bit of notation and a definition.

**Notation** We use the notation of $\asymp, \gtrsim, \lesssim$ to denote equality/inequality up to an absolute constant. We use $\|\cdot\| = \|\cdot\|_{L_2(\mathbb{P})}$ to denote the $L_2(\mathbb{P})$ norm, and $\|\cdot\|_n$ to denote the $L_2(\mathbb{P}^{(n)})$ norm (that is equal to the Euclidean norm scaled by $1/\sqrt{n}$). Finally, given a function $f : S \to T$ between metric spaces, we define its Lipschitz constant as $\|f\|_{Lip} = \sup_{a,b \in T, a \neq b} \frac{d_T(f(a),f(b))}{d_S(a,b)}$, and we say "$f$ is $L$-Lipschitz" when its Lipschitz constant is at most $L$. Finally, we denote by $\mathrm{diam}_{\mathbb{Q}}(\mathcal{H})$ the $L_2(\mathbb{Q})$ diameter of a set of functions $\mathcal{H}$.

**Definition 1.** *Let $\epsilon \geq 0$, $\mathcal{F} \subseteq \{\mathcal{X} \to \mathbb{R}\}$ and $d(\cdot, \cdot)$ a pseudo-metric on $\mathcal{F}$. We call a set $S \subset \mathcal{F}$ an $\epsilon$-net of $\mathcal{F}$ with respect to $d$ if for any $f \in \mathcal{F}$ there exists $g \in S$ with $d(f, g) \leq \epsilon$. We denote by $\mathcal{N}(\epsilon, \mathcal{F}, d)$ the $\epsilon$-covering number of $\mathcal{F}$ with respect to $d$, that is, the minimal positive integer $N$ such that $\mathcal{F}$ admits an $\epsilon$-net of cardinality $N$.*

### 2.2 Variance of ERM in fixed design setting

In this part, we consider some fixed $(\mathcal{F}, \mathbb{P}^{(n)})$ and assume that the noise is standard normal.

**Assumption 2.** *The noise vector $\boldsymbol{\xi}$ is distributed as an isotropic Gaussian, i.e. $\boldsymbol{\xi} \sim N(0, I_{n \times n})$.*

Our first result provides an exact characterization of the variance (up to a multiplicative absolute constant) under Assumptions 1-2. In order to state it, for a fixed $f^* \in \mathcal{F}$, we define the following set:

$$\mathcal{H}_* := \{f \in \mathcal{F} : \|f - \mathbb{E}_{\mathcal{D}}\widehat{f}_n\|_n^2 \leq 4 \cdot V(\widehat{f}_n)\}. \tag{8}$$

In words, when the underlying function $f^*$ is fixed, we consider the ERM as a random vector (depending on the noise), whose expectation we denote by $\mathbb{E}_{\mathcal{D}}\widehat{f}_n$. $\mathcal{H}_*$ is then just a neighborhood around the expected ERM with a radius of order the square root of the variance error term of $\widehat{f}_n$, when the underlying function is $f^* \in \mathcal{F}$.

We can now state our first result, which uses the notion of the set $\mathcal{O}_\delta$ of $\delta$-approximate minimizers from (5).

**Theorem 1.** *Under Assumptions 1-2, the following holds:*

$$V_{\mathcal{D}}(\widehat{f}_n) \asymp \mathcal{M}(\mathcal{H}_*, \mathbb{P}^{(n)}),$$

*and in particular $\mathcal{V}(\widehat{f}_n, \mathcal{F}, \mathbb{P}^{(n)}) \lesssim \mathcal{M}(\mathcal{F}, \mathbb{P}^{(n)})$. Furthermore, for $\delta := \delta(f^*, n) \lesssim \mathcal{M}(\mathcal{H}_*, \mathbb{P}^{(n)})$, the event*

$$\sup_{f \in \mathcal{O}_\delta} \int (f - \mathbb{E}_{\mathcal{D}}\widehat{f}_n)^2 d\mathbb{P}^{(n)} \asymp \mathcal{M}(\mathcal{H}_*, \mathbb{P}^{(n)}). \tag{9}$$

*holds with probability at least $\max\{1 - 2\exp(-cn \cdot \mathcal{M}(\mathcal{H}_*, \mathbb{P}^{(n)})), 0.9\}$, where $c \in (0, 1)$ is an absolute constant.*

Theorem 1 establishes our first claim: the variance of ERM is bounded above (up to a multiplicative absolute constant) by the minimax rate of estimation on $\mathcal{H}_*$. Since $\mathcal{H}_*$ is contained in $\mathcal{F}$, its minimax rate is at most that of $\mathcal{F}$; in particular, the variance of ERM is bounded by $\mathcal{M}(\mathcal{F}, \mathbb{P}^{(n)})$, and hence, any sub-optimality of ERM must arise from its bias. The theorem also incorporates a stability result: not only is the ERM close to its expected value $\mathbb{E}_{\mathcal{D}}\widehat{f}_n$ with high probability, but any approximate minimizer (up to an excess error of $\delta^2$) is close to $\mathbb{E}_{\mathcal{D}}\widehat{f}_n$ as well.

Our next result complements Theorem 1 above, providing a lower bound on $\mathcal{V}(\widehat{f}_n, \mathcal{F}, \mathbb{P}^{(n)})$:

**Theorem 2.** *Under Assumptions 1-2, the following holds:*

$$\mathcal{V}(\widehat{f}_n, \mathcal{F}, \mathbb{P}^{(n)}) \gtrsim \mathcal{M}(\mathcal{F}, \mathbb{P}^{(n)})^2.$$

Note that there is a multiplicative gap of order $\mathcal{M}(\mathcal{F}, \mathbb{P}^{(n)})$ between the bounds of Theorems 1 and 2. We leave it as an open problem whether the bound of Theorem 2 can be improved under these general assumptions.

## 2.3  Variance of ERM in random design setting

We now turn our attention to the random design setting. Here, we establish similar results to the previous sub-section, albeit under additional assumptions, and with significantly more effort. Unlike the fixed design case, we cannot provide an exact characterization of the variance of ERM. We shall use two different approaches to estimate the variance error term. In the first approach, we use classical tools of empirical process theory together with assumptions that are commonly used in M-estimation (van de Geer, 2000). The second approach, which is inspired by our fixed-design approach, relies heavily on isoperimetry and concentration of measure (cf. Ledoux (2001)).

Throughout this part, we assume for simplicity of presentation that the $L_2(\mathbb{P})$-diameter of the function class is independent of $n$.

**Assumption 3.** *There exist absolute constants $C, c > 0$ such that $c \leq \text{diam}_{\mathbb{P}}(\mathcal{F}) \leq C$.*

The classical work of Yang and Barron (1999) provides a characterization of the minimax rate $(n, \mathcal{F}, \mathbb{P})$ under appropriate assumptions (such as normal noise, and uniform boundedness of $\mathcal{F}$, and richness of $\mathcal{F}$). They proved that the minimax rate is the square of the solution of the following (asymptotic) equation

$$\log \mathcal{N}(\epsilon, \mathcal{F}, \mathbb{P}) \asymp n\epsilon^2, \tag{10}$$

where $\mathcal{N}(\epsilon, \mathcal{F}, \mathbb{P})$ is the $\epsilon$-covering number of $\mathcal{F}$ in terms of $L_2(\mathbb{P})$ metric (see Def. 1 above). We denote this point by $\epsilon_* = \epsilon_*(n)$, and even under less restrictive assumptions, $\epsilon_*^2$ also lower bounds the minimax rate, up to a multiplicative factor of $O(\log n)$. Also, we remark that it is well known that ERM may not achieve this optimal rate (Birgé and Massart, 1993) for large (so-called non-Donsker) function classes.

We also introduce the following additional notations and definitions: First, $\mathbb{P}_n$ denotes the (random) uniform measure over $\mathbf{X} = (X_1, \ldots, X_n)$. Next, following Bartlett et al. (2005), we define the lower and upper isometry remainders of $(\mathcal{F}, \mathbb{P})$ for a given $n$. These remainders measure the discrepancy between $L_2(\mathbb{P})$ and a "typical" $L_2(\mathbb{P}_n)$, here "typical" means for most of realizations of $\mathbf{X}$. These remainders first emerged in the field of metric embeddings, specifically in the definition of quasi-isometries (cf. Ostrovskii (2013)).

In order to introduce these isometry remainders, we first define for each realization of the input $\mathbf{X}$, the constants $\mathcal{I}_L(\mathbf{X})$ and $\mathcal{I}_U(\mathbf{X})$ as the minimal numbers $A_X, B_X \geq 0$, respectively, such that the following holds:

$$\forall f, g \in \mathcal{F}: \ 4^{-1} \int (f-g)^2 d\mathbb{P} - A_X \leq \int (f-g)^2 d\mathbb{P}_n \leq 4 \int (f-g)^2 d\mathbb{P} + B_X.$$

Note that as $A_X$ and $B_X$ increase, the geometry of $L_2(\mathbb{P}_n)$ and $L_2(\mathbb{P})$ over $\mathcal{F}$ becomes less similar. For example, in the extreme case of $A_X = B_X = 0$, it implies the $L_2(\mathbb{P})$ and $L_2(\mathbb{P}_n)$ induce the same topology over $\mathcal{F}$. In words, the lower isometry is the minimal threshold that satisfies the following: all $f, g \in \mathcal{F}$ that are $\omega(\mathcal{I}_L(\mathbf{X}))$ far from each other in $L_2(\mathbb{P})$, must be *at least* $\Omega(\|f-g\|)$ far in $L_2(\mathbb{P}_n)$. The upper isometry remainder implies the converse. To provide further intuition on these remainders, for instance, observe that $\mathcal{I}_L(\mathbf{X})$ upper bounds on diameter in $L_2(\mathbb{P})$ of possible solutions of ERM; namely, one has

$$\sup_{f^* \in \mathcal{F}, \boldsymbol{\xi} \in \mathbb{R}^n} \text{Diam}_{\mathbb{P}}(\{f \in \mathcal{F}: (f(X_1), \ldots, f(X_n)) = (\widehat{f}_n(X_1), \ldots, \widehat{f}_n(X_n))\})^2 \leq 4 \cdot \mathcal{I}_L(\mathbf{X}).$$

Finally, the isometry remainders $\mathcal{I}_L(n), \mathcal{I}_U(n)$ are defined as the "typical" values of $\mathcal{I}_L(\mathbf{X}), \mathcal{I}_U(\mathbf{X})$:

**Definition 2.** *The lower and upper isometry remainders $\mathcal{I}_L(n), \mathcal{I}_U(n)$ are defined as the minimal constants $A_n, B_n \geq 0$ (respectively) such that*

$$\Pr_{\mathbf{X}}(\mathcal{I}_L(\mathbf{X}) \leq A_n) \geq 1 - n^{-1} \text{ and } \Pr_{\mathbf{X}}(\mathcal{I}_U(\mathbf{X}) \leq B_n) \geq 1 - n^{-1}.$$

In the classical regime (van de Geer, 2000), it is considered to be a standard assumption that $\max\{\mathcal{I}_L(n), \mathcal{I}_U(n)\} \lesssim \epsilon_*^2$. However, in the high dimensional setting, it may happen that the lower isometry remainder is significantly smaller than the upper isometry remainder, e.g., $\mathcal{I}_L(n) \lesssim \epsilon_*^2$ and

$\mathcal{I}_U(n) \gg \epsilon_*^2$ (cf. Liang et al. (2020); Mendelson (2014)). We discuss these remainders further in Remark 10 below.

Finally, we remind the reader that $\widehat{f}_n$ is uniquely defined on the data points $\mathbf{X}$ when $\mathcal{F}$ is a convex closed function class, but it *may not* be unique over the entire $\mathcal{X}$ (as multiple functions in $\mathcal{F}$ may take the same values at $X_1, \ldots, X_n$). In §2.3.1, the results hold for any possible solution of $\widehat{f}_n$ over $\mathcal{X}$, whereas in §2.3.2, we (implicitly) assume that $\widehat{f}_n$ is equipped with a selection rule such that it is also unique over the entire $\mathcal{X}$ (e.g., choosing the minimal norm solution (Hastie et al., 2022; Bartlett et al., 2020)); i.e, $\widehat{f}_n : \mathcal{D} \to \mathcal{F}$.

### 2.3.1 The empirical processes approach

Here, we assume that the function class and the noise are uniformly bounded.

**Assumption 4.** *There exist universal constants $\Gamma_1, \Gamma_2 > 0$ such that $\mathcal{F}$ is uniformly upper-bounded by $\Gamma_1$, i.e. $\sup_{f \in \mathcal{F}} \|f\|_\infty \leq \Gamma_1$; and the components of $\boldsymbol{\xi} = (\xi_1, \ldots, \xi_n)$ are i.i.d. zero mean with variance one and are almost surely bounded by $\Gamma_2$.*

*Remark* 1. The uniform boundedness assumption on the noise is taken to simplify the proof (which uses of Talagrand's inequality). This can be relaxed to the assumption that the noise is i.i.d. sub-Gaussian, at the a price of a multiplicative factor of $O(\log n)$ in the error term in Theorem 3 below.

**Definition 3.** *Set $\epsilon_U := \max\{\epsilon_*, \tilde{\epsilon}\}$, where $\tilde{\epsilon}$ is the solution of*

$$\mathcal{I}_U(n) \cdot \log \mathcal{N}(\epsilon, \mathcal{F}, \mathbb{P}) \asymp n\epsilon^4. \tag{11}$$

Note that when $\mathcal{I}_U(n) \lesssim \epsilon_*^2$, $\epsilon_U \asymp \epsilon_*$, while if $\mathcal{I}_U(n, \mathbb{P}) \gg \epsilon_*^2$ then $\epsilon_U \gg \epsilon_*$. The following is our main result in this approach to the random design setting:

**Theorem 3.** *Set $\epsilon_V^2 := \max\{\epsilon_U^2, \mathcal{I}_L(n)\}$, then under Assumptions 1,3,4 the following holds with probability of at least $1 - n^{-1}$:*

$$\sup_{f \in \mathcal{O}_{\delta_n}} \int (f - \mathbb{E}_{\mathcal{D}} \widehat{f}_n)^2 d\mathbb{P} \lesssim \epsilon_V^2,$$

*where $\delta_n = O(\epsilon_V^2)$; and in particular $\mathcal{V}(\widehat{f}_n, \mathcal{F}, \mathbb{P}) \lesssim \epsilon_V^2$.*

Theorem 3 is a generalization of Theorem 1 to the random design case, and its proof uses the strong convexity of the loss and Talagrand's inequality. In §5.1 below, we discuss this bound in the context of "distribution unaware" estimators. We remark that this Theorem extends the scope of Caponnetto and Rakhlin (2006) to non-Donsker classes.

An immediate and useful corollary of this result is that if we have sufficient control of the upper and lower isometry remainders, the variance will be minimax optimal:

**Corollary 1.** *Under Assumptions 1,3,4 and $\max\{\mathcal{I}_L(n), \mathcal{I}_U(n)\} \lesssim \epsilon_*^2$, the following holds:*

$$\mathcal{V}(\widehat{f}_n, \mathcal{F}, \mathbb{P}) \lesssim \epsilon_*^2.$$

In the classical regime, the assumption that $\max\{\mathcal{I}_L(n), \mathcal{I}_U(n)\} \lesssim \epsilon_*^2$ is considered to be standard in the empirical process and shape constraints literature (cf. van de Geer (2000) and references therein); it holds for many classical models (see Remark 11 below).

*Remark* 2. Note that Corollary 1 may also be derived directly from Theorem 1 if the noise is assumed to be standard Gaussian. Yet, this corollary holds for any isotropic sub-Gaussian noise – which is significantly more general.

### 2.3.2 The isoperimetry approach

In order to motivate this part, we point out that just requiring that $\mathcal{I}_L(n) \lesssim \epsilon_*^2$ is considered to be a mild assumption (see Remark 10 below). However, the upper bound of Theorem 3 depends on the upper isometry remainder as well; we would like to find some conditions under which this dependency can be removed. Moreover, note that the isometry remainders are connected to the geometry of

$(\mathcal{F}, \mathbb{P})$ and not directly to the stability properties of the *estimator*. Using a different approach, based on isoperimetry, we will upper-bound the variance of ERM based on some "interpretable" stability parameters of the estimator itself. These stability parameters will be data-dependent relatives of the lower isometry remainder. Differently from the previous part, we do not assume that the function class $\mathcal{F}$ is uniformly bounded by a constant independent of the sample size $n$.

First, we introduce the definition of Lipschitz Concentration Property (LCP):

**Definition 4.** *Let* $\mathbf{Z} = (Z_1, \ldots, Z_m)$ *be a random vector taking values in* $\mathcal{Z}^{\otimes m}$. $\mathbf{Z}$ *satisfies the LCP with constant* $c_L > 0$, *with respect to a metric* $d : (\mathcal{Z}^{\otimes m}, \mathcal{Z}^{\otimes m}) \to \mathbb{R}^+$, *if for all* $F : \mathcal{Z}^m \to \mathbb{R}$ *is* 1-*Lipschitz, the following holds:*

$$\Pr(|F(\mathbf{Z}) - \mathbb{E}F(\mathbf{Z})| \geq t) \leq 2 \exp(-c_L t^2). \tag{12}$$

The LCP property is also known as the isoperimetry condition (cf. (Bubeck and Sellke, 2023, §1.3)), and it is stronger than being sub-Gaussian (Boucheron et al., 2013), and yet it is significantly less restrictive than requiring normal noise (in which case $c_L = 1/2$ (Ledoux, 2001)); for further details see Remark 9 below. Now, we state our first assumption:

**Assumption 5.** $\boldsymbol{\xi}$ *is an isotropic random vector satisfying* (12) *with constant* $c_L = \Theta(1)$, *with respect to the Euclidean norm in* $\mathbb{R}^n$.

Recall that $\epsilon_*$ is defined as the stationary point of $n\epsilon^2 \asymp \log \mathcal{N}(\epsilon, \mathcal{F}, \mathbb{P})$, and that the conditional variance of $\widehat{f}_n$, which is a function of the realization $\mathbf{X}$ of the input, is defined as

$$V(\widehat{f}_n | \mathbf{X}) := \mathbb{E}_{\boldsymbol{\xi}}[\|\widehat{f}_n - \mathbb{E}_{\boldsymbol{\xi}}[\widehat{f}_n | \mathbf{X}]\|^2];$$

that is, we fix the data points $\mathbf{X}$ and take expectation over the noise.

The formulation of the following definition involves a yet-to-be-defined (large) absolute constant $M > 0$, which will be specified in the proof of Theorem 4 (see §3.2 below).

**Definition 5.** *For each realization* $\mathbf{X}$ *and* $f^* \in \mathcal{F}$, *let* $\rho_S(\mathbf{X}, f^*)$ *be defined as the minimal constant* $\delta(n)$ *such that*

$$\Pr_{\boldsymbol{\xi}} \left\{ \boldsymbol{\xi} \in \mathbb{R}^n : \forall \boldsymbol{\xi}' \in B_n(\boldsymbol{\xi}, M\epsilon_*) : \|\widehat{f}_n(\mathbf{X}, \boldsymbol{\xi}') - \widehat{f}_n(\mathbf{X}, \boldsymbol{\xi})\|^2 \leq \delta(n) \right\} \geq \exp(-c_2 n \epsilon_*^2). \tag{13}$$

*where* $B_n(\boldsymbol{\xi}, r) = \{\boldsymbol{\xi}' \in \mathbb{R}^n : \|\boldsymbol{\xi} - \boldsymbol{\xi}'\|_n \leq r\}$, *and* $c_2 > 0$ *is an absolute constant.*

We also set $\rho_S(\mathbf{X}) := \sup_{f^* \in \mathcal{F}} \rho_S(\mathbf{X}, f^*)$. $\rho_S(\mathbf{X})$ measures the optimal radius of stability (or "robustness") of $\widehat{f}_n$ to perturbations of the noise *when the underlying function and data points* $\mathbf{X}$ *are fixed*. This is a weaker notion than the lower isometry remainder; in fact, one can verify that $\rho_S(\mathbf{X}) \lesssim \max\{\mathcal{I}_L(\mathbf{X}), \epsilon_*^2\}$ for every realization $\mathbf{X}$ (see Lemma 12 for completeness). Now, we are ready to present our first theorem:

**Theorem 4.** *Under Assumptions 1,3,5, the following holds for every realization* $\mathbf{X}$ *of the data:*

$$V_{\mathcal{D}}(\widehat{f}_n | \mathbf{X}) \lesssim \max\{\rho_S(\mathbf{X}, f^*), \epsilon_*^2\},$$

*and in particular* $\sup_{f^* \in \mathcal{F}} \mathbb{E}_{\mathbf{X}} V_{\mathcal{D}}(\widehat{f}_n | \mathbf{X}) \lesssim \max\{\mathcal{I}_L(n), \epsilon_*^2\}$.

Note that if $\mathcal{I}_L(n) \lesssim \epsilon_*^2$ – a very mild assumption – then we obtain that the expected conditional variance is minimax optimal. However, we believe that it is impossible to bound the total variance via the lower isometry remainder alone. Intuitively, $\widehat{f}_n$ only observes a given realization $\mathbf{X}$, and in general, the geometry of $\mathcal{F}$ may "look different" under different realizations if $\mathcal{I}_L(n)$ is large (see the discussion in §5.1 for further details).

In our next result, we identify a model under which we can bound the *total* variance of $\widehat{f}_n$ by the lower isometry remainder. To state the next assumption, we fix a metric $d : \mathcal{X} \times \mathcal{X} \to \mathbb{R}^+$ on $\mathcal{X}$, and denote by $d_n$ the metric on $\mathcal{X}^n$ given by $d_n(\mathbf{X}, \mathbf{X}')^2 = \sum d(X_i, X_i')^2$.

**Assumption 6.** $\mathbf{X} \sim \mathbb{P}^{\otimes n}$ *satisfies* (12) *with respect to the metric* $d_n(\cdot, \cdot)$, *and with constant* $c_X$ *that only depends on* $\mathbb{P}$

Note that it is insufficient to assume that $X \sim \mathbb{P}$ satisfies an LCP, since this does not imply that $\mathbf{X}$ satisfies an LCP with a constant independent of $n$ (w.r.t. to $d_n(\cdot, \cdot)$). However, if $X \sim \mathbb{P}$ satisfies a concentration inequality which tensorizes "nicely," such as a log-Sobolev or $W_2$-transportation cost inequality (cf. (Ledoux, 2001, §5.2, §6.2)), then $\mathbf{X} \sim \mathbb{P}^{\otimes n}$ does satisfy this LCP property.

Next, we assume that with high probability, $\widehat{f}_n$ is close to interpolating the observations:

**Assumption 7.** *There exist absolute constants $c_I, C_I > 0$, such that the following holds:*

$$\Pr_{\mathcal{D}} \left( \sum_{i=1}^n (\widehat{f}_n(X_i) - Y_i)^2 \leq C_I \epsilon_*^2 \cdot n \right) \geq 1 - \exp(-c_I n \epsilon_*^2).$$

This assumption is quite common in the study of "rich" high dimensional models (i.e. when the function class $\mathcal{F}$ depends on $n$; see, e.g., Belkin et al. (2019); Liang and Rakhlin (2020)) which are prominent in the recent benign overfitting literature.

Finally, we introduce another stability notion. Recall the random set $\mathcal{O}_\delta \subset \mathcal{F}$ of almost-minimizers of the empirical loss, as defined in (5) above; note that, in the random design setting, $\mathcal{O}_\delta$ depends on both $\mathbf{X}$ and $\boldsymbol{\xi}$. The random variable $\operatorname{diam}_{\mathbb{P}}(\mathcal{O}_\delta)$ can be thought of as measuring the stability of the ERM with respect to imprecision in the minimization algorithm (cf. (Caponnetto and Rakhlin, 2006)). The formulation of the following definition involves another yet-to-be-defined (large) absolute constant $M' > 0$, which will be specified in the proof of Theorem 5 (see §6.6 below), as well as the constant $c_I$ from Assumption 7.

**Definition 6.** *$\rho_{\mathcal{O}}(n, \mathbb{P}, f^*)$ is defined as the smallest $\delta(n) \geq 0$ such that*

$$\Pr_{\mathcal{D}}(\operatorname{diam}_{\mathbb{P}}(\mathcal{O}_{M'\epsilon_*^2}) \leq \sqrt{\delta(n)}) \geq 2 \exp(-c_I n \epsilon_*^2), \tag{14}$$

*where $c_I \geq 0$ is the same absolute constant defined in Assumption 7.*

In order to understand the relation between this and the previous stability notions, note that under Assumption 7 and the event $\mathcal{E}$ of Definition 6, we have that on an event of nonnegligible probability, $\rho_S(\mathbf{X}, f^*) \leq \rho_{\mathcal{O}}(n, \mathbb{P}, f^*)$; in addition, $\rho_{\mathcal{O}}(n, \mathbb{P}, f^*) \lesssim \max\{\mathcal{I}_L(n), \epsilon_*^2\}$ (see Lemma 13 below). Under these additional two assumptions and the last definition, we state our bound for the total variance of $\widehat{f}_n$:

**Theorem 5.** *Under Assumptions 1,3,5-7, the following holds:*

$$V_{\mathcal{D}}(\widehat{f}_n) \lesssim c_X^{-1} \cdot \sup_{f^* \in \mathcal{F}} \|f^*\|_{Lip} \cdot \max\{\epsilon_*^2, \rho_{\mathcal{O}}(n, \mathbb{P}, f^*)\},$$

*and in particular one has $\mathcal{V}(\widehat{f}_n, \mathcal{F}, \mathbb{P}) \lesssim c_X^{-1} \cdot \sup_{f^* \in \mathcal{F}} \|f^*\|_{Lip} \cdot \max\{\epsilon_*^2, \mathcal{I}_L(n)\}$.*

Note that when $\mathcal{F}$ is a robust learning architecture (i.e. $\mathcal{F} \subset \{\mathcal{X} \to \mathbb{R} : \|f\|_{Lip} = O(1)\}$), our bound is optimal. Interestingly, the assumptions of Theorem 5 coincide with those of the model considered in the recent paper of Bubeck and Sellke (2023). Also note that the last theorem connects the total variance of $\widehat{f}_n$ to a "probabilistic" threshold for the $L_2(\mathbb{P})$-diameter of the *data-dependent* set of $\Theta(\epsilon_*^2)$−approximating solutions of $\widehat{f}_n$ – two parameters which at first sight are unrelated.

*Remark* 3. One may suspect that the assumptions of almost interpolation and robustness are incompatible, which would render our theorem vacuous. However, perhaps counter-intuitively, in the high-dimensional setting these assumptions can coexist. For example, interpolation with $O(1)$-Lipschitz functions may be possible when the "intrinsic" dimension of $\mathcal{X}$ is $\Omega(\log(n))$ (depending on the richness of $\mathcal{F}$), though it is generally impossible when the dimension is $o(\log(n))$ (this follows from the behaviour of the entropy numbers of the class of Lipschitz functions; cf. Dudley (1999)).

*Remark* 4. Using Assumptions 1,3,5-6, one may prove the same bound as in Theorem 3, i.e. that $\mathcal{V}(\widehat{f}_n, \mathcal{F}, \mathbb{P}) \lesssim \epsilon_V^2$, without requiring the noise or the function class to be uniformly bounded. The idea is to obtain the crucial concentration bounds in the proof of Theorem 3 by using the LCP properties of $\boldsymbol{\xi}$ and $X \sim \mathbb{P}$ along with the robustness of $\mathcal{F}$, rather than via Talagrand's inequality.

# 3 Proof sketches

In this section, we sketch the proofs of less-technical results to give the reader a flavor of our methods. The full proofs are given in the next section. For the proofs we introduce some additional notations. For $m \in \mathbb{N}$, we set $[m] := \{1, \ldots, m\}$. The inner products in $L_2(\mathbb{P}), L_2(\mathbb{P}^{(n)})$ are denoted by $\langle \cdot, \cdot \rangle, \langle \cdot, \cdot \rangle_n$, respectively.

## 3.1 Sketch of proof of Theorem 1

Here, we sketch a simple proof of a weaker version of our result, namely $V_{\mathcal{D}}(\hat{f}_n) \lesssim \mathcal{M}(\mathcal{F}, \mathbb{P}^{(n)})$, under the stronger assumption that

$$\mathcal{M}(\mathcal{F}, \mathbb{P}^{(n)}) \asymp \epsilon_*^2, \tag{15}$$

where $\epsilon_*$ solves $\log \mathcal{N}(\epsilon, \mathcal{F}, \mathbb{P}^{(n)}) \asymp n\epsilon^2$. (This holds under reasonable assumptions on $\mathcal{F}$, but can be dispensed with; see Lemma 1 for the exact characterization.) In §6.1.1, we fill in the details of this sketch, and in §6.1.2, we give the full proof of Theorem 1.

The proof uses the probabilistic method (Alon and Spencer, 2016). Let $f_1, \ldots, f_N$ be centers of a minimal $\epsilon_*$-cover of $\mathcal{F}$ with respect to $L_2(\mathbb{P}^{(n)})$, per Definition 1. First, since for any $i \in [N]$, the map $\boldsymbol{\xi} \mapsto \sqrt{n} \|\hat{f}_n(\boldsymbol{\xi}) - f_i\|_n$ is 1-Lipschitz, (12) and a union bound ensure that with probability at least $1 - \frac{1}{2N}$, for all $i \in [N]$,

$$\mathbb{E}_{\boldsymbol{\xi}} \|\hat{f}_n - f_i\|_n - \|\hat{f}_n - f_i\|_n \lesssim \sqrt{\frac{\log N}{n}}.$$

On the other hand, by the pigeonhole principle, there exists at least one $i^* \in [N]$ such that with probability at least $1/N$, $\|\hat{f}_n - f_{i^*}\|_n \leq \epsilon_*$. Hence, there exists at least one realization of $\boldsymbol{\xi} \in \mathbb{R}^n$ for which both bounds hold, and thus, *deterministically*,

$$\mathbb{E}_{\boldsymbol{\xi}} \|\hat{f}_n - f_{i^*}\|_n \lesssim \epsilon_* + \sqrt{\frac{\log N}{n}} \lesssim \epsilon_*$$

where we used the balancing equation (10). Another application of (12) and integration of tails yields

$$V(\hat{f}_n) = \mathbb{E}_{\mathcal{D}} \|\hat{f}_n - \mathbb{E}_{\mathcal{D}} \hat{f}_n\|_n^2 \leq \mathbb{E}_{\mathcal{D}} \|\hat{f}_n - f_{i^*}\|_n^2 \lesssim \epsilon_*^2,$$

implying that the variance of ERM is minimax optimal.

## 3.2 Sketch of proof of Theorem 4

As is well-known, the Lipschitz concentration condition (12) is equivalent to an isoperimetric phenomenon: for any set $A \subset \mathbb{R}^n$ with $\Pr_{\boldsymbol{\xi}}(A) \geq 1/2$, its $t$-neighborhood $A_t = \{\boldsymbol{\xi} \in \mathbb{R}^n : \inf_{x \in A} \|x - \boldsymbol{\xi}\|_n \leq t\}$ satisfies

$$\Pr_{\boldsymbol{\xi}}(A_t) \geq 1 - 2\exp(-nt^2/2). \tag{16}$$

One sees quickly that this implies that if $A$ has measure at least $2\exp(-nt^2/2)$, then $A_{2t}$ has measure $1 - 2\exp(-nt^2/2)$.

Let $\mathcal{E}$ be the event of Definition 5. As in §3.1 above, one obtains via the pigeonhole principle and the definition of $\epsilon_*$ that there exists some $f_c \in \mathcal{F}$ such that

$$\Pr_{\boldsymbol{\xi}}(\underbrace{\{\hat{f}_n \in B(f_c, \epsilon_*)\} \cap \mathcal{E}}_{A} | \mathbf{X}) \geq \frac{\Pr_{\boldsymbol{\xi}}(\mathcal{E})}{\mathcal{N}(\epsilon_*, \mathcal{F}, \mathbb{P})} \geq \exp(-C_3 n\epsilon_*^2).$$

By isoperimetry, $\Pr_{\boldsymbol{\xi}}(A_{2t}) \geq 1 - 2\exp(-nt^2/2)$, where $t = M\epsilon_*/2$ and $M$ is chosen such that $(M/2)^2 \geq 2C_3$; this fixes the value of the absolute constant $M$ used in (13).

Applying (13) yields that if $\boldsymbol{\xi} \in A \subset \mathcal{E}$ and $\|\boldsymbol{\xi}' - \boldsymbol{\xi}\|_n \leq M\epsilon_* = 2t$, $\|\hat{f}_n(\boldsymbol{\xi}) - \hat{f}_n(\boldsymbol{\xi}')\| \leq \rho_S(\mathbf{X}, f^*)$ and so $\|\hat{f}_n(\boldsymbol{\xi}') - f_c\| \leq \epsilon_* + \rho_S(\mathbf{X}, f^*)$. This implies

$$\Pr_{\boldsymbol{\xi}}(\{\hat{f}_n \in B(f_c, \epsilon_* + \rho_S(\mathbf{X}, f^*))\} | \mathbf{X}) \geq \Pr_{\boldsymbol{\xi}}(A_{2t} | \mathbf{X}) \geq 1 - 2\exp(-nt^2/2),$$

which implies via conditional expectation that $V(\hat{f}_n | \mathbf{X}) \lesssim \max\{\rho_S(\mathbf{X}), \epsilon_*^2\}$, as desired (where we used that $\epsilon_*^2 \gtrsim \log(n)/n$ (see Lemma 3 below), and therefore $\exp(-nt^2) = O(\epsilon_*^2)$).

**Acknowledgements:** This work was supported by the Simons Foundation through Award 814639 for the Collaboration on the Theoretical Foundations of Deep Learning, the ERC under the European Union's Horizon 2020 research and innovation programme (grant agreement No 770127), and the NSF (awards DMS-2031883, DMS-1953181). Part of this work was carried out while the first two authors authors were in residence at the Institute for Computational and Experimental Research in Mathematics in Providence, RI, during the Harmonic Analysis and Convexity program; this residency was supported by the NSF (grant DMS-1929284). Finally, the first two authors also wish to acknowledge Prof. Shiri Artstein-Avidan for introducing them to each other.

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
