# 4 All remaining results

## 4.1 On the admissibility of ERM

### 4.1.1 Fixed design setting

We recall the notion of *admissibility* of $\widehat{f}_n$, first established by Chatterjee (2014), with a simplified proof given by Chen et al. (2017). The result states that for any estimator $\bar{f}_n$, there *exists* a target function $f^* \in \mathcal{F}$ such that ERM over the data drawn according to (1) has error which is no worse (up to an absolute constant) than that of $\bar{f}_n$. Hence, while ERM may be suboptimal for some models $f^* \in \mathcal{F}$, it cannot be ruled out completely as a learning procedure.

It is fairly straightforward to see that Theorem 1 implies $\widehat{f}_n$ is admissible if at least one $f^* \in \mathcal{F}$ exists with a "small bias". Our next result offers an alternative proof for Chatterjee's admissibility theorem. In this proof, under Assumptions 1 and 2, we demonstrate that there always exists an $f^* \in \mathcal{F}$ such that $\widehat{f}_n$ exhibits a small bias.

**Corollary 2** (Chatterjee's Admissibility Theorem). *Let $\bar{f}_n : \mathcal{D} \to \mathbb{R}^n$ be some estimator. Then, under Assumptions 1,2, there exists an underlying function $f^* \in \mathcal{F}$ (that depends on $\bar{f}_n$) such that*

$$\mathbb{E}_{\mathcal{D}} \int (\widehat{f}_n - f^*)^2 d\mathbb{P}^{(n)} \leq C \cdot \mathbb{E}_{\mathcal{D}} \int (\bar{f}_n - f^*)^2 d\mathbb{P}^{(n)}, \tag{17}$$

*where $C > 0$ is a universal constant that is independent of $\bar{f}_n$, $\mathcal{F}$, $\mathbb{P}^{(n)}$.*

*Remark* 5. Under Assumptions 1, 2, and the additional assumption of $\mathcal{F}$ being a *compact* class (in terms of $L_2(\mathbb{P}^{(n)})$), our proof demonstrates that the admissibility property is valid for any $\widehat{g}_n$ such that $\boldsymbol{\xi} \mapsto \widehat{g}_n(\boldsymbol{\xi})$ is $O(1)$-Lipschitz. Also, it offers a new and simplified perspective on this profound theorem. Specifically, we show that admissibility hinges on the existence of a target regression function $f^* \in \mathcal{F}$ such that the estimator not only has a "small bias" but is also "stable" around it. Remarkably, the existence of such a target function is ensured by a purely topological argument—the Brouwer's fixed-point theorem. From a statistical perspective, this has a simple interpretation: a "stable" estimator cannot have a "large" bias on every target function within a compact function class.

Note that if we place $\bar{f}_n$ in Corollary 2 to be a minimax optimal estimator immediately yields the following:

**Corollary 3** (Weak admissibility (fixed design)). *Under Assumptions 1,2, there exists an underlying function $f^* \in \mathcal{F}$ such that*

$$\mathbb{E}_{\mathcal{D}} \int (\widehat{f}_n - f^*)^2 d\mathbb{P}^{(n)} \leq C_1 \cdot \mathcal{M}(\mathcal{F}, \mathbb{P}^{(n)}). \tag{18}$$

*where $C_1 > 0$ is a universal constant that is independent of $\mathcal{F}$, $\mathbb{P}^{(n)}$.*

We say that the ERM is *weakly admissible* if there exists $f^* \in \mathcal{F}$ such that the error of the ERM on such $f^*$ is minimax optimal up to an absolute constant. [1] In these terms, Corollary 3 simply states that in fixed design and under Assumptions 1,2, the ERM is weakly admissible.

### 4.1.2 Random Design Setting

In this section we state our new weak admissibility result for the ERM, which is the analogue of Corollary 3 in the random design setting. We require the following additional technical assumption:

**Assumption 8.** *The function class $\mathcal{F}$ is compact with respect to $L_2(\mathbb{P})$, and that or every $x \in \mathcal{X}$, the evaluation functional $f \mapsto f(x)$ is continuous in the $L_2(\mathbb{P})$ norm when restricted to $\mathcal{F}$.*

As we assumed that $\mathcal{F}$ is closed in Assumption 1, it suffices that $\mathcal{F}$ have finite $\epsilon$-entropy for every $\epsilon$ to ensure that $\mathcal{F}$ is compact. We will use the regularity condition in order to apply a fixed-point theorem for continuous functions on a compact convex set in a Banach space.

---

[1] In the paper of Kur and Rakhlin (2021), a sharp lower bound on the minimal error of ERM in the fixed design setting is proven.

**Theorem 6** (Weak admissibility (random design)). *Under Assumptions 1,8, there exists a target function $f^* \in \mathcal{F}$ (depending only on $n$) such that*

$$\mathbb{E}_{\mathcal{D}} \int (\widehat{f}_n - f^*)^2 d\mathbb{P} \lesssim \max\{\mathcal{V}(\widehat{f}_n, \mathcal{F}, \mathbb{P}), \mathcal{I}_L(n)\}.$$

In particular, Theorem 6 implies that when $\max\{\mathcal{V}(\widehat{f}_n, \mathcal{F}, \mathbb{P}), \mathcal{I}_L(n)\} \lesssim \mathcal{M}(n, \mathcal{F}, \mathbb{P})$, the ERM is weakly admissible.

On the other hand, we conjecture that when $\mathcal{I}_L(n) \gg \mathcal{M}(n, \mathcal{F}, \mathbb{P})$, the ERM is not weakly admissible. This would follow from the stronger conjecture is that the bound of Theorem 6 is optimal.

### 4.2 On The Landscape of ERM in the non-Donsker regime

Finally, we establish a counter-intuitive behavior of the landscape around $\widehat{f}_n$ for various non-parametric models that lie in the non-Donsker regime.

For our purposes, the "non-Donsker regime" simply means that the model satisfies Assumption 9 below . The conditions in Assumption 9 may seem a bit technical at first glance, but they cover many well-studied non-parametric models that appear in the shape-constraints literature including convex regression and $\alpha$-Hölder regression in the suitable dimensions.

**Assumption 9.** *The model $(\mathcal{F}, \mathbb{P})$ satisfies the following:*

1. *$\mathcal{F}$ is uniformly bounded by an absolute constant $\Gamma > 0$.*

2. *The function $\epsilon \mapsto \frac{\epsilon^2}{\log(\epsilon^{-1})} \cdot \log \mathcal{N}(\epsilon, \mathcal{F}, \mathbb{P})$ is decreasing in $\epsilon \in (0, \Gamma)$.*

3. *The lower and upper isometry remainders satisfy: $\mathcal{I}_L(n) = o(\epsilon_*^2)$ and $\mathcal{I}_U(n) = O(\epsilon_*^2)$ .*

Now, we are ready to state our result:

**Theorem 7.** *Let $(\mathcal{F}, \mathbb{P}, \boldsymbol{\xi})$ that satisfies Assumptions 1,2, 9 and set $\epsilon_* = \epsilon(n)$ as in (10) above. Then, there exists a sequence $C_{\mathcal{F}, \mathbb{P}}(n) = \omega(1)$ and a sequence of functions $f^* = f^*(n) \in \mathcal{F}$, such that $\mathbb{E}_{\mathcal{D}}\|\widehat{f}_n - f^*\|^2 \lesssim \epsilon_*^2$ (i.e., each $f^*$ is weakly admissible) and*

$$\left\{ f \in \mathcal{F} : \int (f - \widehat{f}_n)^2 d\mathbb{P} \lesssim \epsilon_*^2 \right\} \not\subset \mathcal{O}_{C_{\mathcal{F}, \mathbb{P}}(n) \cdot \epsilon_*^2}, \tag{19}$$

*with probability of at least $1 - n^{-1}$.*

Theorem 7 says that for some target function, the LS solution $\widehat{f}_n$ displays counterintuitive behavior: on the one hand, $\widehat{f}_n$ estimates $f^*$ optimally, but on the other hand, for most $\boldsymbol{\xi}$ there exist functions which are very close to $\widehat{f}_n$ in $L_2(\mathbb{P})$, and yet far from being minimizers of the squared error.

## 5 Discussion and Additional Remarks

### 5.1 On the optimality of the variance error term in random design setting

When $\max\{\mathcal{I}_L(n), \mathcal{I}_U(n)\} \gg \epsilon_*^2$, we have that $\epsilon_V^2 \gg \epsilon_*^2$ – our upper bound is larger than the minimax error. Therefore, this bound seems at first glance to be suboptimal.

However, to the best of our knowledge, all estimators that attain the minimax rate, such as aggregation and related algorithms (cf. Yang (2004)), depend on the marginal distribution $\mathbb{P}$ of the covariates. In many cases, though, we do not know or have oracle access to the marginal distribution $\mathbb{P}$, and the estimator only has access to $\mathbb{P}_n$ and to $\mathcal{F}$. It is natural to ask what is the minimax rate of "distribution-unaware" estimators that only depend on $\mathbb{P}_n$ and the function class $\mathcal{F}$, when the underlying distribution $\mathbb{P}$ is allowed to vary over some family of distributions.

To this end, given some family of probability distributions $\mathcal{P}$ on a domain $\mathcal{X}$, consider the following measurement of optimality of an estimator:

$$\Delta_{(du)}(n, \mathcal{F}, \mathcal{P}) = \inf_{\bar{f}_n} \sup_{\mathbb{Q} \in \mathcal{P}} \frac{\mathcal{R}(\bar{f}_n, \mathcal{F}, \mathbb{Q})}{\mathcal{M}(n, \mathcal{F}, \mathbb{Q})}.$$

We say that there exists an *optimal* distribution unaware estimator over $(n, \mathcal{F}, \mathcal{P})$ when $\Delta_{(du)}(n, \mathcal{F}, \mathcal{P}) = \Theta(1)$.

Unsurprisingly, if we do not place additional assumptions on $\mathcal{F}$ and $\mathcal{P}$ (beyond convexity), then it may happen that $\Delta_{(du)}(n, \mathcal{F}, \mathcal{P}) = \omega(1)$ – i.e. no single estimator attains the minimax error on every distribution $\mathbb{Q} \in \mathcal{P}$. In other words, a minimax optimal estimator for $\mathbb{Q}$ must "know" $\mathbb{Q}$. In fact, one may construct a set of probability distributions $\mathcal{P}$ on a domain $\mathcal{X}$ and a function class $\mathcal{F}$ such that for any $\mathbb{Q} \in \mathcal{P}$, $\mathcal{M}(n, \mathcal{F}, \mathbb{Q}) = O(n^{-1})$ (the parametric rate), and for any estimator $\bar{f}_n$, one may find $\mathbb{Q} \in \mathcal{P}$ such that $\mathcal{R}(\bar{f}_n, \mathcal{F}, \mathbb{Q}) = \Theta(1)$; and in particular $\Delta_{(du)}(n, \mathcal{F}, \mathcal{P}) = \Theta(n)$ (see Example §1 below).

It's also intuitively clear that the version space diameter, namely,

$$\Psi(n, \mathbb{P}) := \sup_{f^* \in \mathcal{F}} \mathbb{E}[\mathrm{diam}_{\mathbb{P}}(\{f \in \mathcal{F} : \forall i \in 1, \ldots, n \quad f^*(X_i) = f(X_i)\})]$$

should appear in the error of any "distribution-unaware" estimator in terms of $L_2(\mathbb{P})$ (though we do not know how to show this in complete generality). Clearly, for every model, $\Psi(n, \mathbb{P}) \leq \mathcal{I}_L(n)$. Therefore, it is not surprising that the bound of Theorem 3 includes the lower isometry remainder. The upper isometry remainder $\mathcal{I}_U(n)$, though, is not tightly connected to $\Psi(n, \mathbb{P})$. Nonetheless, we conjecture that it cannot be removed from the bound of Theorem 3. Specifically, we propose the following conjecture:

**Conjecture 1.** *For every $n \geq 1$ there exists models $(\mathcal{F}(n), \mathbb{P})$ in which $\widehat{f}_n$ incurs a variance error term of order $\epsilon_U^2 \gg \epsilon_*^2 \gg \mathcal{I}_L(n)$ or $\mathcal{I}_L(n) \gg \epsilon_U^2 \gg \epsilon_*^2$.*

This conjecture implies that the bound of Theorem 3 cannot be improved under Assumptions.

The intuition behind this conjecture is as follows: the ERM sees the geometry of $\mathcal{F}$ with respect to $\mathbb{P}_n$ rather than $\mathbb{P}$, and perturbing the data points $X_1, \ldots, X_n$ by $\delta_1, \ldots, \delta_n$ can change the finite-dimensional geometry of the projected function class, reducing the "stability" of $\widehat{f}_n$. This is because $\widehat{f}_n$ is not a Lipschitz function in the data $\mathbf{X}$ with respect to the $L_2(\mathbb{P})$-norm, in contrast to its Lipschitz properties in $\boldsymbol{\xi}$ with respect to the $L_2(\mathbb{P}^{(n)})$-norm. Only if the upper and lower isometry constants are small, say of the order $\epsilon_*^2$ does the metric geometry of $\mathcal{F}_n$ not depend too much on $\mathbf{X}$, up to $\epsilon_*^2$, in which case we expect the variance to be bounded by $\epsilon_*^2$.

Our confidence that this is the correct explanation for the appearance of $\mathcal{I}_U(n)$, rather than some other phenomenon, derives from Theorem 4, which precisely states that the expected conditional variance of $\widehat{f}_n$ is upper bounded by the lower isometry radius, i.e.

$$\mathbb{E}_{\mathbf{X}} V(\widehat{f}_n | \mathbf{X}) \lesssim \mathcal{I}_L(n). \tag{20}$$

Therefore, if Conjecture 1 is correct and there are models in which $\mathcal{V}(\widehat{f}_n, \mathcal{F}, \mathbb{P}) \gtrsim \epsilon_U^2 \gg \mathcal{I}_L(n)$, this must be due to the variance of conditional expectations:

$$V\left(\mathbb{E}_{\boldsymbol{\xi}}\left[\widehat{f}_n | \mathbf{X}\right]\right) \gtrsim \epsilon_U^2, \tag{21}$$

and $V\left(\mathbb{E}_{\boldsymbol{\xi}}\left[\widehat{f}_n | \mathbf{X}\right]\right)$ is precisely the error term which captures how the geometry of $\mathcal{F}$ varies under different realizations.

## 5.2 Additional Remarks

### 5.2.1 Remarks on §2.2

*Remark* 6. The first part of of Theorem 1 holds for any estimator $\widehat{g}_n$ for which the map $\boldsymbol{\xi} \mapsto \widehat{g}_n(\boldsymbol{\xi})$ is $O(1)$-Lipschitz, namely,

$$V_{\mathcal{D}}(\widehat{g}_n) \asymp \mathcal{M}(\mathcal{H}_*, \mathbb{P}^{(n)}),$$

where $\mathcal{H}_*$ is defined in (8) above. Furthermore, when $\boldsymbol{\xi}$ satisfies Assumption 5, then our proof implies that

$$V_{\mathcal{D}}(\widehat{f}_n) \lesssim \mathcal{M}(\mathcal{H}_*, \mathbb{P}^{(n)}),$$

here $\mathcal{M}(\mathcal{H}_*, \mathbb{P}^{(n)})$ is the *minimax rate under isotropic Gaussian noise.*

*Remark* 7. One can verify that for $\delta(f^*, n) = \omega(\mathcal{M}(\mathcal{F}, \mathcal{H}_*))$, Theorem 1 cannot be true in its full generality. Therefore, the stability threshold of $\delta(f^*, n)$ is tight, up to a multiplicative absolute constant.

*Remark* 8. The proof of Theorem 2 is specific to $\widehat{f}_n$ and cannot be extended immediately to other estimators. However, it holds for any isotropic noise distribution.

### 5.2.2 Remarks on §2.3

*Remark* 9. Herbst's argument (Wainwright, 2019, §3.1.2) implies that the Lipschitz concentration property holds for any random vector $\boldsymbol{\xi}$ satisfying a log-Sobolev inequality; the converse is not true in general. However, in the seminal work of (Milman, 2009), it was shown that if $\boldsymbol{\xi}$ is assumed to be log-concave, then $\boldsymbol{\xi}$ which satisfies a LCP with constant $c_L$ also satisfies a log-Sobolev inequality with constant $\Theta(c_L)$.

*Remark* 10. In the seminal works of Mendelson (cf. the recent paper Mendelson (2017) and references within) the the small ball condition was introduced to estimate the statistical performance of ERM under less restrictive assumptions as uniform boundedness, Koltchinskii–Pollard entropy condition (cf. Rakhlin et al. (2017)) or finite VC-dimension (cf. Mendelson (2014)). Roughly speaking, under this condition, the lower isometry remainder is relatively small, i.e.

$$\mathcal{I}_L(n) \ll \epsilon_*^2.$$

However, it is insufficient to obtain a nice control over the upper isometry remainder, i.e. it may even happen that

$$\mathcal{I}_U(n) \asymp 1.$$

The ideas that appear in the small-ball method suggest that indeed a small lower isometry remainder is a mild assumption over a model $(n, \mathcal{F}, \mathbb{P})$, as for example it also provides an upper bound over

$$\sup_{f^* \in \mathcal{F}} \mathbb{E}_{\mathbf{X}} \mathrm{Diam}_{\mathbb{P}}(\{f \in \mathcal{F} : (f(X_1), \ldots, f(X_n)) = (f^*(X_1), \ldots, f^*(X_n))\}),$$

as we mentioned earlier.

*Remark* 11. The assumption of $\max\{\mathcal{I}_L(n), \mathcal{I}_U(n)\} \asymp \epsilon_*^2$ holds for uniformly bounded classes whose $\epsilon$-covering numbers are asymptotically equal to the $\epsilon$-covering numbers with bracketing (see e.g. van de Geer (2000); Birgé and Massart (1993)), which is considered a mild assumption for analyzing ERM on non-parametric and shape-constrained classes. It also holds for classes that satisfy the Koltchinskii-Pollard condition (Rakhlin et al., 2017) or the $L_2 - L_{2+\delta}$ entropy equivalence condition (see Lecué and Mendelson (2013) and references therein). In the classical regime, i.e. when $\mathcal{F}$ is fixed and $n$ grows, it is hard to construct function classes that does not satisfy this assumption for $n$ that is large enough (Birgé and Massart, 1993).

*Remark* 12. Note that a bound similar to that of Theorem 3 cannot hold for the bias error term. Indeed, one can construct a class $\mathcal{F}$ with $\mathcal{I}_L(n) \lesssim \epsilon_*^2$ and $\mathcal{V}(\widehat{f}_n, \mathcal{F}, \mathbb{P}) \asymp \epsilon_*^2$ for which the bias error term $\sup_{f^* \in \mathcal{F}} B^2(\widehat{f}_n) \asymp 1$, moreover, for this class one has

$$\mathbb{E}_{\mathbf{X}, \boldsymbol{\xi}} \|\widehat{f}_n - \mathbb{E}_{\boldsymbol{\xi}}\left[\widehat{f}_n | \mathbf{X}\right]\|_n^2 \asymp 1.$$

That is, neither the bias nor the *empirical* variance converge to zero. A remarkable consequence of our results is that even though the ERM only observes the random empirical measure $\mathbb{P}_n$, its variance, measured in terms of $\mathbb{P}$, converges to zero when $\mathcal{I}_L(n) \to 0$.

*Remark* 13. Comparing Theorem 5 to Theorem 3 of the previous sub-section, one sees that if $\mathcal{I}_L(n) \lesssim \epsilon_*^2$, the minimax optimality of the variance is implied either by a bound of $\epsilon_*^2$ for the upper isometry constant $\mathcal{I}_U(n)$ or by the Lipschitz and interpolating Assumptions 6-7, one might wonder whether the latter set of assumptions actually themselves imply such a bound on $\mathcal{I}_U(n)$.

In fact, the opposite is true: these assumptions are mutually exclusive as soon as the minimax rate is $o(1)$. Indeed, the assumption that the function class is almost interpolating (Assumption 7) means that $\widehat{f}_n(\mathbf{X}) := (\widehat{f}_n(X_1), \ldots, \widehat{f}_n(X_n))$ closely tracks the observation vector $\mathbf{Y}$ (though Assumption 7 only requires this to hold a non-negligible event, the proof of Theorem 5 shows that up to increasing the absolute constant $C$, almost interpolation actually holds with high probability). The variance of

$\mathbf{Y}$ is bounded below by that of $\boldsymbol{\xi}$, which is 1 (measured with respect to $\|\cdot\|_n$), which implies easily that the *empirical* variance of $\widehat{f}_n$ is of order $O(1)$ as well.

On the other hand, a bound of $\epsilon_*^2$ on the upper isometry constant means that up to a multiplicative factor and an additive error of $\epsilon_*^2$, when $\|\widehat{f}_n - \mathbb{E}\widehat{f}_n\|^2$ is small then so is $\|\widehat{f}_n(\mathbf{X}) - \mathbb{E}\widehat{f}_n(\mathbf{X})\|_n^2$. Taking expectations one obtains that the empirical variance of $\widehat{f}_n$ is asymptotically bounded by the population variance plus $\epsilon_*^2$, which is certainly $o(1)$.

### 5.2.3 Remarks on §4.1

*Remark* 14. Chen et al. (2017) also gave an explicit upper bound of $1.65 \cdot 10^5$ for the the constant $C > 0$ in (17). Making the constants explicit in our proof yields a bound of the order $10^2$ rather than $10^5$; we have not attempted to optimize the constants, so we believe this can be improved further.

*Remark* 15. The assumption that the evaluation functional is continuous in $L_2(\mathbb{P})$ may seem restrictive. In fact, though, the proof of Theorem 6 also goes through if there exists a stronger norm $\|\cdot\|'$ on $\mathcal{F}$ than the $L^2(\mathbb{P})$ norm such that $\mathcal{F}$ is compact and the evaluation functionals $f \mapsto f(x)$ are continuous with respect to the topology induced by $\|\cdot\|'$. (Natural examples of such $\mathcal{F}, \|\cdot\|'$ are provided by Sobolev spaces.) For simplicity, we have stated the theorem under Assumption 8.

## 6 Proofs

We begin with additional notation: Given $x_1, \ldots, x_n$ and $h : \mathcal{X} \to \mathbb{R}$, we denote

$$G_h = n^{-1} \sum_{i=1}^{n} \xi_i h(x_i).$$

For $g \in \mathcal{F}$, we set $B_n(g,t) := \{f \in \mathcal{F} : \|f - g\|_n \leq t\}$, and $B(g,t) := \{f \in \mathcal{F} : \|f - g\| \leq t\}$. Throughout the proof, we denote by $c, c_1, c_2, \ldots \in (0,1)$, an $C, C_1, \ldots \geq 0$ absolute constants (not depending on $\mathcal{F}$ or on $n$) that may change from line to line.

### 6.1 Proof of Theorem 1

In §6.1.1, we fill in the details the proof sketch that was given above under the additional assumption of (15), and in §6.1.2, we give the full proof without additional assumptions. We remark that our proof holds for any noise that satisfies the LCP property (12) defined above.

### 6.1.1 Proof of Theorem 1 under (15)

First, we show that for all $t \geq 0$ and for any fixed $f \in \mathcal{F}$, the following holds:

$$\Pr_{\boldsymbol{\xi}}\left\{\left|\|\widehat{f}_n - f\|_n - \mathbb{E}_{\boldsymbol{\xi}}\|\widehat{f}_n - f\|_n\right| \geq t\right\} \leq 2\exp(-c_L n t^2), \tag{22}$$

Indeed, this will follow immediately from the LCP condition (12) with $c_L = n^{-0.5}$ if we prove that $h(\boldsymbol{\xi}) = \|\widehat{f}_n(\boldsymbol{\xi}) - f^*\|_n$ is a $n^{-0.5}$-Lipschitz function.

To prove this claim, observe that $\widehat{f}_n(\boldsymbol{\xi})$ is the projection of $\mathbf{Y} = f^* + \boldsymbol{\xi}$ onto the convex set

$$\mathcal{F}_n := \{(f(x_1), \ldots, f(x_n)) : f \in \mathcal{F}\} \subset \mathbb{R}^n.$$

Therefore, we obtain

$$|h(\boldsymbol{\xi}_1) - h(\boldsymbol{\xi}_2)| = |\|\widehat{f}_n(\boldsymbol{\xi}_1) - f^*\|_n - \|\widehat{f}_n(\boldsymbol{\xi}_2) - f^*\|_n| \leq \|\widehat{f}_n(\boldsymbol{\xi}_1) - \widehat{f}_n(\boldsymbol{\xi}_2)\|_n$$
$$\leq \|\boldsymbol{\xi}_1 - \boldsymbol{\xi}_2\|_n = n^{-1/2}\|\boldsymbol{\xi}_1 - \boldsymbol{\xi}_2\|_2,$$

where we have used the fact that the projection to a convex set is a contracting operator. This concludes the proof of (22).

Next, fix $\epsilon > 0$ (to be chosen later), let $\mathcal{N}(\epsilon) := \mathcal{N}(\epsilon, \mathcal{F}, \mathbb{P}^{(n)})$, and let $A = \{f_1, \ldots, f_{\mathcal{N}(\epsilon)}\}$ be a minimal $\epsilon$-net of $\mathcal{F}$. By the pigeonhole principle, there exists at least one element $f_\epsilon \in A$ such that

$$\Pr(\|\widehat{f}_n - f_\epsilon\|_n \leq \epsilon) \geq 1/\mathcal{N}(\epsilon). \tag{23}$$

Also, setting $f = f_\epsilon$ in (22) we have

$$\Pr\left(|\|\widehat{f}_n - f_\epsilon\|_n - \mathbb{E}_{\boldsymbol{\xi}}\|\widehat{f}_n - f_\epsilon\|_n)| \geq t\right) \leq 2\exp(-c_L n t^2). \tag{24}$$

Taking $t = \log(4)\sqrt{\frac{\log \mathcal{N}(\epsilon)}{c_L n}}$ in (24) yields

$$\Pr\left(\left|\|\widehat{f}_n - f_\epsilon\|_n - \mathbb{E}_{\boldsymbol{\xi}}\|\widehat{f}_n - f_\epsilon\|_n\right| \geq \log(4)\sqrt{\frac{\log \mathcal{N}(\epsilon)}{c_L n}}\right) \leq \frac{1}{2\mathcal{N}(\epsilon)}. \tag{25}$$

Combining (23) and (25) via the union bound we obtain

$$\Pr\left(\|\widehat{f}_n - f_\epsilon\|_n \leq \epsilon, \left|\|\widehat{f}_n - f_\epsilon\|_n - \mathbb{E}_{\boldsymbol{\xi}}\|\widehat{f}_n - f_\epsilon\|_n\right| \leq \log(4)\sqrt{\frac{\log \mathcal{N}(\epsilon)}{c_L n}}\right) \geq \frac{1}{2|\mathcal{N}(\epsilon)|} > 0$$

Since the event of the last equation holds with positive probability, we must have

$$\mathbb{E}_{\boldsymbol{\xi}}\|\widehat{f}_n - f_\epsilon\|_n \leq \epsilon + \log(4)\sqrt{\frac{\log \mathcal{N}(\epsilon)}{c_L n}}.$$

To optimize the RHS over $\epsilon$, we take $\epsilon$ such that $\epsilon = \sqrt{\log \mathcal{N}(\epsilon)/(c_L n)}$ — i.e., $\epsilon = \epsilon_*$ — and get

$$\mathbb{E}_{\boldsymbol{\xi}}\|\widehat{f}_n - f_\epsilon\|_n \leq C\epsilon_*/\sqrt{c_L}.$$

Substituting in (24) and taking $t = \epsilon_*$, we obtain

$$\Pr(|\|\widehat{f}_n - f_\epsilon\|_n \geq C\epsilon_*/\sqrt{c_L}) \leq 2\exp(-cn\epsilon_*^2).$$

This easily implies that $\mathbb{E}[\|\widehat{f}_n - f_\epsilon\|_n^2] \leq C_1 \epsilon_*^2/c_L$, and therefore also

$$\mathbb{E}[\|\widehat{f}_n - \mathbb{E}\widehat{f}_n\|_n] \leq (\mathbb{E}[\|\widehat{f}_n - \mathbb{E}\widehat{f}_n\|_n^2])^{1/2} \leq (\mathbb{E}[\|\widehat{f}_n - f_\epsilon\|_n^2])^{1/2} \leq C_2\epsilon_*/\sqrt{c_L}.$$

Applying (22) once again, now with $f = \mathbb{E}\widehat{f}_n$, we obtain

$$\Pr(\|\widehat{f}_n - \mathbb{E}\widehat{f}_n\|_n^2 \geq C_2\epsilon_*^2/c_L) \leq 2\exp(-cn\epsilon_*^2). \tag{26}$$

### 6.1.2 Full proof of Theorem 1

First note that for any class $\mathcal{H}$, we have $\mathcal{M}(\mathcal{H}, \mathbb{P}^{(n)}) \leq \text{diam}_{\mathbb{P}^{(n)}}(\mathcal{H})^2$ (consider a constant estimator),and applying this to $\mathcal{H}_*$ yields $\mathcal{M}(\mathcal{H}_*, \mathbb{P}^{(n)}) \leq 4V_{\mathcal{D}}(\widehat{f}_n)$. Thus we need only prove the nontrivial inequality $\mathcal{M}(\mathcal{H}_*, \mathbb{P}^{(n)}) \gtrsim V_{\mathcal{D}}(\widehat{f}_n)$.

We will consider two cases: First, when $\mathcal{M}(\mathcal{H}, \mathbb{P}^{(n)}) \leq Sn^{-1}$ for sufficiently large $S \geq 0$ (i.e. the parametric case), the result follows from classical theory. The remaining case will be handled in similar fashion as in §6.1.1 above, but with a more careful analysis.

**Case I:** $V(\widehat{f}_n) \leq Sn^{-1}$. Certainly, there exists $g \in \mathcal{F}$ such that $\|g - \mathbb{E}\widehat{f}_n\|_n^2 \geq V_{\mathcal{D}}(\widehat{f}_n)$, which implies, by the convexity of $\mathcal{F}$ that there exists $h \in \mathcal{H}_*$ with $\sqrt{V(\widehat{f}_n)} \leq \|h - \mathbb{E}\widehat{f}_n\|_n \leq 2\sqrt{S/n}$. Applying the two-point method to $h$ and $\mathbb{E}\widehat{f}_n$ (see e.g., (Wainwright, 2019, Example 15.4)), one sees easily that the minimax rate of $\mathcal{H}_*$ is $\Omega(V(\widehat{f}_n))$.

**Case II:** $V(\widehat{f}_n) \geq Sn^{-1}$. To treat this case we the following characterization of the minimax rate in the fixed design setting (Neykov, 2022; Yang and Barron, 1999):

**Lemma 1** (Theorem 2.11 in Neykov (2022)). *Under Assumptions 1,2, $\mathcal{M}(\mathcal{H}, \mathbb{P}^{(n)}) \asymp \epsilon_*^2$, where $\epsilon_*$ solves the equation*

$$\log \mathcal{N}^{loc}(\epsilon, \mathcal{H}, \mathbb{P}^{(n)}) \asymp n\epsilon^2, \tag{27}$$

*where $\mathcal{N}^{loc}(\epsilon, \mathcal{F}, \mathbb{P}^{(n)}) = \sup_{f \in \mathcal{F}} \mathcal{N}(\epsilon/4, \{h \in \mathcal{H} : \|h - f\|_n \leq \epsilon\}, \mathbb{P}^{(n)})$.*

By the lemma, it suffices to show that

$$\log \mathcal{N}^{loc}(2\sqrt{V(\hat{f}_n)}, \mathcal{H}_*, \mathbb{P}^{(n)}) \gtrsim nV(\hat{f}_n),$$

as this will imply that $2\sqrt{V(\hat{f}_n)} \leq \epsilon_*$ and hence $V(\hat{f}_n) \lesssim \mathcal{M}(\mathcal{H}_*, \mathbb{P}^{(n)})$. Noting that

$$\{h \in \mathcal{H}_* : \|h - \mathbb{E}\widehat{f}_n\|_n \leq 2\sqrt{V(\hat{f}_n)}\} = \mathcal{H}_*$$

we have $\mathcal{N}^{\text{loc}}(\sqrt{V(\widehat{f}_n)}/2, \mathcal{F}, \mathbb{P}^{(n)}) \geq \mathcal{N}(\sqrt{V(\widehat{f}_n)}/2, \mathcal{H}_*, \mathbb{P}^{(n)})$ and hence it suffices to show that

$$\log \mathcal{N}(\sqrt{V(\widehat{f}_n)}/2, \mathcal{H}_*, \mathbb{P}^{(n)}) \geq c_1 n \cdot V(\widehat{f}_n) \tag{28}$$

for an appropriate $c_1 > 0$ to be chosen later.

Suppose to the contrary that $\log N \leq c_1 n \cdot V(\widehat{f}_n)$, where $N = \mathcal{N}(\sqrt{V(\widehat{f}_n)}/2, \mathcal{H}_*, \mathbb{P}^{(n)})$.

We consider the distribution of $\widehat{f}_n$ when the true function is $f^*$. First, note that as $\mathbb{E}\|\widehat{f}_n - \mathbb{E}\widehat{f}_n\|^2 = V(\widehat{f}_n)$, we have that

$$\Pr(\widehat{f}_n \in \mathcal{H}_*) = \Pr(\widehat{f}_n \in B_n(\mathbb{E}_{\mathcal{D}}\widehat{f}_n, 2\sqrt{V(\widehat{f}_n)})) = 1 - \Pr(\|\widehat{f}_n - \mathbb{E}_{\mathcal{D}}\widehat{f}_n\|_n^2 \geq 4V(\widehat{f}_n)) \geq 3/4$$

by Chebyshev's inequality. Let $A = \{f_1, \ldots, f_{\mathcal{N}}\}$ be a minimal $\sqrt{V(\widehat{f}_n)}/2$-net in $\mathcal{H}_*$; by the pigeonhole principle, there exists at least one element $g \in A$ such that

$$\Pr(\|\widehat{f}_n - g\|_n \leq \sqrt{V(\widehat{f}_n)}/2) \geq \frac{3}{4N} \geq 3\exp(-c_1 nV(\widehat{f}_n))/4. \tag{29}$$

Next, we apply (22) with $f = g$ and $t = \sqrt{V(\widehat{f}_n)}/6$, to obtain

$$\Pr_{\boldsymbol{\xi}}\left(|\|\widehat{f}_n - g\|_n - \mathbb{E}_{\boldsymbol{\xi}}\|\widehat{f}_n - g\|_n| \leq \sqrt{V(\widehat{f}_n)}/6\right) \geq 1 - 2\exp(-nV(\widehat{f}_n)/18). \tag{30}$$

Recalling that we are in the case $V(\widehat{f}_n) \geq Sn^{-1}$, by choosing $c_1 > 0$ small enough and $S > 0$ large enough we can ensure that $\exp(nV(\widehat{f}_n)(1/18 - c_1)) > 8/3$, or equivalently

$$\frac{3}{4}\exp(-c_1 nV(\widehat{f}_n)) - 2\exp(-nV(\widehat{f}_n)/18) > 0. \tag{31}$$

Combining (29), (30), and (31) yields

$$\Pr(\|\widehat{f}_n - g\|_n \leq \sqrt{V(\widehat{f}_n)}/2) + \Pr\left(|\|\widehat{f}_n - g\|_n - \mathbb{E}_{\boldsymbol{\xi}}\|\widehat{f}_n - g\|_n| < \sqrt{V(\widehat{f}_n)}/6\right) > 1,$$

so the two events

$$\left\{|\|\widehat{f}_n - g\|_n - \mathbb{E}_{\boldsymbol{\xi}}\|\widehat{f}_n - g\|_n| < \sqrt{V(\widehat{f}_n)}/6\right\}, \{\|\widehat{f}_n - g\|_n \leq \sqrt{V(\widehat{f}_n)}/2\}$$

have nonempty intersection, which implies that $\mathbb{E}_{\mathcal{D}}\|\widehat{f}_n - g\|_n < 2\sqrt{V(\widehat{f}_n)}/3$.

Let $h(\xi) = \|\widehat{f}_n - g\|_n$. We have $\mathbb{E}h^2 = (\mathbb{E}h)^2 + \mathbb{E}(h - \mathbb{E}h)^2 < 4V(\widehat{f}_n)/9$. As $h$ is $\frac{1}{\sqrt{n}}$-Lipschitz, the LCP implies that $h$ is $\frac{1}{\sqrt{n}}$-subgaussian. Thus $h - \mathbb{E}h$ is a centered $\frac{1}{\sqrt{n}}$-subgaussian random variable, so $\mathbb{E}(h - \mathbb{E}h)^2 \leq \frac{2}{n}$ (Vershynin, 2018, Proposition 2.5.2), and hence

$$\mathbb{E}_{\mathcal{D}}\|\widehat{f}_n - g\|_n^2 < \frac{4}{9}V(\widehat{f}_n) + \frac{2}{n}.$$

Again recalling that $V(\widehat{f}_n) > Sn^{-1}$, by taking $S$ large enough we can ensure that $\mathbb{E}_{\boldsymbol{\xi}}\|\widehat{f}_n - \mathbb{E}\widehat{f}_n\|_n^2 < V(\widehat{f}_n)$, which contradicts the definition of $V(\widehat{f}_n)$.

It remains to prove the last statement of the theorem, namely that $\sup_{f \in \mathcal{O}_{\delta_n}} \|f - \mathbb{E}\widehat{f}_n\|_n^2 \asymp \mathcal{M}(\mathcal{H}_*, \mathbb{P}^{(n)}))$ with high probability. We have seen that $\mathbb{E}\|\widehat{f}_n - \mathbb{E}\widehat{f}_n\|_n^2 \lesssim \mathcal{M}(\mathcal{H}_*, \mathbb{P}^{(n)})$, so $\mathbb{E}\|\widehat{f}_n - \mathbb{E}\widehat{f}_n\|_n \leq C\sqrt{\mathcal{M}(\mathcal{H}_*, \mathbb{P}^{(n)})}$. Applying (22) once again with $t = \sqrt{\mathcal{M}(\mathcal{H}_*, \mathbb{P}^{(n)})}$, we have

$$\mathbb{P}(\|\widehat{f}_n - \mathbb{E}\widehat{f}_n\|_n^2 \geq C\mathcal{M}(\mathcal{H}_*, \mathbb{P}^{(n)})) \leq 2\exp(-cn\mathcal{M}(\mathcal{H}_*, \mathbb{P}^{(n)})). \tag{32}$$

Condition on the high-probability event of (32) above, and consider some $f \in \mathcal{O}_{\delta_n}$. Since

$$\|f - \mathbb{E}\widehat{f}_n\|_n^2 \leq 2(\|f - \widehat{f}_n\|_n^2 + \|\widehat{f}_n - \mathbb{E}\widehat{f}_n\|_n^2),$$

to obtain the theorem it suffices to show that for any $f \in \mathcal{O}_{\delta_n}$, we have

$$\|f - \widehat{f}_n\|_n^2 \leq \delta_n^2$$

deterministically.

This is a matter of elementary convex geometry: we know that $\widehat{f}_n$ is the closest point in the convex set $\mathcal{F}_n$ to the point $\mathbf{Y}$, which implies that the ball $B = B(\mathbf{Y}, \|\widehat{f}_n - \mathbf{Y}\|_n)$ is tangent to $\mathcal{F}_n$ at $\widehat{f}_n$. This implies that $\mathcal{F}_n$ is contained within the positive half-space $H^+$ defined by the supporting hyperplane of $B$ at $\widehat{f}_n$, i.e.,

$$\mathcal{F}_n \subset H^+ = \{f : \langle \widehat{f}_n - \mathbf{Y}, f - \mathbf{Y} \rangle \geq \|\widehat{f}_n - \mathbf{Y}\|^2\}. \tag{33}$$

We now compute:

$$\|f - \mathbf{Y}\|_n^2 = \|f - \widehat{f}_n\|_n^2 + \|\widehat{f}_n - \mathbf{Y}\|_n^2 + 2\langle \widehat{f}_n - \mathbf{Y}, f - \widehat{f}_n \rangle_n.$$

Since $f \in \mathcal{F}_n$, (33) implies that $\langle f - \mathbf{Y}, \widehat{f}_n - \mathbf{Y} \rangle_n \geq \langle \widehat{f}_n - \mathbf{Y}, \widehat{f}_n - \mathbf{Y} \rangle_n$, or equivalently, $\langle f - \widehat{f}_n, \widehat{f}_n - \mathbf{Y} \rangle_n \geq 0$. Hence we obtain

$$\|f - \widehat{f}_n\|_n^2 \leq \|f - \mathbf{Y}\|_n^2 - \|\widehat{f}_n - \mathbf{Y}\|_n^2,$$

but the RHS is at most $\delta_n$ by the definition of $\mathcal{O}_{\delta_n}$. This concludes the proof.

## 6.2 Proof of Theorem 2

By the definition of the minimax risk, there exists some $f^* \in \mathcal{F}$ with risk at least $\delta^2 := \mathcal{M}(\mathcal{F}, \mathbb{P}^{(n)})$. By translating $\mathcal{F}$, we may assume $f^* = 0$ without loss of generality, so that $\mathbb{E}_{\boldsymbol{\xi}}[\|\widehat{f}_n\|_n^2] \gtrsim \delta^2$.

Write $\widehat{f}_n(\boldsymbol{\xi})$ for the ERM computed when the target function is $f^* = 0$ and the noise is $\boldsymbol{\xi}$, namely, the projection of $\boldsymbol{\xi}$ onto $\mathcal{F}$. We wish to show that $\mathbb{E}_{\boldsymbol{\xi}}[\|\widehat{f}_n - \mathbb{E}\widehat{f}_n\|_n^2] \gtrsim \delta^4$.

The fact that $\widehat{f}_n(\boldsymbol{\xi})$ is the projection of the observation vector $\boldsymbol{\xi}$ on the convex set $\mathcal{F}$ implies, by convexity, that $\langle f - \widehat{f}_n, \widehat{f}_n - \boldsymbol{\xi} \rangle_n \geq 0$ for any $f \in \mathcal{F}$ (see §6.1 for the easy argument). Substituting $f = f^* = 0$ and rearranging immediately yields that for any $\boldsymbol{\xi}$,

$$\langle \widehat{f}_n(\boldsymbol{\xi}), \boldsymbol{\xi} \rangle_n \geq \|\widehat{f}_n(\boldsymbol{\xi})\|_n^2.$$

Write $f_e = \mathbb{E}_{\boldsymbol{\xi}}\widehat{f}_n(\boldsymbol{\xi})$. Since $\mathbb{E}_{\boldsymbol{\xi}}\boldsymbol{\xi} = 0$, we may take expectations and insert $\widehat{f}_e$ to obtain

$$\mathbb{E}_{\boldsymbol{\xi}}\langle \widehat{f}_n(\boldsymbol{\xi}) - f_e, \boldsymbol{\xi} \rangle_n \geq \mathbb{E}_{\boldsymbol{\xi}}\|\widehat{f}_n(\boldsymbol{\xi})\|_n^2 \geq \delta^2.$$

Applying Cauchy-Schwarz, we obtain

$$\mathbb{E}_{\boldsymbol{\xi}}\|\widehat{f}_n(\boldsymbol{\xi}) - f_e\|_n^2 \cdot \mathbb{E}_{\boldsymbol{\xi}}\|\boldsymbol{\xi}\|_n^2 \geq \delta^4,$$

and because the noise is isotropic we immediately obtain

$$V(\widehat{f}_n) = \mathbb{E}_{\boldsymbol{\xi}}[\|\widehat{f}_n(\boldsymbol{\xi}) - f_e\|_n^2] \geq \delta^4,$$

as desired.

## 6.3 Proof of Corollary 2

Here, we prove this result when $\mathcal{F}$ is compact with respect to $L_2(\mathbb{P}^{(n)})$. For completeness, we provide a proof without this assumption in §7.1 below.

Consider the map $F : \mathcal{F}_n \to \mathbb{R}^n$ defined via

$$f^* \to \mathbb{E}_{\boldsymbol{\xi}} \widehat{f}_n,$$

i.e., $f^*$ maps to the expectation of the ERM $\widehat{f}_n$ when the underlying function is $f^* \in \mathcal{F}_n$. One verifies easily that $F$ is continuous, since projection to a convex set is a 1-Lipschitz function; in addition, the convexity of $\mathcal{F}_n$ implies that $F(f^*) \in \mathcal{F}_n$ for all $f^* \in \mathcal{F}_n$. Thus $F$ is a continuous map from the compact convex set $\mathcal{F}_n$ to itself, so by the Brouwer fixed point theorem, there exists an $f^* \in \mathcal{F}_n$ such that

$$f^* = \mathbb{E}_{\boldsymbol{\xi}} \widehat{f}_n.$$

Let $\mathrm{E}_{f^*}^2 = \mathbb{E}_{\boldsymbol{\xi}} \|\widehat{f}_n - f^*\|_n^2 = V(\widehat{f}_n)$, and let $P$ denote the projection to $\mathcal{F}$, which is is 1-Lipschitz. For any $f \in \mathcal{G}_* = B(f^*, 2\mathrm{E}_{f^*}) \cap \mathcal{F}$ and any $\xi$ we have

$$\|P(f + \boldsymbol{\xi}) - f\|_n^2 \le 3(\|P(f + \boldsymbol{\xi}) - P(f^* + \boldsymbol{\xi})\|_n^2 + \|P(f^* + \boldsymbol{\xi}) - f^*\|_n^2 + \|f^* - f\|_n^2)$$
$$\le 3(\|P(f^* + \boldsymbol{\xi}) - f^*)\|_n^2 + 8\mathrm{E}_{f^*}^2) = 3(\|\widehat{f}_n - f^*)\|_n^2 + 8\mathrm{E}_{f^*}^2),$$

and taking the expectation over $\boldsymbol{\xi}$ we see that $\mathbb{E}_{\boldsymbol{\xi}} \|P(f + \boldsymbol{\xi}) - f\|_n^2$, the squared error of the ERM when the underlying function is $f$, is at most $27\mathrm{E}_{f^*}^2$.

Now let $\hat{h} : \mathbb{R}^n \to \mathcal{F}$ be any estimator. By Theorem 1, we have

$$\mathrm{E}_{f^*}^2 \lesssim \max_{f \in \mathcal{G}_*} \|\hat{h}(f + \xi) - f\|_n^2.$$

Picking $f \in \mathcal{G}_*$ which maximizes the error of $\hat{h}$, we have that the squared error of $\widehat{f}_n$ on $f$ is upper-bounded by $c \cdot \mathrm{E}_{f^*}^2$ and the squared error of $\hat{h}$ on $f$ is lower-bounded by $c_1 \cdot \mathrm{E}_{f^*}^2$, which is precisely what we want.

*Remark* 16. The Brouwer fixed point theorem, which we use in the first step of the proof to obtain the existence of $f^* \in \mathcal{F}$ for which $\mathbb{E}_{\boldsymbol{\xi}} \widehat{f}_n = f^*$, is a deep result, and one may ask whether it is essential to the proof. Another commonly used fixed-point theorem is that due to Banach; the Banach fixed point theorem is elementary, but requires a bound $\|F(f) - F(g)\|_n \le c\|f - g\|_n$ for some $c < 1$ and all $f, g \in \mathcal{F}$.

One has
$$\|F(f) - F(g)\|_n \le \mathbb{E}_{\boldsymbol{\xi}} \|P(f + \boldsymbol{\xi}) - P(g + \boldsymbol{\xi})\|_n, \tag{34}$$
where $P$ denotes the projection to $\mathcal{F}$. Note that $\|P(f + \boldsymbol{\xi}) - P(g + \boldsymbol{\xi})\|_n \le \|f - g\|_n$ because $P$ is 1-Lipschitz. Also, it's easy to see that there exists some $\boldsymbol{\xi}$ for which $\|P(f + \boldsymbol{\xi}) - P(g + \boldsymbol{\xi})\|_n$ is strictly smaller than $\|f - g\|_n$, and continuity of $P$ ensures that the same holds for all $\boldsymbol{\xi}'$ sufficiently close to $\boldsymbol{\xi}$, implying $\|F(f) - F(g)\|_n < \|f - g\|_n$. But this is not yet sufficient to apply the Banach fixed point theorem.

Via more delicate convex-geometric arguments, though, one can show that if $\|\xi\|_n$ is sufficiently large compared to the diameter of $\mathcal{F}$ (say, $\|\xi\|_n > C \cdot \mathrm{diam}(\mathcal{F})$) and $\langle f - g, \boldsymbol{\xi} \rangle_n \ge \epsilon \|f - g\|_n \|\xi\|_n$ (i.e., the angle between $f - g$ and $\boldsymbol{\xi}$ is bounded away from 90 degrees) then

$$\|P(f + \boldsymbol{\xi}) - P(g + \boldsymbol{\xi})\|_n \le (1 - \delta)\|f - g\|_n$$

for some $\delta$ depending on $\epsilon$ and $C$, which allows one to conclude, using (34), that $\|F(f) - F(g)\|_n \le c\|f - g\|_n$ for some $c < 1$ and all $f, g \in \mathcal{F}$. Hence, the Banach fixed point theorem can be used in the proof instead of the Brouwer fixed point theorem, rendering it elementary but more technical.

## 6.4 Proof of Theorem 3

**Preliminaries** The main tool we use from the theory of empirical processes is Talagrand's inequality (Koltchinskii, 2011, Theorem 2.6):

**Lemma 2.** *Let $\mathcal{H}$ be a class of functions on a domain $\mathcal{Z}$ all of which are uniformly bounded by $M$. Let $Z_1, \ldots, Z_n \underset{i.i.d.}{\sim} \mathbb{P}$. Then, there exist universal constants $C, c > 0$ such that*

$$\Pr(|\|\mathcal{H}\|_n - \mathbb{E}\|\mathcal{H}\|_n| \geq t) \leq C \exp\left(-\frac{cnt}{M} \log\left(1 + \frac{t}{\mathbb{E}\|\mathcal{H}^2\|_n}\right)\right),$$

*where $\|\mathcal{H}\|_n := \sup_{h \in \mathcal{H}} n^{-1} \sum_{i=1}^n h(Z_i)$, and $\|\mathcal{H}^2\|_n = \sup_{h \in \mathcal{H}} n^{-1} \sum_{i=1}^n h(Z_i)^2$.*

**Lemma 3.** *Under Assumptions 1,3,4, the following holds:*
$$\epsilon_*^2 \gtrsim \log(n)/n,$$

*where $\epsilon_*^2$ is defined in (10) above.*

**Proof of Lemma 3.** Note that by Assumptions 1,3, we have that
$$\log \mathcal{N}(\epsilon, \mathcal{F}, \mathbb{P}) \gtrsim \log(\epsilon^{-1}),$$

Therefore, we obtain that $\epsilon_*$ is greater or equal to the stationary point of the following equation:
$$\log(\epsilon^{-1})/n \asymp \epsilon^2,$$

so $\epsilon_* \gtrsim \sqrt{\log n/n}$. $\qquad\square$

**Proof of Theorem 3:** We abbreviate $\mathcal{I}_L := \mathcal{I}_L(n)$, $\mathcal{I}_U := \mathcal{I}_U(n)$ and note that by the last lemma, we may assume that $\epsilon_*^2 > C \log(n)/n$ for sufficiently large $C \geq 0$. For every fixed $\mathbf{X}, \boldsymbol{\xi}$, the function $\widehat{\mathcal{L}} : \mathcal{F} \to \mathbb{R}$ defined by

$$\widehat{\mathcal{L}}(f) := \|\mathbf{Y} - f\|_n^2 - \|\boldsymbol{\xi}\|_n^2 = -2G_{f-f^*} + \|f - f^*\|_n^2, \tag{35}$$

satisfies $\widehat{f}_n = \operatorname{argmin}_{f \in \mathcal{F}} \widehat{\mathcal{L}}(f)$. (Of course, $\widehat{\mathcal{L}}(f)$ is just the empirical loss of $f$, up to subtracting a constant.) Note that $\widehat{\mathcal{L}}(f^*) = 0$, so $\widehat{\mathcal{L}}(\widehat{f}_n)$ must be non-positive.

Let $\{f_1, \ldots, f_N\}$ be an $\epsilon_U$-net of $\mathcal{F}$ with respect to $\mathbb{P}$ of cardinality $N = \mathcal{N}(\epsilon_U, \mathcal{F}, \mathbb{P})$; for each $i \in [N]$, let $B(f_i) := B(f_i, \epsilon_U)$ denote the ball of radius $\epsilon_U$ around $f_i$, so that the $B(f_i)$ cover $\mathcal{F}$. For each $i$, let $L_i$ denote the minimal loss on the ball $B(f_i)$:

$$L_i := \min_{f \in B(f_i)} \widehat{\mathcal{L}}(f) \tag{36}$$

The main technical result is the following lemma:

**Lemma 4.** *Fix $i_* \in [N]$. For any absolute constant $A > 0$, there exist absolute constants $C_1, C, > 0$, such that the following holds with probability of at least $1 - 2\exp(-An\epsilon_U^4/\max\{\mathcal{I}_U, \epsilon_*^2\})$:*

$$\forall i \in [N], \ |(L_i - L_{i_*}) - \mathbb{E}(L_i - L_{i_*})| \leq C_1 \epsilon_U^2 + \frac{1}{4}\|f_i - f_{i_*}\|^2. \tag{37}$$

We defer the proof of Lemma 4 to the end of the section, and show how it implies the theorem.

**Proof of Theorem 3 (assuming Lemma 4).** We apply Lemma 4 with $i_* = \operatorname{argmin}_{i \in [N]} \mathbb{E}L_i$. Let $\mathcal{E}$ denote the event of Lemma 4 (the constant $A > 0$ in the lemma will be chosen shortly), and let $\mathcal{E}'$ be the event that

$$\|f - g\|_n^2 \geq \frac{1}{2}\|f - g\|^2 - \mathcal{I}_L \tag{38}$$

for all $f, g \in \mathcal{F}$. By the definition of $\mathcal{I}_L$, $\mathcal{E}'$ holds with probability $1 - n^{-1}$; in addition, a mildly tedious computation, which we defer to Lemma 6, shows that $A$ can be chosen such that $\Pr(\mathcal{E}) \geq 1 - n^{-1}$ as well. In the remainder of the proof, we work on $\mathcal{E} \cap \mathcal{E}'$.

Let $\widehat{f}_{i_*} = \operatorname{argmin}_{f \in B(f_{i_*})} \widehat{\mathcal{L}}(f)$, so that $L_{i_*} = \widehat{\mathcal{L}}(\widehat{f}_{i_*})$. Consider the function $h = \frac{\widehat{f}_{i_*} + \widehat{f}_n}{2}$, which lies in $\mathcal{F}$ as $\mathcal{F}$ is convex. We have

$$\widehat{\mathcal{L}}(h) = -2G_{h-f^*} + \|h - f^*\|_n^2$$
$$= \frac{\widehat{\mathcal{L}}(\widehat{f}_{i_*}) + \widehat{\mathcal{L}}(\widehat{f}_n)}{2} + \left(\|h - f^*\|_n^2 - \frac{\|\widehat{f}_{i_*} - f^*\|_n^2 + \|\widehat{f}_n - f^*\|_n^2}{2}\right). \tag{39}$$

Applying the parallelogram law

$$\|a + b\|_n^2 + \|a - b\|_n^2 = 2\|a\|_n^2 + 2\|b\|_n^2$$

with $a = \frac{\widehat{f}_{i^*} - f^*}{2}, b = \frac{\widehat{f}_n - f^*}{2}$ yields

$$\|h - f^*\|_n^2 - \frac{\|\widehat{f}_{i^*} - f^*\|_n^2 + \|\widehat{f}_n - f^*\|_n^2}{2} = -\left\|\frac{\widehat{f}_{i^*} - \widehat{f}_n}{2}\right\|_n^2.$$

Combining this equation with (39) yields

$$\widehat{\mathcal{L}}(h) \le \frac{\widehat{\mathcal{L}}(\widehat{f}_{i^*}) + \widehat{\mathcal{L}}(\widehat{f}_n)}{2} - \left\|\frac{\widehat{f}_{i^*} - \widehat{f}_n}{2}\right\|_n^2.$$

But we also know that $\widehat{\mathcal{L}}(h) \ge \widehat{\mathcal{L}}(\widehat{f}_n)$ by the definition of $\widehat{f}_n$, so rearranging we obtain

$$\widehat{\mathcal{L}}(\widehat{f}_n) \le \widehat{\mathcal{L}}(\widehat{f}_{i^*}) - \frac{1}{2}\|\widehat{f}_{i^*} - \widehat{f}_n\|_n^2. \tag{40}$$

Now let $i \in N$ such that $\widehat{f}_n \in B(f_i)$ and substitute $L_i = \widehat{\mathcal{L}}(\widehat{f}_n)$, giving

$$L_i \le L_{i_*} - \frac{1}{2}\|\widehat{f}_{i^*} - \widehat{f}_n\|_n^2.$$

Since we are on $\mathcal{E}$, we may apply (37) and obtain

$$\mathbb{E}L_i \le \mathbb{E}L_{i_*} + C_3\epsilon_U^2 - \frac{1}{4}\|\widehat{f}_{i^*} - \widehat{f}_n\|_n^2 \le \mathbb{E}L_{i_*} + C_4\epsilon_U^2 + \mathcal{I}_L - \frac{1}{8}\|\widehat{f}_{i^*} - \widehat{f}_n\|^2.$$

But $\mathbb{E}L_{i_*} \le \mathbb{E}L_i$ by our choice of $i_*$, which implies finally that $\|\widehat{f}_{i^*} - \widehat{f}_n\|_n^2 \lesssim \max\{\epsilon_U^2, \mathcal{I}_L\} = \epsilon_V^2$.

Recall that we are also interested in $f \in \mathcal{O}_{\delta_n}$ for $\delta_n = O(\epsilon_V^2)$. By the geometric argument in the proof of Theorem 1, for such $f$, we have $\|f - \widehat{f}_n\|_n = O(\delta_n)$.

Applying the lower isometry property (38), we obtain

$$\|\widehat{f}_n - \widehat{f}_{i^*}\|, \|f - \widehat{f}_n\| \le C\epsilon_V$$

for any $f \in \mathcal{O}_{\delta_n}$ on $\mathcal{E} \cap \mathcal{E}'$. Since $\|\widehat{f}_{i^*} - f_{i_*}\| \le \epsilon_V$ (as $\widehat{f}_{i^*} \in B(f_{i_*})$ by definition), we also have

$$\|\widehat{f}_n - f_{i_*}\|, \|f - f_{i_*}\| \le C\epsilon_V.$$

In sum, thus far we have shown that under $\mathcal{E} \cap \mathcal{E}'$, an event of probability at least $1 - 2n^{-1}$ any $f \in \mathcal{O}_{\delta_n}$ satisfies $\|f - f_{i_*}\| \lesssim \epsilon_V$. It remains to show that this implies that $\|f - \mathbb{E}\widehat{f}_n\| \lesssim \epsilon_V$, for which it suffices to show that $\|\mathbb{E}\widehat{f}_n - f_{i_*}\| \lesssim \epsilon_V$. But

$$\mathbb{E}\widehat{f}_n - f_{i_*} = (\mathbb{E}[\widehat{f}_n | \mathcal{E} \cap \mathcal{E}'] - f_{i_*})\Pr(\mathcal{E} \cap \mathcal{E}') + (\mathbb{E}[\widehat{f}_n | (\mathcal{E} \cap \mathcal{E}')^c] - f_{i_*})\Pr((\mathcal{E} \cap \mathcal{E}')^c).$$

By what we have shown, $\|\mathbb{E}[\widehat{f}_n | \mathcal{E} \cap \mathcal{E}'] - f_{i_*}\| \le C\epsilon_V$, while $\|\mathbb{E}[\widehat{f}_n | (\mathcal{E} \cap \mathcal{E}')^c] - f_{i_*}\| = O(1)$ by Assumption 4 and so the norm of the second term is asymptotically bounded by $O(n^{-1}) \ll \epsilon_* \le \epsilon_V$ because $\epsilon_* \gtrsim \sqrt{\frac{\log n}{n}}$ by Lemma 3. This concludes the proof. $\qquad\square$

It remains to prove the deferred lemmas. We begin with the most substantial one, Lemma 4.

**Proof of Lemma 4.** Recall that

$$-L_i = \sup_{f \in B(f_i)} (2G_{f-f^*} - \|f - f^*\|_n^2).$$

We write $f - f^* = (f - f_i) + (f_i - f^*)$, expand, and decompose this expression into terms depending on $f - f_i$ and terms depending only on $f_i - f^*$:

$$-L_i = A_i + A'_i, \tag{41}$$

where

$$A_i := \sup_{f \in B(f_i)} \left(2G_{f-f_i} - \|f - f_i\|_n^2 - 2\langle f - f_i, f_i - f^*\rangle_n\right),$$

$$A'_i := 2G_{f_i-f^*} - \|f_i - f^*\|_n^2.$$

We also write

$$
\begin{aligned}
B_i := A'_i - A'_{i_*} &= 2G_{f_i-f^*} - \|f_i - f^*\|_n^2 - \left(2G_{f_{i_*}-f^*} - \|f_{i_*} - f^*\|_n^2\right) \\
&= 2G_{f_i-f_{i_*}} - \|f_i\|_n^2 + \|f_{i_*}\|_n^2 + 2\langle f_i - f_{i_*}, f^*\rangle \\
&= 2G_{f_i-f_{i_*}} + \|f_i - f_{i_*}\|_n^2 - 2\|f_i\|_n^2 + 2\langle f_i, f_{i^*}\rangle + 2\langle f_i - f_{i_*}, f^*\rangle \\
&= 2G_{f_i-f_{i_*}} + \|f_i - f_{i_*}\|_n^2 + 2\langle f_i - f_{i_*}, f^* - f_i\rangle
\end{aligned}
$$

We claim that with probability $1 - C\exp(-cn\epsilon_U^4/\max\{\epsilon_*^2, \mathcal{I}_U\})$ the following holds:

$$\forall i \in [N], \quad |A_i - \mathbb{E}A_i| \leq C_1\epsilon_U^2, \tag{42}$$

$$\forall i \in [N], \quad |B_i - \mathbb{E}B_i| \leq C_2\epsilon_U^2 + \frac{1}{4}\|f_i - f_{i_*}\|^2. \tag{43}$$

Since $L_i - L_{i_*} = A_{i_*} - A_i - B_i$, combining (42) and (43) yields the lemma.

We first prove (42). For each $i \in [N]$, we control fluctuations of $A_i$ by applying Talagrand's inequality. To this end, write

$$A_i = \sup_{f \in B(f_i)} \frac{1}{n} \sum_{j=1}^n a_f(X_j, \xi_j)$$

where

$$a_f(x, \xi) = 2\xi(f(x) - f_i(x)) - 2(f(x) - f_i(x))(f_i(x) - f^*(x)) - (f(x) - f_i(x))^2.$$

To apply Talagrand's inequality, we need to bound $\mathbb{E}\sup_{f \in B(f_i)} n^{-1}\sum_{j=1}^n a_f(X_j, \xi_j)^2$.

Using the identity $(a + b + c)^2 \leq 3(a^2 + b^2 + c^2)$, we see that

$$
\mathbb{E}_{\mathbf{X},\boldsymbol{\xi}} \sup_{f \in B(f_i)} \int a_f(X_j, \xi_j)^2 d\mathbb{P}_n
$$
$$
\leq 3 \cdot \mathbb{E}_{\mathbf{X},\boldsymbol{\xi}} \sup_{f \in B(f_i)} \int \left(2\xi^2(f(x) - f_i(x))^2 + 2(f(x) - f_i(x))^2(f_i(x) - f^*(x))^2 + (f(x) - f_i(x))^4\right) d\mathbb{P}_n.
$$

Using the assumptions $|\xi_i| \leq \Gamma_1$, $\|f\|_\infty \leq \Gamma_2$, where $\Gamma_1, \Gamma_2 > 0$ are some absolute constants, one can obtain that

$$
\mathbb{E}_{\mathbf{X},\boldsymbol{\xi}} \sup_{f \in B(f_i)} \int a_f(X_j, \xi_j)^2 d\mathbb{P}_n \leq C \cdot \mathbb{E}_{\mathbf{X}} \sup_{f \in B(f_i)} \int (f - f_i)^2 d\mathbb{P}_n
$$

for $C \leq 12\max\{\Gamma_1^2, \Gamma_2^2\}$. Using the definition of the upper isometry constant $\mathcal{I}_U$ and the stationary point $\epsilon_U$, we obtain

$$
\mathbb{E}_{\mathbf{X},\boldsymbol{\xi}} \sup_{f \in B(f_i)} \int a_f(X_j, \xi_j)^2 d\mathbb{P}_n \lesssim \max\{\mathcal{I}_U, \epsilon_U^2\} \asymp \max\{\mathcal{I}_U, \epsilon_*^2\},
$$

where the last step uses Lemma 5 below.

Thus we may apply Talagrand's inequality to $A_i$ with $\mathbb{E}\|\mathcal{H}^2\|_n \lesssim \max\{\epsilon_*^2, \epsilon_U^2\}$, giving

$$\Pr_{\mathbf{X},\boldsymbol{\xi}}\{|A_i - \mathbb{E}A_i| \geq t\} \leq C\exp(-cnt\log(1 + t/\max\{\epsilon_*^2, \mathcal{I}_U\})). \tag{44}$$

Taking a union bound over $i \in [N]$, we obtain

$$\Pr_{\boldsymbol{\xi}}\{\exists i \in [N] : |A_i - \mathbb{E}A_i| \geq t\} \leq C \exp(-cnt^2/\max\{\epsilon_*^2, \mathcal{I}_U\} + \log N).$$

Choosing $t = C_1 \epsilon_U^2$ for $C_1$ sufficiently large and recalling that $\log N = \log \mathcal{N}(\epsilon, \mathcal{F}, \mathbb{P}) \leq n\epsilon_U^4/\max(\mathcal{I}_U, \epsilon_*^2)$ by (11), we obtain that

$$\forall i \in [N], \ |A_i - \mathbb{E}A_i| \leq C_1 \epsilon_U^2$$

with probability at least $1 - 2\exp(-c_1 n\epsilon_U^4/\max\{\epsilon_*^2, \mathcal{I}_U\})$, which is (42).

Next, we handle $B_i$ for every $i \in [N]$. As in the case of $A_i$, we may write $B_i = n^{-1}\sum_{j=1}^n b_i(X_i, \xi_i)$ where

$$b_i(x, \xi) = 2\xi(f_i(x) - f_{i_*}(x)) + (f_i(x) - f_{i_*}(x))^2 + 2(f_i(x) - f_{i_*}(x))(f^*(x) - f_i(x)).$$

We have $|b_i(x, \xi)| \leq C|f_i(x) - f_{i_*}(x)|$, so as before,

$$\frac{1}{n}\sum_{j=1}^n \mathbb{E}[b_i(X_j, \xi_j)^2] \leq C\mathbb{E}[\|f_i - f_{i_*}\|_n^2] = C\|f_i - f_{i_*}\|^2,$$

and hence by Bernstein's inequality,

$$\Pr(|B_i - \mathbb{E}B_i| \geq t) \leq \exp\left(-\frac{cnt^2}{C_3 t + \|f_i - f_{i_*}\|^2}\right).$$

Substituting $t = t_i := C\epsilon_U^2 + \|f_i - f_{i_*}\|^2/4$, we obtain

$$\Pr\left(|B_i - \mathbb{E}B_i| \geq C_2\epsilon_U^2 + \frac{\|f_i - f_{i_*}\|^2}{4}\right) \leq 2\exp\left(\frac{-c_1 nt_i^2}{C_3 t_i + \|f_i - f_{i_*}\|^2}\right)$$

$$\leq 2\exp\left(-c_2 n \max\left\{C\epsilon_U^2, \frac{\|f_i - f_{i_*}\|^2}{4}\right\}\right) \quad (45)$$

$$\leq 2\exp(-c_3 n \cdot C\epsilon_U^2).$$

By the same exact argument as in the case of $A_i$, we may choose $C > 0$ sufficiently large such that with probability $1 - C\exp(-cn\epsilon_U^4/\max(\mathcal{I}_U, \epsilon_*^2))$,

$$|B_i - \mathbb{E}B_i| \leq C_2\epsilon_U^2 + \|f_i - f_{i_*}\|^2/4$$

for every $i \in [N]$, which is (43). This concludes the proof of Lemma 4. $\qquad\square$

**Lemma 5.** *The following holds:*

$$\max\{\epsilon_U^2, \mathcal{I}_U\} \asymp \max\{\epsilon_*^2, \mathcal{I}_U\}. \quad (46)$$

**Proof.** If $\mathcal{I}_U \lesssim \epsilon_*^2$, then $\epsilon_*^2 \asymp \epsilon_U^2$ by definition. If $\mathcal{I}_U \gtrsim \epsilon_*^2$, assume to the contrary $\mathcal{I}_U \ll \epsilon_U$; as we have $n\epsilon_U^4 \asymp \mathcal{I}_U \log \mathcal{N}(\epsilon_U, \mathcal{F}, \mathbb{P})$, this implies

$$\log \mathcal{N}(\epsilon_U, \mathcal{F}, \mathbb{P})/n \gg \epsilon_U^2.$$

But this implies, by definition of $\epsilon_*$, that $\epsilon_* \geq \epsilon_U$, contradicting the definition of $\epsilon_U$ (Definition 3). $\quad\square$

**Lemma 6.** *For a sufficiently large absolute constant $A > 0$, one has $\exp(-An\epsilon_U^4/\max(\epsilon_U^2, \mathcal{I}_U)) \leq n^{-1}$.*

**Proof.** First, we show that $N = \mathcal{N}(\mathcal{F}, \epsilon_U, \mathbb{P}) \gtrsim n^{1/4}/\log n$. Suppose to the contrary that $N \ll n^{1/4}/\log n$. Then

$$n\epsilon_U^4 \asymp \mathcal{I}_U \cdot \log N \lesssim \log n,$$

since $\mathcal{I}_U$ is at most the squared diameter of $\mathcal{F}_n$, which is $\Theta(1)$. This yields $\epsilon_U \lesssim (\log n/n)^{1/4}$. But $\mathcal{N}(\epsilon, \mathcal{F}, \mathbb{P}) \geq \frac{1}{\epsilon}$ because $\mathrm{diam}_{\mathbb{P}}(\mathcal{F}) = \Theta(1)$ and $\mathcal{F}_n$ is convex, so we obtain $\mathcal{N}(\epsilon_U, \mathcal{F}, \mathbb{P}) \gtrsim n^{1/4}/\log n$, contradiction.

To upper-bound $\exp(-An\epsilon_U^4/\max(\epsilon_U^2, \mathcal{I}_U))$, we split into cases. If $\mathcal{I}_U \lesssim \epsilon_U^2$ then $\epsilon_U \asymp \epsilon_*$ and we have $n\epsilon_*^2 \gtrsim \log n$ by Lemma 3, so $\exp(-An\epsilon_U^4/\max\{\epsilon_U^2, \mathcal{I}_U\}) \le \exp(-An\epsilon_*^2) \le n^{-1}$ for sufficiently large $A > 0$.

Otherwise, if $\mathcal{I}_U \gg \epsilon_U^2$ we have $n\epsilon_U^4/\mathcal{I}_U \gtrsim \log N$ by the definition of $\epsilon_U$, and since $N \gtrsim n^{1/5}$, we have $\log N \gg \log n$. Hence, by choosing $A > 0$ large enough we can ensure that $\exp(-An\epsilon_U^4/\mathcal{I}_U) \le n^{-1}$ in this case as well. $\qquad\square$

### 6.5  Proof of Theorem 4

Assume for simplicity that $c_L = 1$. We say $\boldsymbol{\xi}$ has a Gaussian Isoperimetric Profile (GIP) with respect to $\|\cdot\|_n$, if for any measurable set $A \subset \mathbb{R}^n$ such that $\Pr_{\boldsymbol{\xi}}(A) \ge 1/2$, we have that

$$\Pr_{\boldsymbol{\xi}}(A_t) \ge 1 - 2\exp(-nt^2/2). \tag{47}$$

where $A_t = \{\boldsymbol{\xi} \in \mathbb{R}^n : \inf_{x \in A} \|x - \boldsymbol{\xi}\|_n \le t\}$. It is not hard to verify that the GIP and LCP are equivalent (cf. (Artstein-Avidan et al., 2015, Thm 3.1.30)).

The main observation is the following simple and useful lemma which leverages the power of isoperimetry:

**Lemma 7.** *For any measurable $A \subset \mathbb{R}^n$ such that $\Pr_{\boldsymbol{\xi}}(A) \ge 2\exp(-nt^2/2)$, $\Pr_{\boldsymbol{\xi}}(A_{2t}) \ge 1 - 2\exp(-nt^2/2)$.*

**Proof.** Since $A_{2t} = (A_t)_t$, (47) implies that it's sufficient to show that $\Pr(A_t) \ge 1/2$, and indeed it suffices to show that $\Pr(A_{t+\epsilon}) \ge 1/2$ for any $\epsilon > 0$. Fix $\epsilon > 0$, and assume to the contrary that $\Pr(B) > 1/2$, where $B = \mathbb{R}^n \backslash A_{t+\epsilon}$. It's easy to see that $A \subset \mathbb{R}^n \backslash B_{t+\epsilon}$. Hence, using (47), we obtain

$$\Pr(\mathbb{R}^n \backslash A) \ge \Pr(B_{t+\epsilon}) \ge 1 - 2\exp(-n(t+\epsilon)^2/2),$$

i.e., $\Pr(A) \le 2\exp(-n(t+\epsilon)^2/2) < 2\exp(-nt^2/2)$, contradiction. $\qquad\square$

Denote the event of Definition 5 by $\mathcal{E}$, and recall the definition of $\epsilon_*^2$ via $n\epsilon_*^2 \asymp \log \mathcal{N}(\epsilon_*, \mathcal{F}, \mathbb{P})$. Letting $S$ be an $\epsilon_*$-net of $\mathcal{F}$ of cardinality $\mathcal{N}(\epsilon_*, \mathcal{F}, \mathbb{P})$, the pigeonhole principle implies the existence of $f_c \in S$ such that

$$\Pr_{\boldsymbol{\xi}}(\underbrace{\{\widehat{f}_n \in B(f_c, \epsilon_*)\} \cap \mathcal{E}}_{A}|\mathbf{X}) \ge \frac{\Pr_{\boldsymbol{\xi}}(\mathcal{E})}{\mathcal{N}(\epsilon_*, \mathcal{F}, \mathbb{P})} \ge \frac{\exp(-c_2 n\epsilon_*^2)}{\mathcal{N}(\epsilon_*, \mathcal{F}, \mathbb{P})} \ge \exp(-c_3 n\epsilon_*^2).$$

By isoperimetry, $\Pr_{\boldsymbol{\xi}}(A_{2t}) \ge 1 - 2\exp(-nt^2/2)$, where $t = M\epsilon_*/2$ and $M$ is chosen such that $(M/2)^2 \ge 2C_3$; this fixes the value of the absolute constant $M$ used in (13).

Applying (13) yields that if $\boldsymbol{\xi} \in A \subset \mathcal{E}$ and $\|\boldsymbol{\xi}' - \boldsymbol{\xi}\| \le M\epsilon_* = 2t$, $\|\widehat{f}_n(\boldsymbol{\xi}) - \widehat{f}_n(\boldsymbol{\xi}')\|^2 \le \rho_S(\mathbf{X}, f^*)$ and so $\|\widehat{f}_n(\boldsymbol{\xi}') - f_c\| \le \epsilon_* + \sqrt{\rho_S(\mathbf{X}, f^*)}$. This implies

$$\Pr_{\boldsymbol{\xi}}(\{\widehat{f}_n \in B(f_c, \epsilon_* + \sqrt{\rho_S(\mathbf{X}, f^*)})\}|\mathbf{X}) \ge \Pr_{\boldsymbol{\xi}}(A_{2t}|\mathbf{X}) \ge 1 - 2\exp(-nt^2/2).$$

To bound the variance of $\widehat{f}_n$, we use conditional expectation as in Theorem 3. We have

$$V(\widehat{f}_n|\mathbf{X}) \le \mathbb{E}\|\widehat{f}_n - f_c\|^2 \le (1 - 2\exp(-nt^2/2)) \cdot (\epsilon_* + \sqrt{\rho_S(\mathbf{X}, f^*)})^2 + 2C\exp(-nt^2/2),$$

where we have used the fact that $\mathrm{diam}_{\mathbb{P}}(\mathcal{F}) = \Theta(1)$. Recalling that $\epsilon_*^2 \gtrsim \log(n)/n$, and $t = \Theta(\epsilon_*)$, we have $\exp(-nt^2) = O(\epsilon_*^2)$ and hence the RHS is bounded by $C\max\{\epsilon_*^2, \rho_S(\mathbf{X}, f^*)\}$, as desired.

### 6.6  Proof of Theorem 5

For simplicity, we assume that $c_X = c_L = c_I = O(1)$, We abbreviate $\rho_{\mathcal{O}} := \rho_{\mathcal{O}}(n, \mathbb{P}, f^*)$.

We shall use the joint metric on $\mathcal{X}^n \times \mathbb{R}^n$ given by

$$\Delta_n((\mathbf{X}_1, \boldsymbol{\xi}_1), (\mathbf{X}_2, \boldsymbol{\xi}_2)) := n^{-1/2} \cdot d_n(\mathbf{X}_1, \mathbf{X}_2) + \|\boldsymbol{\xi}_1 - \boldsymbol{\xi}_2\|_n.$$

As $(\mathbf{X}, d_n)$ and $(\boldsymbol{\xi}, \|\cdot\|_2)$ both satisfy Lipschitz concentration inequalities with parameter $\Theta(1)$, so does the product space $\mathcal{X}^n \times \mathbb{R}^n$ with the usual product metric

$$((\mathbf{X}_1, \boldsymbol{\xi}_1), (\mathbf{X}_2, \boldsymbol{\xi}_2)) \mapsto d_n(\mathbf{X}_1, \mathbf{X}_2) + \|\boldsymbol{\xi}_1 - \boldsymbol{\xi}_2\|_2,$$

and since $\Delta_n$ is obtained by scaling this metric by $n^{-1/2}$, we obtain that $(\mathcal{X}^n \times \mathbb{R}^n, \Delta)$ satisfies an LCP condition with parameter $\Theta(n)$.

Let $\mathcal{E}_1$ be the event of Assumption 7, namely, the event that the ERM is almost interpolating, and let $\mathcal{E}_2$ be the event that $\operatorname{diam}_{\mathbb{P}}(\mathcal{O}_{M'\epsilon_*^2}) \leq \rho_{\mathcal{O}}$. Since $\Pr(\mathcal{E}_1) + \Pr(\mathcal{E}_2) > 1 + \exp(-c_I n \epsilon_*^2)$, we have $\Pr(\mathcal{E}_1 \cap \mathcal{E}_2) > \exp(-c_I n \epsilon_*^2)$.

Set $\mathcal{E}_3 = \mathcal{E}_1 \cap \mathcal{E}_2$. Since $\Pr(\mathcal{E}_3) \geq \exp(-c_I n \epsilon_*^2)$, the same pigeonhole principle argument used in the proofs of Theorems 1 and 4 shows that there exists an absolute constant $c_1 \in (0, c_I)$ and $f_c \in \mathcal{F}$ such that

$$\Pr_{\mathbf{X}, \boldsymbol{\xi}}(\mathcal{E}_3 \cap \{\widehat{f}_n \in B(f_c, \epsilon_*)\}) \geq \exp(-c_1 n \epsilon_*^2).$$

Denote this event by $\mathcal{E}$ (in this case, it is better to think about it as a subset of $\mathcal{X}^n \times \mathbb{R}^n$). By the same argument as in Theorem 4, $\tilde{\mathcal{E}} := \mathcal{E}_{C_1 \epsilon_*}$ will be an event of probability $1 - \exp(-c_1 n \epsilon_*^2)$, where $A_r = \{(\mathbf{X}, \boldsymbol{\xi}) \in \mathcal{X}^n \times \mathbb{R}^n : \Delta_n((\mathbf{X}, \boldsymbol{\xi}), A) \leq r\}$ as above, and $C_1$ is an absolute constant depending on $c_1$ and the LCP parameter of $\Delta$.

Thus, we would like to show that any $(\mathbf{X}, \boldsymbol{\xi}) \in \tilde{\mathcal{E}}$ is not too far from $f_c$. More precisely, we claim that for any $(\mathbf{X}, \boldsymbol{\xi})$ at distance at most $C_1 \epsilon_*$ from $\mathcal{E}$, the corresponding $\widehat{f}_n$ is at distance at most $C_2 \cdot \max\{\epsilon_*, \sqrt{\rho_{\mathcal{O}}}\}$ from $f_c$.

For $f \in \mathcal{F}$, let $f_{\mathbf{X}}$ denote $(f(X_1), \ldots, f(X_n)) \in \mathbb{R}^n$. We claim that it suffices to prove the following: for every $(\mathbf{X}_1, \boldsymbol{\xi}_1) \in \mathcal{E}$ and such that $\Delta_n((\mathbf{X}_2, \boldsymbol{\xi}_2), (\mathbf{X}_1, \boldsymbol{\xi}_1)) \leq C_1 \epsilon_*$, we have

$$\|\widehat{f}_n(\mathbf{X}_2, \boldsymbol{\xi}_2)_{\mathbf{X}_1} - \widehat{f}_n(\mathbf{X}_1, \boldsymbol{\xi}_1)_{\mathbf{X}_1}\|_n \lesssim \epsilon_*, \tag{48}$$

where $\widehat{f}_n(\mathbf{X}_i, \boldsymbol{\xi}_i)$ is the ERM for the input points $\mathbf{X}_i$ and noise $\boldsymbol{\xi}_i$,. Indeed, assuming (48) we have

$$\|\widehat{f}_n(\mathbf{X}_2, \boldsymbol{\xi}_2)_{\mathbf{X}_1} - \mathbf{Y}_1\|_n^2 \leq 2\|\widehat{f}_n(\mathbf{X}_1, \boldsymbol{\xi}_1)_{\mathbf{X}_1} - \mathbf{Y}_1\|_n^2 + 2\|\widehat{f}_n(\mathbf{X}_2, \boldsymbol{\xi}_2)_{\mathbf{X}_1} - \widehat{f}_n(\mathbf{X}_1, \boldsymbol{\xi}_1)_{\mathbf{X}_1}\|_n^2 \lesssim \epsilon_*^2, \tag{49}$$

as the first term on the RHS is bounded by $2\epsilon_*$ because $(\mathbf{X}_1, \boldsymbol{\xi}_1) \in \mathcal{E}$, and the second term is bounded by $4\epsilon_*^2$ by construction. We now specify the constant $M'$ in the definition of $\rho_{\mathcal{O}}$ (Definition 6) to be any upper bound for the implicit absolute constant in (49). Under this definition, (49) implies that $\widehat{f}_n(\mathbf{X}_2, \boldsymbol{\xi}_2)_{\mathbf{X}_1} \in \mathcal{O}_{M'\epsilon_*^2}$ and hence $\|\widehat{f}_n(\mathbf{X}_2, \boldsymbol{\xi}_2) - \widehat{f}_n(\mathbf{X}_1, \boldsymbol{\xi}_1)\|^2 \leq \rho_{\mathcal{O}}$. Since $\widehat{f}_n(\mathbf{X}_1, \boldsymbol{\xi}_1) \in \mathcal{E}$, this implies that

$$\|\widehat{f}_n(\mathbf{X}_2, \boldsymbol{\xi}_2) - f_c\|^2 \leq 2(\|\widehat{f}_n(\mathbf{X}_2, \boldsymbol{\xi}_2) - \widehat{f}_n(\mathbf{X}_1, \boldsymbol{\xi}_1)\|^2 + \|\widehat{f}_n(\mathbf{X}_2, \boldsymbol{\xi}_2) - f_c\|^2 \lesssim \max\{\epsilon_*^2, \rho_{\mathcal{O}}\}$$

as desired.

Thus, on the high-probability event $\tilde{\mathcal{E}}$, $\widehat{f}_n \in B(f_c, C \max\{\epsilon_*^2, \rho_O\})$. As in the proof of Theorem 3, one concludes by conditional expectation that $V(\widehat{f}_n) \leq C \max\{\epsilon_*^2, \rho_O\}$.

**Proof of** (48): For convenience, denote $f_{i,j} = \widehat{f}_n(\mathbf{X}_i, \boldsymbol{\xi}_i)_{\mathbf{X}_j}$, and similarly $f_j^* = (f^*)_{\mathbf{X}_j}$. As $d((\mathbf{X}_2, \boldsymbol{\xi}_2), (\mathbf{X}_1, \boldsymbol{\xi}_1)) \leq 2\epsilon_*$, we have by the Lipschitz property that $\|f_{i,1} - f_{i,2}\|_n \leq 2\epsilon_*$ and also $\|f_1^* - f_2^*\|_n \leq 2\epsilon_*$. In addition, letting $\mathbf{Y}_i = f_i^* + \boldsymbol{\xi}_i$ be the observation vector, the Lipschitz property of $f^*$ and the bound on $\|\boldsymbol{\xi}_1 - \boldsymbol{\xi}_2\|_n$ together imply that $\|\mathbf{Y}_1 - \mathbf{Y}_2\|_n \leq 4\epsilon_*$. The definition of $f_{i,i}$ as the ERM with data points $\mathbf{X}_i$ and observations $\mathbf{Y}_i$ implies that for $i = 1, 2$,

$$\|f_{i,i} - \mathbf{Y}_i\|_n \leq \|f_{\mathbf{X}_i} - \mathbf{Y}_i\|_n \tag{50}$$

for any $f \in \mathcal{F}$. Finally, the almost interpolating assumption (Assumption 7) yields $\|\mathbf{Y}_1 - f_{1,1}\|_n \leq C\epsilon_*$.

We obtain (48) by putting these bounds all together. Indeed, we have

$$\|f_{1,1} - f_{2,1}\|_n \leq \|f_{1,1} - \mathbf{Y}_1\|_n + \|\mathbf{Y}_1 - \mathbf{Y}_2\|_n + \|\mathbf{Y}_2 - f_{2,2}\|_n + \|f_{2,2} - f_{2,1}\|_n$$
$$\leq (C + 6)\epsilon_* + \|f_{2,2} - \mathbf{Y}_2\|_n,$$

and substituting $i = 2$, $f = f_1$ into (50) yields

$$\|f_{2,2} - \mathbf{Y}_2\|_n \le \|f_{1,2} - \mathbf{Y}_2\|_n$$
$$\le \|f_{1,2} - f_{1,1}\|_n + \|f_{1,1} - \mathbf{Y}_1\|_n + \|\mathbf{Y}_1 + \mathbf{Y}_2\|_n$$
$$\le (C + 6)\epsilon_*,$$

so we finally obtain

$$\|f_{1,1} - f_{2,1}\|_n \le 2(C + 6)\epsilon_* \lesssim \epsilon_*,$$

as desired.

## 6.7 Proof of Theorem 6

The proof strategy is identical to that of Corollary 2: use a fixed-point theorem to find a function $f^*$ for which $f^* = \mathbb{E}_{\mathbf{X}, \boldsymbol{\xi}} \widehat{f}_n$, for which we have $\mathbb{E}\|\widehat{f}_n - f^*\|^2 \le \sup_{f^* \in \mathcal{F}} V(\widehat{f}_n)$. However, the infinite-dimensional random-design setting makes things a bit trickier.

For given $f^*, \mathbf{X}, \boldsymbol{\xi}$, let $F_{\mathbf{X}, \boldsymbol{\xi}}(f^*)$ denote the corresponding ERM (which we have previously denoted $\widehat{f}_n$). Recall that while the ERM is uniquely defined as a vector in $\mathcal{F}_n$, its lift to $\mathcal{F}$ is in general far from unique. We will make two temporary assumptions to streamline the proof, and explain at the end of the proof how to remove them, at the cost of some additional technical complexity. First, we assume that $F_{\mathbf{X}, \boldsymbol{\xi}}(f^*)$ is the (unique) element of $\mathcal{F}$ of minimal $L^2(\mathbb{P})$-norm mapping to the finite-dimensional ERM; second, we assume that for each $\mathbf{X}$, the minimal-norm lifting map, defined by

$$L_{\mathbf{X}}(v) = \operatorname{argmin}\{\|f\| : f \in \mathcal{F} \,|\, v = (f(x_1), \dots, f(x_n))\},$$

is continuous.

The map $F_{\mathbf{X}, \boldsymbol{\xi}}$ is the composition of the following maps:

$$\mathcal{F} \xrightarrow{P_n} \mathcal{F}_n \xrightarrow{v \mapsto P_{\mathcal{F}_n}(v + \boldsymbol{\xi})} \mathcal{F}_n \xrightarrow{L_{\mathbf{X}}} \mathcal{F}$$

where $P_n(f) = (f(x_1), \dots, f(x_n))$, $P_{\mathcal{F}_n}$ is the projection from $L^2(\mathbb{P}^{(n)})$ onto the convex set $\mathcal{F}_n$, which is the LSE in fixed design, and $L_{\mathbf{X}}$ is the lifting map defined above. The linear map $P_n$ is continuous by Assumption 8, and the map $v \mapsto P_{\mathcal{F}_n}(v + \boldsymbol{\xi})$ is continuous because projection onto a convex set is continuous. As we have assumed (for now) that $L_{\mathbf{X}}$ is continuous, this proves that for every $\mathbf{X}, \boldsymbol{\xi}$, $f \mapsto F_{\mathbf{X}, \boldsymbol{\xi}}(f)$ is a continuous map of the compact set $\mathcal{F}$ to itself.

We claim that the expectation of this map, $f \mapsto \mathbb{E}_{\mathbf{X}, \boldsymbol{\xi}}[F_{\mathbf{X}, \boldsymbol{\xi}}(f)]$, is also continuous: indeed, if $f_k \to f$ then

$$\|F_{\mathbf{X}, \boldsymbol{\xi}}(f_k) - F_{\mathbf{X}, \boldsymbol{\xi}}(f)\| \to 0$$

for each $\mathbf{X}, \boldsymbol{\xi}$ and is bounded by the diameter $D_{\mathbb{P}}(\mathcal{F})$, so Jensen's inequality and dominated convergence imply

$$\|\mathbb{E}[F_{\mathbf{X}, \boldsymbol{\xi}}(f_k)] - \mathbb{E}[F_{\mathbf{X}, \boldsymbol{\xi}}(f)]\| \le \mathbb{E}[\|F_{\mathbf{X}, \boldsymbol{\xi}}(f_k) - F_{\mathbf{X}, \boldsymbol{\xi}}(f)\|] \to 0$$

which is continuity.

We can thus apply the Schauder fixed point theorem (Aliprantis and Border, 2006, Theorem 17.56):

**Theorem 8.** *Let $K$ be a nonempty compact convex subset of a Banach space, and let $f : K \to K$ be a continuous function. Then the set of fixed points of $f$ is compact and nonempty.*

The fixed point we obtain is a function $f^* \in \mathcal{F}$ for which $f^* = \mathbb{E}[F_{\mathbf{X}, \boldsymbol{\xi}}(f)] = \mathbb{E}\widehat{f}_n$ and hence,

$$\mathbb{E}\|\widehat{f}_n - f^*\|^2 = \mathbb{E}\|\widehat{f}_n - \mathbb{E}\widehat{f}_n\|^2 \lesssim \mathcal{V}(\widehat{f}_n, \mathcal{F}, \mathbb{P}).$$

This concludes the proof in the case that the lifting maps $L_{\mathbf{X}}$ are continuous.

Unfortunately, the assumption that the $L_{\mathbf{X}}$ are continuous turns out to be unjustified in general. Indeed, it is not difficult to construct an example of a convex set $K \subset \mathbb{R}^3$ for which the minimal-norm

lift $P_{\mathbb{R}^2}(K) \to K$ is not continuous; in fact, one can construct $K \subset \mathbb{R}^3$ with no continuous section $P_{\mathbb{R}^2}(K) \to K$. So we need to explain how to proceed without this assumption.

Fortunately, each $L_{\mathbf{X}}$ is always continuous on the *relative interior* of $\mathcal{F}_n$ (we sketch the proof of this at the end of the section), so the following modification of $F_{\mathbf{X},\boldsymbol{\xi}}$ does turn out to be continuous:

$$\mathcal{F} \xrightarrow{P_n} \mathcal{F}_n \xrightarrow{v \mapsto P_{\mathcal{F}_n}(v+\boldsymbol{\xi})} \mathcal{F}_n \xrightarrow{\varphi_\delta} \mathcal{F}_n \xrightarrow{L_{\mathbf{X}}} \mathcal{F}, \tag{51}$$

where

$$\varphi_\delta(v) = (1-\delta)(v-v_0) + v_0$$

is simply a contraction of $\mathcal{F}_n$ into a $(1-\delta)$-scale copy of itself ($v_0$ is some arbitrarily chosen point in the interior of $\mathcal{F}_n$).

Let $\tilde{F}_{\mathbf{X},\boldsymbol{\xi}}$ denote the composition of the maps in (51). By the argument above, $\tilde{F}_{\mathbf{X},\boldsymbol{\xi}}$ is continuous and $\mathbb{E}[\tilde{F}_{\mathbf{X},\boldsymbol{\xi}}]$ has a fixed point $f^*$.

Of course, $f^*$ is not a fixed point of $\mathbb{E}[F_{\mathbf{X},\boldsymbol{\xi}}]$ as we would like. However, note that $\|\varphi_\delta(v) - v\|_n \le 2\delta$ for any $v \in \mathcal{F}_n$ (as the diameter of $\mathcal{F}_n$ is at most 2). Hence, we have for any $v \in \mathcal{F}_n$ that

$$\|L_{\mathbf{X}}(\varphi_\delta(v)) - L_{\mathbf{X}}(v)\|^2 \le 2\|\varphi_\delta(v) - v\|_n^2 + C\mathcal{I}_L(n) \le 8\delta^2 + \mathcal{I}_L(n)$$

on an event $\mathcal{E}$ of high probability; in particular this holds for $v = P_{\mathcal{F}_n}(P_n(f^*) + \boldsymbol{\xi})$, which means that on $\mathcal{E}$,

$$\|\tilde{F}_{\mathbf{X},\boldsymbol{\xi}}(f^*) - F_{\mathbf{X},\boldsymbol{\xi}}(f^*)\| \le 8\delta^2 + C\mathcal{I}_L(n).$$

Choosing $\delta \lesssim \mathcal{I}_L(n)$ and applying conditional expectation (using the fact that $\mathcal{E}^c$ is negligible) and Jensen's inequality, we get that the $f^*$ thus obtained satisfies $\|f^* - \mathbb{E}\widehat{f}_n\| \lesssim \max\{\sup_{f^* \in \mathcal{F}} V(\widehat{f}_n), \mathcal{I}_L(n)\}$, which shows that the ERM is admissible for this $f^*$.

By the same argument, we may discard the assumption that the ERM is computed by finding the element of $\mathcal{F}$ of minimal norm mapping to the finite-dimensional ERM $\widehat{f}_n^{(fd)}$: indeed, under the event $\mathcal{E}$, the set of functions in $\mathcal{F}$ mapping to $\widehat{f}_n^{(fd)}$ has diameter $C \cdot \mathcal{I}_L(n)$, so changing the selection rule for the ERM will shift its expectation by a perturbation of norm at most $C \cdot \mathcal{I}_L(n)$.

It remains to explain why the lifting map $L = L_{\mathbf{X}} : \mathcal{F}_n \to \mathcal{F}$ is continuous on the relative interior of $\mathcal{F}_n$. Replacing the ambient space with the affine hull of $\mathcal{F}_n$, we may assume $\mathcal{F}_n$ has nonempty interior.

Suppose $v_k \to v$ in $\mathcal{F}_n$ and $v \in \text{int}\,\mathcal{F}_n$; we wish to show that $L(v_k) \to f = L(v)$. As $\mathcal{F}_n$ is compact, by passing to a subsequence we may assume $L(v_k)$ converges to some $g \in \mathcal{F}$. Since $P_n$ is continuous, we have $v = P_n(L(v_k)) \to P_n(g)$, i.e., $g$ is a lift of $v$. Hence, by definition, $\|g\| \ge \|f\|$, and we wish to show that equality holds.

Suppose not. Then $\|g\| > \|f\|$ and hence $\|L(v_k)\| \ge \|f\| + \epsilon$ for all $k$ and some $\epsilon > 0$; that is, there exist $v_k$ arbitrarily close to $v$ whose minimal-norm lift has much larger norm than that of $v$. It suffices to show this is impossible (i.e., that $u \mapsto \|L(u)\|$ is upper semicontinuous at $v$). This follows from the fact that $u \mapsto \|L(u)\|$ is convex, as is easily verified, and a convex function is continuous on the interior of its domain (Schneider, 2014, Theorem 1.5.3); for completeness, we give a direct proof.

Since $v \in \text{int}\,\mathcal{F}_n$, there exists $r > 0$ such that $B(v,r) \subset \mathcal{F}_n$. This implies that for any $\delta > 0$, one has

$$B(v,\delta) \subset v + \frac{\delta}{r}(\mathcal{F}_n - v).$$

Let $D$ be the diameter of $\mathcal{F}$ in $L^2(\mathbb{P})$. We have $\mathcal{F} \subset B(f,D)$ and hence $\mathcal{F}_n \subset P_n(B(f,D) \cap \mathcal{F})$. By linearity, this implies that

$$v + a(\mathcal{F}_n - v) \subset P_n(B(f, aD) \cap \mathcal{F})$$

for any $a > 0$; choosing $a = \frac{\delta}{r}$ we obtain

$$B(v,\delta) \cap \mathcal{F}_n \subset P_n\left(B\left(f, \frac{D\delta}{r}\right) \cap \mathcal{F}\right).$$

In other words, if $\|u - v\|_n < \delta$ and $u \in \mathcal{F}_n$, there exists an element of $\mathcal{F}$ in $B(f, \frac{D\delta}{r})$ mapping to $u$, which in particular implies that $\|L(u)\| \leq \|f\| + \frac{D\delta}{r}$. This means that $u \mapsto \|L(u)\|$ is upper semicontinuous at $v$, which was precisely what we needed in order to conclude that $L$ is continuous at $v$.

## 6.8 Proof of Theorem 7

**Preliminaries** The following classical and standard results appear for example in Vershynin (2018).

**Lemma 8.** *[Maximal inequality] Let $Z_1, \ldots, Z_k$ be zero mean $\sigma$-sub-Gaussian random variables with bounded variance. Then, we have that*

$$\mathbb{E} \max_{1 \leq i \leq k} Z_i \lesssim \sigma \sqrt{\log k}.$$

**Lemma 9** (Dudley's lemma). *The following holds for all $\epsilon \in (0, 1)$:*

$$\mathbb{E} \sup_{f_i \in \mathcal{N}_\epsilon} G_{f_i - f^*} \leq \frac{C_4}{\sqrt{n}} \int_\epsilon^{\mathrm{Diam}_{\mathbb{P}^{(n)}}(\mathcal{F})} \sqrt{\log \mathcal{N}(u, \mathcal{F}, \mathbb{P}^{(n)})} du, \tag{52}$$

*where $\mathcal{N}_\epsilon$ denotes the minimal $\epsilon$-net of $\mathcal{F}$ in terms of $L_2(\mathbb{P}^{(n)})$.*

**Lemma 10** (Sudakov's minoration lemma). *The following holds for all $\epsilon \in (0, 1)$:*

$$\mathbb{E}_{\boldsymbol{\xi}} \sup_{f \in \mathcal{F}} G_f \gtrsim \sup_{\epsilon \geq 0} \epsilon \sqrt{\frac{\log \mathcal{N}(\epsilon, \mathcal{F}, \mathbb{P}^{(n)})}{n}}. \tag{53}$$

**Proof of Theorem 7.** Throughout this proof, we fix a realization in $\mathbb{P}^{(n)} = \mathbb{P}_n$ that satisfies $\mathcal{I}_L(n) = o(\epsilon_*^2)$ and $\mathcal{I}_U(n) = O(\epsilon_*^2)$. For such $\mathbb{P}^{(n)}$, note that

$$\log \mathcal{N}(\epsilon, \mathcal{F}, \mathbb{P}) \asymp \log \mathcal{N}(\epsilon, \mathcal{F}, \mathbb{P}^{(n)}) \quad \forall \epsilon \in (\epsilon_*, \Gamma),$$

we will use the last equation in various places in this proof.

We start by finding a weakly admissible $f^*$ by a more constructive method than that used in the proof of Corollary 3 (the method here is closer to the original proof of Chatterjee (2014)). Then, we prove (19) for this choice of $f^*$.

Let $\mathcal{N} := \{f_1, \ldots, f_{\mathcal{N}(\epsilon, \mathcal{F}, \mathbb{P}^{(n)})}\}$ be a minimal $\epsilon_* := \epsilon_*(n)$-net of $\mathcal{F}$ in terms of $L_2(\mathbb{P}^{(n)})$, and denote $B(f_i) := B_n(f_i, \epsilon_*)$, $i = 1, \ldots, |\mathcal{N}|$. Our weakly-admissible $f^* \in \mathcal{F}$ is defined as

$$f^* := \operatorname*{argmax}_{f_i \in \mathcal{N}} \mathbb{E} \sup_{f \in B(f_i)} G_{f - f_i}. \tag{54}$$

**Lemma 11.** *The following event holds with probability (over $\boldsymbol{\xi}$) of at least $1 - 2\exp(-cn\epsilon_*^2)$*

$$\forall i \in [|\mathcal{N}|] \quad \left| \sup_{f \in B(f_i)} G_{f - f_i} - \mathbb{E} \sup_{f \in B(f_i)} G_{f - f_i} \right| \leq C_1 \epsilon_*^2. \tag{55}$$

*In addition, for a fixed $j \in 1, \ldots, \lceil \epsilon_*^{-1} \rceil$, the event*

$$\sup_{f_i \in \mathcal{N} \cap B(f^*, j\epsilon_*)} G_{f_i - f^*} \leq C_2 j \epsilon_*^2. \tag{56}$$

*holds with probability of at least $1 - 2\exp(-c_3 n \epsilon_*^2)$.*

The proof of this lemma appears below. We denote the event of (55) by $\mathcal{E}$, and by $\mathcal{E}(j)$ the event of (56). Following Chatterjee (2014), we define

$$\Psi_{\boldsymbol{\xi}}(t) = \sup_{f \in B_n(f^*, t)} 2 G_{f - f^*} - t^2.$$

One easily verifies (see (Chatterjee, 2014, Proof of Theorem 1.1)) that $\Psi_{\boldsymbol{\xi}}(t)$ is strictly concave and that $\operatorname*{argmax}_{t \geq 0} \Psi_{\boldsymbol{\xi}}(t) = \|f^* - \widehat{f}_n\|_n^2$.

This implies that if, for any particular $\boldsymbol{\xi}$, we identify $t_1, t_2$ such that $\Psi_{\boldsymbol{\xi}}(t_1) > \Psi_{\boldsymbol{\xi}}(t_2)$, the unique maximum of $\Psi_{\boldsymbol{\xi}}$ occurs for some $t$ smaller than $t_2$, i.e., $\|f^* - \widehat{f}_n\|_n \le t_2$. We will take $t_1 = \epsilon_*$ and $t_2 = D\epsilon_*$ for a sufficiently large constant $D$ and show that $\Psi_{\boldsymbol{\xi}}(t_1) > \Psi_{\boldsymbol{\xi}}(t_2)$ on $\mathcal{E} \cap \mathcal{E}(D)$. This implies that $\|f^* - \widehat{f}_n\|_n \le D\epsilon_*$ on $\mathcal{E} \cap \mathcal{E}(D)$, which precisely means that $\widehat{f}_n$ is admissible for $f^*$.

On the one hand, conditioned on $\mathcal{E}$ we have

$$
\begin{aligned}
\Psi_{\boldsymbol{\xi}}(\epsilon_*) &= \sup_{f \in B(f^*)} 2G_{f-f^*} - \epsilon_*^2 \\
&\ge \max_{f_i \in \mathcal{N}} \mathbb{E} \sup_{f \in B(f_i)} 2G_{f-f^*} - C_1\epsilon_*^2,
\end{aligned}
\tag{57}
$$

where we used the definition of $f^*$ and Eq. (55) above. On the other hand, for $D \ge 2$ we have under $\mathcal{E}(D) \cap \mathcal{E}$ that

$$
\begin{aligned}
\sup_{f \in B_n(f^*, D\epsilon_*)} G_{f-f^*} &\le \max_{f_i \in \mathcal{N} \cap B(f^*, D\epsilon_*)} \sup_{f \in B(f_i)} G_{f-f_i} + \sup_{f_i \in \mathcal{N} \cap B(f^*, D\epsilon_*)} G_{f_i-f^*} \\
&\le \sup_{f \in B(f^*, \epsilon_*)} G_{f-f^*} + C_2 D\epsilon_*^2,
\end{aligned}
$$

where we used the definition of $f^*$ and (56). Substituting in the definition of $\Psi$, we obtain

$$
\begin{aligned}
\Psi_{\boldsymbol{\xi}}(D\epsilon_*) &= \sup_{f \in B_n(f^*, D\epsilon_*)} 2G_{f-f^*} - D^2\epsilon_*^2 \\
&\le 2\left(\sup_{f \in B(f^*)} G_{f-f^*} + C_2 D\epsilon_*^2\right) - D^2\epsilon_*^2 \\
&\le \Psi_{\boldsymbol{\xi}}(\epsilon_*) + (2C_2 + 1)D\epsilon_*^2 - D^2\epsilon_*^2.
\end{aligned}
$$

Comparing with (57) we see that for $D \ge 2C_2 + C_1 + 1$ (say) we have $\Psi_{\boldsymbol{\xi}}(D\epsilon_*) < \Psi_{\boldsymbol{\xi}}(\epsilon_*)$ on $\mathcal{E} \cap \mathcal{E}(D)$. Since $\mathcal{I}_L(n) = o(\epsilon_*^2)$, we obtain that

$$
\|\widehat{f}_n - f^*\|^2 \le 4\|\widehat{f}_n - f^*\|_n^2 + o(\epsilon_*^2) \lesssim \epsilon_*^2,
$$

where we used that, and therefore weakly admissible in $L_2(\mathbb{P})$.

Now, we are ready to prove (19). First, we apply Sudakov's inequality, and note that

$$
\mathbb{E}_{\boldsymbol{\xi}} \sup_{f \in \mathcal{F}} G_{f-f^*} \gtrsim \sup_{\epsilon \ge 0} \epsilon \sqrt{\frac{\log \mathcal{N}(\epsilon, \mathcal{F}, \mathbb{P}^{(n)})}{n}} \gtrsim \sup_{\epsilon \gtrsim \sqrt{\mathcal{I}_L(n)}} \epsilon \sqrt{\frac{\log \mathcal{N}(\epsilon, \mathcal{F}, \mathbb{P})}{n}} \gg \epsilon_*^2, \tag{58}
$$

where we used that $\mathcal{I}_L(n) = o(\epsilon_*^2)$ and that $\frac{\epsilon^2}{\log(1/\epsilon)} \cdot \log \mathcal{N}(\epsilon, \mathcal{F}, \mathbb{P})$ is decreasing in $\epsilon \in (0, \Gamma)$. We first claim that with probability $1 - 2\exp(-cn\epsilon_*^2)$,

$$
\sup_{f \in B(f^*) - f^*} G_f = \omega(\epsilon_*^2), \tag{59}
$$

where $f^*$ is our admissible function. To see this, by Lemma 9 and (54)

$$
\begin{aligned}
\omega(\epsilon_*^2) = \mathbb{E} \sup_{f \in \mathcal{F}} G_{f-f^*} &\le \max_{f_i \in \mathcal{N} \cap B_n(f^*, \mathbb{E}_{f^*})} \mathbb{E} \sup_{f \in B(f_i)} G_{f-f_i} + \mathbb{E} \max_{f_i \in \mathcal{N}} G_{f_i-f^*} \\
&\le \mathbb{E} \sup_{f \in B(f^*)} G_{f-f^*} + \frac{C_1}{\sqrt{n}} \int_{\epsilon_*}^{\Gamma} \sqrt{\log \mathcal{N}(t, \mathcal{F}, \mathbb{P})} dt \\
&\le \mathbb{E} \sup_{f \in B(f^*)} G_{f-f^*} + \frac{C_2}{\sqrt{n}} \int_{\epsilon_*}^{\Gamma} \sqrt{\log \mathcal{N}(t, \mathcal{F}, \mathbb{P}^{(n)})} dt \\
&\le \mathbb{E} \sup_{f \in B(f^*)} G_{f-f^*} + O(\epsilon_*^2),
\end{aligned}
$$

where we used (58) and Lemma 9 and that $\mathcal{I}_L(n) = o(\epsilon_*^2)$ and $\mathcal{I}_U(n) = \Theta(\epsilon_*^2)$. Hence,

$$
\mathbb{E} \sup_{f \in B(f^*)} G_{f-f^*} = \omega(\epsilon_*^2) - O(\epsilon_*^2) = \omega(\epsilon_*^2). \tag{60}
$$

This gives us a lower bound for $\sup_{f \in B(f^*)} G_{f-f^*}$ in expectation, and high-probability bound follows from the proof of Lemma 11 below; so we only sketch it: $\sup_{f \in B(f^*)} G_{f-f^*}$ is convex and $O(\epsilon_* n^{-1/2})$-Lipschitz, which means that it deviates from its expectation by $\epsilon_*^2$ with probability at most $2\exp(-cn\epsilon_*^2)$. Combining this with (60) proves (59).

Let $\mathcal{V}$ denote the set of noise vectors for which $\|\widehat{f}_n - f^*\|_n \leq C\epsilon_*$ and $\sup_{f \in B(f^*)} G_{f-f^*} = \omega(\epsilon_*^2)$; by what we have already proven, we have

$$\Pr(\boldsymbol{\xi} \in \mathcal{V}) \geq 1 - C\exp(-cn\epsilon_*^2).$$

Let $\mathcal{V}' = \mathcal{V} \cap (-\mathcal{V}) = \{\boldsymbol{\xi} : \boldsymbol{\xi}, -\boldsymbol{\xi} \in \mathcal{V}\}$. By the union bound,

$$\Pr(\boldsymbol{\xi} \in \mathcal{V}') \geq 1 - 2C\exp(-cn\epsilon_*^2).$$

Fix any $\boldsymbol{\xi} \in \mathcal{V}'$, and denote by $\widehat{f}_n^-$ the ERM with the flipped noise vector $-\boldsymbol{\xi}$:

$$\widehat{f}_n^- := \operatorname*{argmin}_{f \in \mathcal{F}} \left( \sum_i (-\xi_i + f^*(x_i) - f(x_i))^2 \right).$$

Since $\boldsymbol{\xi}, -\boldsymbol{\xi} \in \mathcal{V}' \subset \mathcal{V}$, we have

$$\|\widehat{f}_n - \widehat{f}_n^-\|_n \leq \|\widehat{f}_n - f^*\|_n + \|\widehat{f}_n^- - f^*\|_n \leq C\epsilon_*.$$

In other words, $\widehat{f}_n^- \in B_n(\widehat{f}_n, C\epsilon_*)$, so to prove (19), it thus suffices to show that $\widehat{f}_n^-$ is not a $\delta$-approximate minimizer for $\delta = \omega(\epsilon_*^2)$ with respect to the noise $\boldsymbol{\xi}$, i.e.,

$$\frac{1}{n}\sum_{i=1}^{n}(f^*(x_i) + \xi_i - \widehat{f}_n^-(x_i))^2 \geq \frac{1}{n}\sum_{i=1}^{n}(f^*(x_i) + \xi_i - \widehat{f}_n(x_i))^2 + \omega(\epsilon_*^2).$$

Equivalently, (by subtracting $\|\boldsymbol{\xi}\|_n^2$ from both sides as in (35)), we wish to prove that

$$-2G_{\widehat{f}_n^- - f^*} + \|\widehat{f}_n^- - f^*\|_n^2 \geq -2G_{\widehat{f}_n - f^*} + \|\widehat{f}_n - f^*\|_n^2 + \omega(\epsilon_*^2).$$

Since $\|\widehat{f}_n^- - f^*\|_n^2, \|\widehat{f}_n - f^*\|_n^2 = O(\epsilon_*^2)$ as $\boldsymbol{\xi}, -\boldsymbol{\xi} \in \mathcal{V}$, this reduces to showing that

$$-2G_{\widehat{f}_n^- - f^*} \geq -2G_{\widehat{f}_n - f^*} + \omega(\epsilon_*^2). \tag{61}$$

On the one hand, by using Eqs. (57) and (59) above it is easy to see that on $\mathcal{V}' \subset \mathcal{V}$, we have $G_{\widehat{f}_n - f^*} \geq \omega(\epsilon_*^2)$. On the other hand,

$$G_{\widehat{f}_n^- - f^*} = \frac{1}{n}\langle \widehat{f}_n^- - f^*, \boldsymbol{\xi} \rangle = -\frac{1}{n}\langle \widehat{f}_n^- - f^*, -\boldsymbol{\xi} \rangle. \tag{62}$$

But note that $\frac{1}{n}\langle \widehat{f}_n^- - f^*, -\boldsymbol{\xi} \rangle$ is the process for the noise vector $-\boldsymbol{\xi}$ evaluated at the corresponding ERM, namely $\widehat{f}_n^-$, and we have $-\boldsymbol{\xi} \in \mathcal{V}'$, which implies that

$$\frac{1}{n}\langle \widehat{f}_n^- - f^*, -\boldsymbol{\xi} \rangle \geq \omega(\epsilon_*^2).$$

Combining these last two inequalities we see that (61) indeed holds over all $\mathcal{V}'$, which implies that (19) holds on $\mathcal{V}'$, as desired. $\qquad\square$

It remains to prove Lemma 11.

**Proof of Lemma 11.** First, define

$$F_i(\boldsymbol{\xi}) := 2n^{-1}\sup_{f \in B(f_i)}\sum_{k=1}^{n}(f - f_i)(x_k) \cdot \xi_k.$$

Since $\|f - f_i\|_n \leq 2\|f - f_i\| + O(\epsilon_*) = O(\epsilon_*)$ for all $f \in B(f_i)$, we see that $F_i(\cdot)$ is a $O(\epsilon_* n^{-1/2})$-Lipschitz and also convex function (with respect to the usual Euclidean norm on $\mathbb{R}^n$). Hence, we apply (12) and obtain

$$\Pr_{\boldsymbol{\xi}}\left\{\left|\sup_{f \in B(f_i)} G_{f-f_i} - \mathbb{E}\sup_{f \in B_n(f_i)} G_{f-f_i}\right| \geq t\right\} \leq 2\exp(-cnt^2/\epsilon_*^2). \tag{63}$$

Therefore, by taking a union bound over $1 \leq i \leq |\mathcal{N}|$

$$\Pr_{\boldsymbol{\xi}}\left\{\forall i \in [|\mathcal{N}|] : \left|\sup_{f \in B_n(f_i)} G_{f-f_i} - \mathbb{E}\sup_{f \in B_n(f_i)} G_{f-f_i}\right| \geq t\right\} \leq 2\exp(-cnt^2/\epsilon_*^2 + \log|\mathcal{N}|).$$

Now, recall that $\mathcal{N} := \mathcal{N}(\epsilon_*, \mathcal{F}, \mathbb{P}^{(n)})$ and that by the definition of the minimax rate,

$$\log|\mathcal{N}(\epsilon, |/n \sim \epsilon_*^2.$$

This allows us to choose $t = C\epsilon_*^2$ (for large enough $C > 0$) such that with probability of at least $1 - 2\exp(-cn\epsilon_*^2)$, the following holds:

$$\forall i \in [|\mathcal{N}|] : \left|\sup_{f \in B_n(f_i)} G_{f-f_i} - \mathbb{E}\sup_{f \in B_n(f_i)} G_{f-f_i}\right| \leq C\epsilon_*\sqrt{\log|\mathcal{N}|/n} \leq C_1\epsilon_*^2, \tag{64}$$

which proves (55).

For the second part of the lemma, fix $j \geq 1$, and define

$$F_j(\boldsymbol{\xi}) = \sup_{f \in \mathcal{N} \cap B_n(f^*, j\epsilon_*)} 2G_{f_j - f^*}.$$

Again, it is easy to verify that $F_j(\cdot)$ is convex and $2j\epsilon_* n^{-1/2}$-Lipschitz. Using (12) once again, we obtain that

$$\Pr\left\{\left|\sup_{f \in \mathcal{N} \cap B_n(f^*, j\epsilon_*)} G_{f-f^*} - \mathbb{E}\sup_{f \in \mathcal{N} \cap B_n(f^*, j\epsilon_*)} G_{f-f^*}\right| \geq t\right\} \leq 2\exp(-cn(t/j)^2/\epsilon_*^2).$$

Choosing $t \sim j\epsilon_*/\sqrt{n} \lesssim j\epsilon_*^2$, we obtain

$$\Pr\left\{\left|\sup_{f \in \mathcal{N} \cap B_n(f^*, j\epsilon_*)} G_{f-f^*} - \mathbb{E}\sup_{f \in \mathcal{N} \cap B_n(f^*, j\epsilon_*)} G_{f-f^*}\right| \geq j\epsilon_*^2\right\} \leq 2\exp(-c\epsilon_*^2). \tag{65}$$

Next, by applying the maximal inequality (Lemma 8) over $|\mathcal{N}|$ random variables, and the definition of the minimax rate,

$$\mathbb{E}\sup_{f \in \mathcal{N} \cap B_n(f^*, j\epsilon_*)} G_{f-f^*} \leq Cj\epsilon_*\sqrt{\log|\mathcal{N}|/n} \leq C_1 j\epsilon_*^2. \tag{66}$$

Combining (65) and (66) yields (56), concluding the proof. $\qquad\square$

## 7  Loose Ends

**Lemma 12.** *Under Assumption 1 the following holds:* $\rho_S(\mathbf{X}) \lesssim \max\{\mathcal{I}_L(\mathbf{X}), \epsilon_*^2\}$ *for any realization* $\mathbf{X}$.

**Proof.** It is always the case that $\rho_S(\mathbf{X}) \leq \max\{\mathcal{I}_L(\mathbf{X}), \epsilon_*^2\}$ for any $\mathbf{X}$. Indeed, if (13) holds, then for any estimator $\bar{f}_n$ such that $\boldsymbol{\xi} \mapsto \bar{f}_n(\boldsymbol{\xi})$ is 1-Lipschitz, $\|\boldsymbol{\xi} - \boldsymbol{\xi}'\|_n \lesssim \epsilon_*$ implies that

$$\|\bar{f}_n(\boldsymbol{\xi}') - \bar{f}_n(\boldsymbol{\xi})\|^2 \leq 2(\|\bar{f}_n(\boldsymbol{\xi}') - \bar{f}_n(\boldsymbol{\xi})\|_n^2 + \mathcal{I}_L(\mathbf{X})) \leq 2(\|\boldsymbol{\xi}' - \boldsymbol{\xi}\|_n^2 + \mathcal{I}_L(\mathbf{X})) \lesssim \max\{\epsilon_*^2, \mathcal{I}_L(\mathbf{X})\}$$

deterministically, not just with non-negligible probability. $\qquad\square$

**Lemma 13.** *Under Assumptions 1, 6, 7, we have that*

$$\rho_{\mathbf{S}}(\mathbf{X}, f^*) \le \rho_{\mathcal{O}}(n, \mathbb{P}, f) \lesssim \max\{\mathcal{I}_L(n, \mathbb{P}), \epsilon_*^2\}.$$

**Proof.** Let $\mathcal{E} \subset \mathcal{X}^n \times \mathbb{R}^n$ be the event that $\mathrm{diam}_{\mathbb{P}}(\mathcal{O}_{M'\epsilon^2}) \le \rho_{\mathcal{O}}(n, \mathbb{P}, f^*)$ and $\|\widehat{f}_n(\mathbf{X}, \boldsymbol{\xi}) - \mathbf{Y}\|_n^2 \le C_I \epsilon_*^2$. For any $\mathbf{X} \in \mathcal{X}^n$, let $\mathcal{E}_{\mathbf{X}} = \{\boldsymbol{\xi} \in \mathbb{R}^n \,|\, (\mathbf{X}, \boldsymbol{\xi}) \in \mathcal{E}\}$.

Since $\Pr(\mathcal{E}) \ge \exp(-c_I n \epsilon_*^2)$ by definition, the set

$$\mathcal{E}' = \{\mathbf{X} \in \mathcal{X}^n \,|\, \Pr_{\boldsymbol{\xi}}(\mathcal{E}_{\mathbf{X}}) \le \rho_{\mathcal{O}}(n, \mathbb{P}, f^*)) \ge \exp(-c_I n \epsilon_*^2) \in \mathcal{E}\}$$

satisfies $\Pr_{\mathcal{X}^n}(\mathcal{E}') \ge \exp(-(c_I/2)n \epsilon_*^2)$ by Fubini's theorem.

Fix $\mathbf{X} \in \mathcal{E}', \boldsymbol{\xi} \in \mathcal{E}_{\mathbf{X}}$, and let $\mathbf{Y} = f^* + \boldsymbol{\xi}|_{\mathbf{X}}$ as usual. If $\|\boldsymbol{\xi}' - \boldsymbol{\xi}\| \le \frac{\sqrt{M'-C_I}}{2}\epsilon_*$ then

$$\|\widehat{f}_n(\mathbf{X}, \boldsymbol{\xi}') - \mathbf{Y}\|_n^2 \le 2(\|\widehat{f}_n(\mathbf{X}, \boldsymbol{\xi}') - \widehat{f}_n(\mathbf{X}, \boldsymbol{\xi})\|_n^2 + \|\widehat{f}_n(\mathbf{X}, \xi) - \mathbf{Y}\|_n^2)$$
$$\le 2((M'-C_I)\epsilon_*^2 + C_I)\epsilon_*^2 = M'\epsilon_*^2$$

where we have used $\|\widehat{f}_n(\mathbf{X}, \xi') - \widehat{f}_n(\boldsymbol{\xi})\|_n \le \|\boldsymbol{\xi} - \boldsymbol{\xi}'\|_n$ as $\widehat{f}_n$ is 1-Lipschitz in the noise. In particular $\widehat{f}_n(\mathbf{X}, \boldsymbol{\xi}') \in \mathcal{O}_{M'\epsilon^2}$ and so $\|\widehat{f}_n(\mathbf{X}, \boldsymbol{\xi}') - \widehat{f}_n(\mathbf{X}, \boldsymbol{\xi})\| \le \rho_{\mathcal{O}}(n, \mathbb{P}, f^*)$, as $(\mathbf{X}, \boldsymbol{\xi}) \in \mathcal{E}$. Thus, if $M'$ is chosen large enough so that $M \le \frac{\sqrt{M'-C_I}}{2}\epsilon_*$, one obtains (13) is satisfied with $\delta(n) = \rho_{\mathcal{O}}(n, \mathbb{P}, f)$ and $c_2 = c_I/2$, implying that $\rho_{\mathbf{S}}(\mathbf{X}, f^*) \le \rho_{\mathcal{O}}(n, \mathbb{P}, f)$.

To see that $\rho_{\mathcal{O}}(n, \mathbb{P}, f^*) \lesssim \max\{\mathcal{I}_L(n, \mathbb{P}), \epsilon_*^2\}$ is even easier: it's easy to see that $\mathrm{diam}_{\mathbb{P}_n}(\mathcal{O}_{M'\epsilon_*^2}) \lesssim \epsilon_*$ (see the end of the proof of Theorem 1 for details), and the definition of the lower isometry remainder implies that $\mathrm{diam}_{\mathbb{P}}(\mathcal{O}_{M'\epsilon_*^2}) \lesssim \mathrm{diam}_{\mathbb{P}}(\mathcal{O}_{M'\epsilon_*^2}) + \sqrt{\mathcal{I}_L(n, \mathbb{P})}$ on the high-probability event $\mathcal{I}_L(\mathbf{X}) \le \mathcal{I}_L(n, \mathbb{P})$. This yields that for an appropriate choice of $C > 0$, $\delta(n) = C \max\{\mathcal{I}_L(n, \mathbb{P}), \epsilon_*^2\}$ satisfies (14) and hence $\rho_{\mathcal{O}}(n, \mathbb{P}, f^*) \le \max\{\mathcal{I}_L(n, \mathbb{P}), \epsilon_*^2\}$. $\square$

## 7.1 Full proof of Corollary 2

Note that when the convex set

$$\mathcal{F}_n := \{(f(X_1), \ldots, f(X_n)) : f \in \mathcal{F}\}$$

is not compact, we cannot apply the fixed point theorem directly. However, if we find $f^* \in \mathcal{F}$ such that

$$B^2(\widehat{f}_n) \le \max\{C_1 \cdot V(\widehat{f}_n), C_2/n\}, \tag{67}$$

where $C > 0$ is some absolute constant, then the proof follows from the argument of §6.3, since the fixed point theorem was only used to find a $f^* \in \mathcal{F}$ satisfying the last equation.

First let us provide some intuition to our proof. The idea is to find $f^* \in \mathcal{F}$ with low bias by an iterative argument similar to the proof of Banach's fixed point theorem. If $\widehat{f}_n$ has a "high" bias on some underlying $f_0 \in \mathcal{F}$, then, $\widehat{f}_n$ should have a lower or equal bias when the underlying function is $f_1 = \mathbb{E}_0 \widehat{f}_n$, where $\mathbb{E}_0$ means taking expectation when $f^* = f_0$. If $f_1$ has a "low" bias, then we are done. Otherwise, consider the underlying $f_2 = \mathbb{E}_1 \widehat{f}_n$, and repeat this process for $n$ times. We will show that some $m \le n$, $f^* = f_m$ will be our "admissible" function.

This idea is captured in the following lemma (that we will prove below):

**Lemma 14.** *Let $f_0 \in \mathcal{F}$ and for any $i \ge 1$ denote by $f_i = \mathbb{E}_{i-1} \widehat{f}_n$. Then, there exists $m = O(n)$ such that*

$$B^2(\widehat{f}_n^{(m)}) \le 3 \cdot V(\widehat{f}_n^{(m)}) + C/n, \tag{68}$$

*where $\widehat{f}_n^{(m)}$ is the ERM when $f^* = f_m$.*

This certainly implies that $B^2(\widehat{f}_n) \le \max\{C_1 \cdot V(\widehat{f}_n), C_2/n\}$, so this lemma is all we need to complete the proof of Corollary 2.

**Proof of Lemma 14.** First, recall that

$$\widehat{f}_n = \underset{f \in \mathcal{F}}{\operatorname{argmax}}(2\langle \boldsymbol{\xi}, f - f^* \rangle_n - \|f - f^*\|_n^2) = \underset{f \in \mathcal{F}}{\operatorname{argmax}} \|f - (f^* + \xi)\|_n^2 - \|\xi\|_n^2. \quad (69)$$

For each $(\boldsymbol{\xi}, f^*) \in \mathbb{R}^{2n}$ define the score function $L_{\boldsymbol{\xi}, f^*} : \mathbb{R}^n \to \mathbb{R}^+$

$$L_{\boldsymbol{\xi}, f^*}(f) = (2\langle \boldsymbol{\xi}, f - f^* \rangle_n - \|f - f^*\|_n^2.$$

and let $L_{f^*}(\widehat{f}_n) = \mathbb{E}L_{\boldsymbol{\xi}}(\widehat{f}_n)$. Note that as $\boldsymbol{\xi}$ is isotropic Gaussian then

$$1 - O(1/\sqrt{n}) \le L_{f^*}(\widehat{f}_n) \le 2 + O(1/\sqrt{n}).$$

**Claim.** For every $f^* \in \mathcal{F}$, either

$$B^2(\widehat{f}_n) \le 3V(\widehat{f}_n) + C/n \quad (70)$$

or

$$L_{\mathbb{E}\widehat{f}_n}(\widehat{f}_n) - \mathbb{E}L_{f^*}(\widehat{f}_n) \ge \frac{B^2(\widehat{f}_n)}{3}. \quad (71)$$

The lemma follows from the claim by an iterative argument. Indeed, let $f_0 = f^* \in \mathcal{F}$ be some function. If $f_0$ satisfies (70), we are done. Otherwise, $f_1 = \mathbb{E}\widehat{f}_n$ satisfies $L_{f_1}(\widehat{f}_n) - L_{f_0}(\widehat{f}_n) \ge \frac{C'}{n}$. if $f_1$ satisfies (68), then we stop. Otherwise, repeat the same argument with $f_1$ and $f_2 = \mathbb{E}\widehat{f}_n(f_1)$, and so on. Since the score $L_f(\widehat{f}_n)$ is bounded above by a constant, eventually some $f_m$ with $m \le n/C$ will have to satisfy (68). It thus remains to prove the claim.

**Proof of the claim** Suppose that (70) does not hold, i.e., $B^2(\widehat{f}_n) \le 3V(\widehat{f}_n) + C/n$. Write $f_0 = f^*$, $f_1 = \mathbb{E}\widehat{f}_n$, set $\tilde{V}(\widehat{f}_n) := \max\{3V(\widehat{f}_n), C/n\}$ and let $\tilde{f}_n$ be the restricted LS on

$$\mathcal{G} := B_n(f_1, \sqrt{\tilde{V}(\widehat{f}_n)}) = \{f \in \mathcal{F} : \|f - f_1\|_n^2 \le \tilde{V}(\widehat{f}_n)\},$$

namely, $\tilde{f}_n := \operatorname{argmin}_{f \in \mathcal{G}} \|\mathbf{Y} - f\|_n^2$.

Using (69), we have that

$$L_{\boldsymbol{\xi}, f_1}(\widehat{f}_n) \ge L_{\boldsymbol{\xi}, f_1}(\tilde{f}_n) = \sup_{f \in \mathcal{G}} 2\langle \boldsymbol{\xi}, f - f_1 \rangle_n - \|f - f_1\|_n^2 \ge \sup_{f \in \mathcal{G}} 2\langle \boldsymbol{\xi}, f - f_1 \rangle_n - \tilde{V}^2(\widehat{f}_n),$$

where the first inequality follows from $\mathcal{G} \subseteq \mathcal{F}$, and the second equality follows from the fact that $\operatorname{Diam}(\mathcal{G})^2 \le \tilde{V}(\widehat{f}_n)$.

By Chebyshev's inequality, with probability at least $2/3$, we have that $\widehat{f}_n \in \mathcal{G}$; we denote this event by $\mathcal{E}$. Under this event, we have that

$$\sup_{f \in \mathcal{G}} \langle \boldsymbol{\xi}, f \rangle_n \ge \langle \boldsymbol{\xi}, \widehat{f}_n \rangle_n.$$

Hence, using the last two equations, the following holds on the event $\mathcal{E}$:

$$L_{\boldsymbol{\xi}, f_1}(\widehat{f}_n) - L_{\boldsymbol{\xi}, f_0}(\widehat{f}_n) \ge \tilde{L}_{\boldsymbol{\xi}, f_1}(\tilde{f}_n) - L_{\boldsymbol{\xi}, f_0}(\widehat{f}_n)$$
$$\ge \|\widehat{f}_n - f_0\|_n^2 - \langle \boldsymbol{\xi}, f_1 - f_0 \rangle_n - \tilde{V}(\widehat{f}_n).$$

Next , note that $\langle \boldsymbol{\xi}, f_1 - f_0 \rangle_n \sim N(0, \|f_1 - f_0\|_n^2) = N(0, B^2(\widehat{f}_n))$. Hence, there exists an event $\mathcal{E}_1 \subset \mathcal{E}$ that holds with probability of at least $0.6$ such that

$$L_{\boldsymbol{\xi}, f_1}(\widehat{f}_n) - L_{\boldsymbol{\xi}, f_0}(\widehat{f}_n) \ge \|\widehat{f}_n - f_0\|_n^2 - \tilde{V}_0(\widehat{f}_n) - C \cdot \sqrt{B_0^2(\widehat{f}_n)/n}. \quad (72)$$

Next, using the fact that $\|\widehat{f}_n - f_0\|_n$ is $1/\sqrt{n}$ Lipschitz and the LCP inequality (12), we know that $\|\widehat{f}_n - f_0\|_n - \mathbb{E}\|\widehat{f}_n - f_0\|_n$ is zero mean and $1/\sqrt{n}$ sub-Gaussian, so by a standard tail integration, one has
$$\mathbb{E}_0\|\widehat{f}_n - f_0\|_n^2 = \mathbb{E}_0\|\widehat{f}_n - f_0\|_n^2 + O(\max\{\mathbb{E}_0\|\widehat{f}_n - f_0\|_n/\sqrt{n}, 1/n\}).$$

Finally, we take a median over (72), and use the last equation and obtain:

$$L_{f_1}(\widehat{f}_n) - L_{f_0}(\widehat{f}_n) \geq \mathbb{E}_0\|\widehat{f}_n - f_0\|_n^2 - 3V_0^2(\widehat{f}_n) - O(\max\{\sqrt{B_0^2(\widehat{f}_n)/n}, \mathbb{E}_0\|\widehat{f}_n - f_0\|_n/\sqrt{n}, 1/n\})$$
$$= B_0^2(\widehat{f}_n) + V_0^2(\widehat{f}_n) - 3V_0^2(\widehat{f}_n) - O(\max\{\mathbb{E}_0\|\widehat{f}_n - f_0\|_n/\sqrt{n}, 1/n\})$$
$$= B_0^2(\widehat{f}_n) - 2V_0^2(\widehat{f}_n) - O(\max\{\mathbb{E}_0\|\widehat{f}_n - f_0\|_n/\sqrt{n}, 1/n\})$$
$$\geq B^2(\widehat{f}_n)/3,$$

where we used the assumption $B_0^2(\widehat{f}_n) \geq 3V_0(\widehat{f}_n) + C/n$, for $C$ that is large enough; the claim follows. $\qquad\square$

## 7.2 Addendum to §5.1

**Example 1.** Let $\mathcal{X} = \mathbb{S}^n$, the Euclidean unit sphere of dimension $n$ contained in $\mathbb{R}^{n+1}$, let $\mathbb{P}$ be the uniform measure on $\mathcal{X}$, and for each hyperplane $H$ passing through the origin, let $\mathbb{P}^H$ be the uniform measure on $\mathcal{X} \cap H \cong \mathbb{S}^{n-1}$. The set of such hyperplanes is the real $(n+1, n)$ Grassmanian, denoted $\mathrm{Gr}_{n+1,n}$.

For each $H$, let $1_H$ denote the characteristic function of $H$, and let $\mathcal{F} = \mathrm{conv}\{1_H, 1 - 1_H : H \in \mathrm{Gr}_{n+1,n}\}$. Note that in $L^2(\mathbb{P})$, $\mathcal{F}$ reduces to the class of constant functions between 0 and 1 because for any $H$, $1 - 1_H \equiv 0$ and $1 - 1_H \equiv 1$ almost everywhere on $\mathcal{X}$. Similarly, in $L^2(\mathbb{P}^H)$, $1_H \equiv 1$ and $1 - 1_H \equiv 0$, while for any $H' \neq H$, $1'_H \equiv 1$ and $1 - 1_H \equiv 0$ because $H \cap H'$ has $n$-dimensional Hausdorff measure 0. In particular, regressing $\mathcal{F}$ on $L^2(\mathbb{P})$ or $L^2(\mathbb{P}^H)$ reduces to estimating an element of $[0, 1]$ given $n$ noisy observations, for which the minimax rate is $\frac{1}{n}$.

On the other hand, let $\mathcal{P} = \{\mathbb{P}\} \cup \{\mathbb{P}^H : H \in \mathrm{Gr}_{n+1,n}\}$, and consider the distribution-unaware minimax risk $\mathcal{M}_n^{(du)}(\mathcal{F}, \mathcal{P})$. We claim that $\mathcal{M}_n^{(du)}(\mathcal{F}, \mathcal{P}) = \omega(1)$ *even when there is no noise*. The intuition behind this is that when the estimator $\bar{f}_n$ observes $(X_1, Y_1), \ldots, (X_n, Y_n)$, it does not "know" whether the distribution is $\mathbb{P}$ or $\mathbb{P}^H$ where $H = \mathrm{Span}(X_1, \ldots, X_n)$, and thus it does not know whether to generalize the observations in a way which is consistent with the $L^2(\mathbb{P})$ norm or the $L^2(\mathbb{P}^H)$ norm.

To see this formally, let $\bar{f}_n : \mathcal{D} \to \mathcal{F}$ be some estimator. For any $\mathbf{X} \in \mathcal{X}^n$, let $\mathrm{span}(\mathbf{X}) := \mathrm{span}(X_1, \ldots, X_n)$, which is an element of $\mathrm{Gr}_{n+1,n}$ with probability 1. In the noiseless setting, any sample $\mathbf{X}, \mathbf{Y}$ such that $\mathrm{span}(\mathbf{X}) \in \mathrm{Gr}_{n+1,n}$ will have $\mathbf{Y}$ a multiple of $(1, \ldots, 1)$ with probability 1, so we may as well consider $\mathbf{Y}$ to be a constant function. To get a lower bound on the minimax rate, it is also sufficient to consider only the extreme points of $\mathcal{F}$, namely the functions in $\mathcal{F}' = \{1_H, 1 - 1_H : H \in \mathrm{Gr}_{n+1,n}\}$, which are $\{0, 1\}$-functions, so we think of our estimators as functions $\bar{f}_n : \mathcal{X}^n \times \{0, 1\} \to \mathcal{F}$. We also assume for simplicity that $\bar{f}_n$ always returns a function in $\mathcal{F}'$; if $\bar{f}_n$ is allowed to take values in the full convex hull $\mathcal{F}$, one obtains the same lower bound on the risk of $\bar{f}_n$ by a more complicated version of the argument below.

Suppose, for example, that the estimator is given the sample $(\mathbf{X}, 1)$. In the noiseless setting, there is no point in returning a function inconsistent with the observations, so $\bar{f}_n$ must return either $1_H$ for $H = \mathrm{span}(\mathbf{X})$ or $1 - 1_{H'}$ for some $H' \neq H$ (not containing any of the $X_i$). Similarly, on the sample $(\mathbf{X}, 0)$ an optimal estimator $\bar{f}_n$ will return either $1 - 1_H$ or $1_{H'}$ for some $H' \neq H$. Let

$$p_{H,0} = \Pr_{X_i \sim \mathbb{P}^H}\{\bar{f}_n(\mathbf{X}, 0) = 1 - 1_H\}$$
$$p_{H,1} = \Pr_{X_i \sim \mathbb{P}^H}\{\bar{f}_n(\mathbf{X}, 1) = 1_H\}$$
$$p_0 = \Pr_{X_i \sim \mathbb{P}}\{\bar{f}_n(\mathbf{X}, 1) = 1 - 1_{\mathrm{span}(\mathbf{X})}\}$$
$$p_1 = \Pr_{X_i \sim \mathbb{P}}\{\bar{f}_n(\mathbf{X}, 1) = 1_{\mathrm{span}(\mathbf{X})}\}.$$

The Grassmannian $\mathrm{Gr}_{n+1,n}$, which is itself isomorphic to $\mathbb{S}^n$, has a uniform (i.e., rotationally invariant) probability measure, and one has $p_i = \mathbb{E}_{H \sim U(\mathrm{Gr}_{n+1,n})}[p_{H,i}]$: choosing $n$ points from the unit sphere in $\mathbb{R}^{n+1}$ is the same as choosing a uniform hyperplane and then choosing $n$ points uniformly from that hyperplane, by rotational invariance.

One computes that $p_{H,i}$ and $p_i$ determine the error of $\bar{f}_n$ on $\mathcal{F}'$ as follows:

$$\varepsilon_{f^*,\mathbb{P}^H}^2 := \mathbb{E} \int (\bar{f}_n - f^*)^2 \, d\mathbb{P}^H = \begin{cases} p_{H,0} & f^* \in \{1 - 1_H\} \cup \{1'_H\}_{H' \neq H} \\ p_{H,1} & f^* \in \{1_H\} \cup \{1 - 1'_H\}_{H' \neq H} \end{cases},$$

$$\varepsilon_{f^*,\mathbb{P}}^2 := \mathbb{E} \int (\bar{f}_n - f^*)^2 \, d\mathbb{P} = \begin{cases} 1 - p_0 & f^* \in \{1_H\}_{H \in \mathrm{Gr}_{n+1,n}} \\ 1 - p_1 & f^* \in \{1 - 1_H\}_{H \in \mathrm{Gr}_{n+1,n}} \end{cases}.$$

To lower-bound the distribution-unaware minimax risk, we consider the expected error of $\bar{f}_n$ under two different scenarios: when we choose a hyperplane $H$ uniformly at random and measure the error of $\bar{f}_n$ on the function $1_H$ when the input distribution is $\mathbb{P}^H$, and when we fix $f^* = 1 - 1_{H_0}$ and measure the expected error of $\bar{f}_n$ when the distribution is $\mathbb{P}$. By the above, the error in the first scenario is $\mathbb{E}_{H \sim U(\mathrm{Gr}_{n+1,n})}[p_{H,1}] = p_1$, while the error in the second scenario is $1 - p_1$. As $\max\{1 - p_1, p_1\} \geq \frac{1}{2}$, this shows that $\mathcal{M}_n^{(du)}(\mathcal{F}, \mathcal{P}) = \omega(1)$, as desired.

*Remark* 17. Example 1 may seem unnatural, as the measures $\mathbb{P}$ and $\mathbb{P}^H$ are all mutually singular, which leads to the "collapse" of the function class in different ways in $L^2(\mathbb{P})$ and each $L^2(\mathbb{P}^H)$. To exclude such pathology, one might wish to consider only families of distributions all of which are absolutely continuous with respect to some reference measure $\mathbb{P}^{(0)}$. It is not difficult, though, to modify Example 1 in such a way which avoids any measurability issues: for given $n$, let $F = \mathbb{F}_q$ be the finite field of cardinality $q$ for some $q > n$, let $\mathcal{X} = F^{n+1} \backslash \{0\}$, let $\mathbb{P}$ be the uniform probability measure on $\mathcal{X}$, and for each $H$ in the set $\mathrm{Gr}_{n+1,n}(F)$ of $n$-dimensional linear subspaces of $F^{n+1}$, let $\mathbb{P}^H$ be the uniform probability measure on $H \backslash \{0\} \subset \mathcal{X}$, let $\mathcal{F} = \mathrm{conv}\{1_H, 1 - 1_H : H \in \mathrm{Gr}_{n+1,n}(F)\}$, and let $\mathcal{P} = \{\mathbb{P}\} \cup \{\mathbb{P}^H\}_{H \in \mathrm{Gr}_{n+1,n}(F)}$ as above. (Note that all measures in $\mathcal{P}$ are absolutely continuous with respect to $\mathbb{P}$.) As $q > n$, each hyperplane $H$ has $\mathbb{P}$-measure $O(n^{-1})$ in $\mathcal{X}$ and for any $H' \neq H$, $H \cap H'$ has $\mathbb{P}^H$-measure $O(n^{-1})$, so one easily verifies all the computations in the example are still valid, up to errors of order $O(n^{-1})$.