# OpenReview forum: "On the Variance, Admissibility, and Stability of Empirical Risk Minimization"
_NeurIPS.cc/2023/Conference — NeurIPS 2023 spotlight_

### Official Review · Reviewer_hXZP · 2023-07-06

**Soundness:** 3 good
**Presentation:** 3 good
**Contribution:** 4 excellent
**Rating:** 7
**Confidence:** 3

**Summary:**


This paper proves that variance for ERM enjoys a a minimax rate. The findings indicate that in scenarios where ERM is not optimal, the source of suboptimality lies within the bias component. Furthermore, these insights are extended to encompass an admissibility-type theorem for both fixed and random design, as well as a stability result for ERM.
The paper's contributions can be summarized as follows:
1)Demonstrating the optimality of the variance term in both fixed and random design situations.
2)The obtained results automatically yield a stability result for ERM.
3)Presenting a simpler proof for the admissibility theorem in the fixed and random design setting.
4)Highlighting a counterintuitive outcome concerning the ERM estimator $\widehat{f}_n$: when $f'$ is close to $\widehat{f}_n$, it can lead to a large squared error.



**Strengths:**

The paper studies an important problem, provides convincing conclusion. The problem is not only intellectually interesting, it also advances our understanding of ERM, which is arguably the most important estimator for modern machine learning.


**Weaknesses:**

Typo: above Equation 1 $f^* \in f^*$.
Remark should be referred to in the appendix.

**Questions:**

Given dense results here, it would help reader to better understand the paper if the theorems could be summarized in a table.

---

> ### Author Rebuttal · Authors · 2023-08-09
>
> Thank you for your kind report and for pointing out that typo. In the revised version, we will follow your suggestion and embed a tabular summary of the results. We will move many of the remarks to the appendix.

---

> > ### Comment · Reviewer_hXZP · 2023-08-17
> > **The rebuttal solves all my concerns**
> >
> > The rebuttal solves all my concerns

---

### Official Review · Reviewer_ZyPC · 2023-07-06

**Soundness:** 3 good
**Presentation:** 3 good
**Contribution:** 3 good
**Rating:** 7
**Confidence:** 3

**Summary:**

This paper explores the minimax optimality of ERM in terms of the bias and variance of the ERM method.
They find in some settings that the variance is always at the minimax rate, implying that suboptimality can only occur due to bias. This paper also explores stability of ERM, finding that almost-minimizers are close to the ERM w.r.t the population distribution, but not necessarily the empirical distribution.

**Strengths:**

* Cool results and proofs in the fixed design setting
* The random design setting results are very interesting as well, albeit harder to digest all the quantities.

**Weaknesses:**

* The independence relations should be clarified in the preliminaries. I assume that $\mathbf{\xi}$ is independent from $\mathbf{X}$ in the random design case, but I don't think that's mentioned.



Minor Typos:
* In the proof of Theorem 2, $\mathcal{G}$ and $\mathcal{G}_\star$ seem to be used interchangeably.
* The statement of Lemma 2 in the proof of Theorem 2 seems to have a typo? I think it should be $f$ instead of $f^\star$.

**Questions:**

* Given that assumption 8 is necessary to get from a $\epsilon_V^2$ to an $\epsilon_\star^2$ bound, how restrictive is this assumption?

**Limitations:**

The authors have adequately addressed the limitations.

---

> ### Author Rebuttal · Authors · 2023-08-09
>
> Thank you for your kind report and your typo fixes. We have clarified that the noise vector is drawn independently of the data points.
>
> Regarding Assumption 8, is motivated by  high-dimensional models, especially in the ``benign overfitting'' literature where we they assume that ERM interpolates the noisy observations. However, there are function classes which do not satisfy it. As mentioned below its statement, the assumption cannot hold if $\mathcal F$ is taken to be a compact class of functions on some compact domain $\mathcal X \subset \mathbb R^d$. For instance, if $\mathcal F$ is the class of $1$-Lipschitz functions, $d$ must increase at least logarithmically in $n$.

---

> > ### Comment · Reviewer_ZyPC · 2023-08-19
> > **Reply to authors**
> >
> > Thank you for addressing my questions. I will maintain my score.

---

### Official Review · Reviewer_DCZr · 2023-07-08

**Soundness:** 3 good
**Presentation:** 3 good
**Contribution:** 3 good
**Rating:** 7
**Confidence:** 3

**Summary:**

This paper studies the suboptimality of Empirical Risk minimization (ERM) of the squared loss, or equivalently, Least Squares (LS), for convex function classes, in both fixed and random design settings. In the context of non-parametric statistics, necessary and sufficient conditions for the optimality of LS are not known (though sufficient conditions were given by Birge and Massart, '93).
Roughly speaking, the main message of this paper is showing in both settings that the suboptimality is due to the bias term, where the variance term enjoys minimax rates, under some reasonable assumptions.



**Strengths:**

This paper studies a classic and fundamental question in statistics, understanding the suboptimality of Least Squares.
This paper makes nice progress in this direction. Besides the main result (mentioned in the summary), there are several other contributions, such as the (weak) admissibility property of the ERM in the random design setting; sometimes ERM can be optimal.
Moreover, one of the results shows that ERM is stable; all approximate minimizers of the squared loss are close to each other in the functions space, and on the other end, the converse doesn't hold, sometimes the ERM is an optimal estimate and there exist close functions to it with high empirical error.

Also, the proof techniques (the isoperimetry approach for example) look interesting and elegant.

**Weaknesses:**

I don't see any.
There is a chance that I didn't understand some details, this is not my main research field and some proof techniques were new to me.

**Questions:**

Typos:
line 69: should be $f*\in F$.

line 16: should be "in detail".

**Limitations:**

I don't see any.

---

> ### Author Rebuttal · Authors · 2023-08-09
>
> Thank you for your kind report and your typo fixes.

---

### Official Review · Reviewer_vswn · 2023-07-09

**Soundness:** 3 good
**Presentation:** 2 fair
**Contribution:** 2 fair
**Rating:** 6
**Confidence:** 2

**Summary:**

The paper considers  the Empirical Risk Minimization (ERM) problem with squared loss and shows that the suboptimality of ERM is due to the bias rather than variance.

**Strengths:**

This paper is quite theoretical and technical. The paper provides useful insights in different aspects. The paper finds that (1) the variance term of ERM agrees with the minimax rate of estimation in both fixed and random design, (2) ERM is stable, (3) the landscape of near-solution for non-Donsker class is irregular, (4) the relation between isoperimetry and variance of ERM.


**Weaknesses:**

The main weakness is organization. Too many results wrapped in nine pages makes reader feel the paper is very crowded.

Some definitions and results could be explained more detailedly. For example, what does it mean by the minimax rate of ERM and variance, respectively? What is local optimality? Donsker and non-donsker classes. Any specific application of current results? Technical differences between fixed and random design could be explained on a very high level.

**Questions:**

I found that this paper is more suitable for conference like COLT instead of NIPS. Audience from learning theory filed might be more interested in this topic.

**Limitations:**

More examples and applications should be included. Otherwise, it is a merely technical paper with no practical usage.

---

> ### Author Rebuttal · Authors · 2023-08-09
>
>  Thank you for your report. We agree that the intuitive explanations of our assumptions and results are a bit briefer than we would like, which was forced upon us by the page limit for the submission.  In the revised version, we will move Theorem 2 to the appendix, along with the additional page in the revised version, to provide fuller explanations to improve the accessibility of this manuscript.
>
> - Applications: our results are applicable to the ERM under extremely mild assumptions on the function class, as well as to other estimators satisfying Lipschitz conditions, such as regularized least-squares. A wide variety of problems intensively studied in non-parametric statistics, from estimation of a bounded $L$-Lipschitz or convex function, to kernel regression, satisfy these assumptions.
> - "I found that this paper is more suitable for conference like COLT....'' - At first glance, this paper may look very theoretical. However, our results reveal that with regard to the large learning models commonly used in practice, if the ERM performs poorly then it does so because it is biased. In contrast (according to our knowledge) to the setting of models with few parameters, efficient debiasing methods for ERM on these large models do not exists (see Item 1 in the general rebuttal). We believe that the main message of our work -- efficient debiasing methods are all you need to improve the statistical performance of ERM over large models -- is insightful for the entire learning community. We moved this takeaway from the end of the paper to the introduction.

---

### Official Review · Reviewer_XB3R · 2023-07-19

**Soundness:** 4 excellent
**Presentation:** 2 fair
**Contribution:** 3 good
**Rating:** 6
**Confidence:** 3

**Summary:**

This paper studies the classic problem of regression under noisy observations, with a convex and closed function class $\mathcal{F}$. In particular, the paper considers the performance of the Empirical Risk Minimizer (ERM) under the mean squared error, in both the fixed design and random design settings.

At a high level, the author(s) show that:
1. The variance of the ERM is comparable to the minimax rate of regression, implying that when ERM is not minimax-optimal, it must be due to large bias
2. ERM is admissible up to constant factors, via a simpler proof using fixed point theorems
3. ERM enjoys stability, in the sense that all almost-empirical-loss-minimizers also have similar expected loss as the actual empirical loss minimizer.
4. For "non-Donsker" function classes, the converse of 3 is false, that there is always some ground truth function $f^* \in \mathcal{F}$ such that, with high probability over the samples and noise, there is a function $f_{\mathrm{bad}}$ whose expected loss close to the ERM, but that its empirical loss is much larger than the ERM's.

I did the following review as an emergency review, so I did not check the details/appendices carefully.

**Strengths:**

The ERM is perhaps the most commonly used regression estimator. This paper furthers our understanding of its behavior, particularly for function classes where ERM isn't minimax-optimal. I found the introduction well-written, describing each result at a high level. I also find the observation that "non-minimax-optimality must be due to bias" to be an interesting result.

**Weaknesses:**

While I liked the paper, and recommend a weak accept, I think there are the following writing/presentational issues that can be improved. Overall, my comment is that the paper currently perhaps reads better for experts who have worked on ERM (or at least, in empirical process theory), but not that easy to read for a more general learning theory audience.

- The paper reads a bit like a collection of related results, but without a "main point", and could maybe be strung together better as a story. For example, the "variance <= minimax-rate" result and the stability result seem somewhat disjoint in the introduction, even though later on in Theorem 1, they are shown together as one big result. I was also a bit confused about the writing in Section 2, in terms of the narrative. For example, I don't quite understand how Theorem 2 "complements" Theorem 1 (cf. Line 150), though Theorem 2 is a cool result itself with the local minimax optimality and is used to prove Corollary 1 (the admissibility result). Moreover, Theorem 3 also "complements" Theorem 1 (cf. Line 185), but there's a quadratic gap. Is the gap necessary?

- I also found that the technical Section 2 is a bit too heavy on definitions and assumptions and not enough interpretation. When I was reading, I felt that the author(s) tried hard to succinctly give the most general results that are shown, but in my opinion, for readability, it might be better to start with simpler cases (and perhaps even informal versions) of statements (particularly the assumptions), and provide more interpretation.

- Related to the above point, some of the assumptions/quantities in the results are stated without much interpretation or intuition (in the main body). There are also a few comments along the lines of "this assumption is considered mild in the literature/by some other authors" without much additional interpretation. This makes the paper not quite as self-contained as it could be. This issue is particularly prevalent in Section 2.2 (especially the isoperimetry assumptions), and it became quite hard to interpret the results. I can see that there are some remarks in the appendices, but I think a lot of them really should be in the main body for readability.

- The term non-Donsker was never defined in the main body, even though it is a key element in a main result. While Donsker classes are a basic object in empirical process theory, I don't think it should be assumed knowledge for the general theory reader. There is also a lack of discussion on whether Theorem 4 only hold for non-Donsker classes, or more generally what's the significance of the assumption: whether it's a necessary condition for the result, or just that this is the result that can be proved.

**Questions:**

Minor questions and comments:
1. Please consider using the same number-counter across all of definitions/theorems/assumptions. It was hard to scroll through the paper looking at backward references when reading.
2. (Line 164) The author(s) mention that $\mathcal{G}_\ast$ in Theorem 2 can replace 2 with any other absolute constant. How does the replacement propagate to Theorem 2? Does it change the constants in the $\asymp$?
3. Theorem 6, the assumptions 1,2,4,6 have nothing to do with the instantiation of $\mathbf{X}$?

---

> ### Author Rebuttal · Authors · 2023-08-09
>
> Thank you for your detailed report. Following your comments as well as those of other reviewers regarding the density of results in the paper, in the revised version, we moved the relatively technical Theorem 2 to the appendix and inserted more explanations/intuition into the main body, which the more generous page limit for the revision enabled us to do.
>
> Regarding some of the specific points raised by this review:
>
> - The minimax optimality of the ERM's variance and its stability are conceptually distinct phenomena and thus explained separately in the introduction, but mathematically they are so closely linked that it is easier to present them together in Theorem 1.
> - Theorem 2 complements Theorem 1 mainly in the sense that they are both needed to prove the ERM's admissibility. In any case, Theorem 2 will be relegated to the appendix in the revision.
> - Regarding the quadratic gap between Theorem 3 and Theorem 1, we do not know if it is necessary, but we do not currently know how to improve it.
> - "The paper reads a bit like a collection of related results, but without a "main point", and could maybe be strung together better as a story." - see Item 1 in the rebuttal for all reviewers. Also, we improved the organization to the paper that the main points and the applications appear before the main results -- that are more techincal in their nature.
> - "The term non-Donsker was never defined in the main body" - we will add a definition and some examples for non-Donsker classes and move this result to the random design section, and provide some more motivation as well.
> - "I also found that the technical Section 2.... Related to the above point, some of the assumptions/quantities in the results are stated without much interpretation'' - Again, we have moved some results to the appendix, which along with the extra page frees up room to add more explanations and concrete examples, especially in the random design setting.

---

> > ### Comment · Reviewer_XB3R · 2023-08-10
> >
> > I thank the authors for their response. I hope the authors will actually implement the changes described, since they all will make the paper a lot more readable and self-contained. My final comment is that the sentence "Beyond mere intellectual curiosity, our results demonstrate the salience of finding computationally efficient methods to debias ERM" from the overall response is useful to emphasize further in the final paper.
> >
> > I will maintain my current score.

---

### Author Rebuttal · Authors · 2023-08-09

We thank all reviewers for their thoughtful comments. We provide the following general remarks and comments for all reviewers and to the area chair:

1. The main contribution/message of the paper:  Empirical Risk Minimization (ERM) with squared loss, or any "Lipschitz" loss in the observations, enjoys a minimax optimal variance error term. Therefore, if the ERM is minimax suboptimal, this is due to its bias. Beyond mere intellectual curiosity, our results demonstrate the salience of finding computationally efficient methods to debias ERM (see the remark below). This finding aligns with a few recent works that demonstrate that ERM (or MLE) suffers from high bias in certain classical models in their high dimensional settings, such as logistic regression or $l_1$ regression. We have proven that the potential high bias of ERM is a general phenomenon over rich function classes or in "high dimensions."

    This takeaway appeared at the end of the manuscript (page 9); in the revised version, we improved the phrasing and placed it more prominently in the introduction section.

    Remark: In the classical regime (i.e., classes with a low number of parameters), one may reduce the bias by using bootstrap methods (see, e.g., B. Efron, "Bootstrap methods: another look at the jackknife," Ann. Statist. 7(1): 1-26). Unfortunately, according to our knowledge, in the non-parametric case or with models with a large number of parameters (such as neural networks), such methods do not work.

2. Clarity and organization of the manuscript: We have enhanced the organization of the manuscript by adding more intuitive explanations for the quantities that appear in the random design, and moved some of the more technical results to the appendix. This change is aimed at improving the accessibility of the paper. Additionally, in line with the reviewers' suggestions, we have incorporated a concise summary of all our results, potentially presented in a table format, to provide readers with a quick overview.

3. Generality of our results: Although we presented our results specifically for Empirical Risk Minimization (ERM) with squared loss, otherwise known as the LS estimator, our findings are more broadly applicable. As indicated in our remarks, many of our results hold for estimators that exhibit certain stability properties, such as Lipschitzness in noise or data. This includes many variants of convex regularizers, such as ridge regression.

---

### Decision · Program_Chairs · 2023-09-21

**Decision:**

Accept (spotlight)

**Comment:**

This paper considers ERM with squared loss, which can be decomposed into a bias term and a variance term, and shows that the variance term necessarily enjoys the minimax rate, therefore any suboptimality must be due to the bias term. The authors then discuss its implications on the admissibility and stability of the ERM estimator with squared loss.

The contributions are solid, and all the reviewers agree that it should be accepted. Maybe a minor concern is that the paper mainly talks about ERM with squared loss, but "squared loss" is not mentioned in the title, which could be a little misleading.